A review of the carotid artery and facial nerve canal systems in extant turtles

Rollot Yann yann.rollot@gmail.com
http://orcid.org/0000-0002-2393-5621 Evers Serjoscha W.
Joyce Walter G.
Department of Geosciences, University of Fribourg , Fribourg , Switzerland
Knoll Fabien
Electronic publication date: 2021 Jan 21
Publication date: 2021
Volume: 8
Electronic Location ID: e10475
Received 2020 Jun 2; Accepted 2020 Nov 11
Copyright: © 2021 Rollot et al.
Copyright year: 2021
Copyright holder: Rollot et al.
License: This is an open access article distributed under the terms of the Creative Commons Attribution License, which permits unrestricted use, distribution, reproduction and adaptation in any medium and for any purpose provided that it is properly attributed. For attribution, the original author(s), title, publication source (PeerJ) and either DOI or URL of the article must be cited.
License URL: https://creativecommons.org/licenses/by/4.0/

Keywords: Testudines, Carotid artery, Facial nerve, Interspecific variation

Funding: Swiss National Science Foundation SNF 200021_178780/1 This study was funded by Swiss National Science Foundation grant SNF 200021_178780/1. There was no additional external funding received for this study. The funders had no role in study design, data collection and analysis, decision to publish, or preparation of the manuscript.

==============================
The cranial circulation and innervation systems of turtles have been studied for more than two centuries and extensively used to understand turtle systematics. Although a significant number of studies related to these structures exists, a broader comprehension of variation across the tree has been hindered by poor sampling and a lack of synthetic studies that addressed both systems together. We here provide new insights regarding the carotid circulation and facial nerve innervation systems in a broad set of extant turtles using CT (computed tomography) scans, which allow us to trace the canals these structures form in bone and understand the interaction between both systems. We document that the palatine artery, including the lateral carotid canal, is absent in all pleurodires and carettochelyids and was likely reduced or lost several times independently within Testudinoidea. We also highlight osteological correlates for the location of the mandibular artery. We finally summarize variation regarding the placement of the mandibular artery, location of the geniculate ganglion, placement of the hyomandibular and vidian nerves, and situations where we recommend caution when assessing canals in fossils. A morphometric study confirms that the relative sizes of the carotid canals are correlated with one another. Our results have the potential for building new phylogenetic characters and investigating the circulation systems of fossil taxa, which are expected to shed light on the evolution of the circulation system of turtles and clarify some unresolved relationships between fossil turtle clades.

Introduction

The vertebrate head is the primary body part for sensory perception and interaction with the environment. Over the course of the last two centuries, two systems, the cranial nerve innervation and the cranial arterial circulation, have been studied extensively across extant and extinct tetrapods (Bojanus, 1819–1821; Haughton, 1929; Shishkin, 1968; Lawson, 1970; Starck, 1979; Evans, 1987; O’Keefe, 2001; Müller, Sterli & Anquetin, 2011; Pardo & Anderson, 2016). The cranial innervation system provides sensory cues to the brain while the carotid circulation system provides blood to sensory and other organs. Therefore, differences in cranial blood supply and innervation can give clues about the sensory capacities of animals (e.g., mechanoreception; Dehnhardt & Mauck, 2008; Muchlinski, 2008; George & Holliday, 2013), their physiology (e.g., thermoregulation; Porter & Witmer, 2015; Yu, Ashwell & Shulruf, 2019), the presence and type of cranial soft tissues in fossils (Benoit et al., 2018), as well as the evolution of these features along stem-lineages (Benoit, Manger & Rubidge, 2016; Benoit et al., 2018; Joyce, Volpato & Rollot, 2018; Evers, Barrett & Benson, 2019). Despite showing specific adaptations that often relate to ecological specializations, both the nervous and carotid systems are relatively conservative between major amniote clades in terms of their morphological evolution (Kardong, 1998; Müller, Sterli & Anquetin, 2011) and can therefore be used to examine variational patterns along deep diverging lineages, possibly providing phylogenetic character support for specific divergences. Importantly, arteries and nerves are commonly housed in bony canals for at least parts of their courses, and can thus be identified and analyzed in specimens that lack soft tissues, such as dry museum specimens, or fossils. Therefore, phylogenetic approaches often focused on indirect evidence for the arterial circulation and cranial innervation, namely the respective bony canals, which can readily be compared across large specimen samples, including fossils. The difficulty here lies with the correct identification (and thus, homology) of said canals. Within turtles, variation pertaining to the carotid arterial system has been used as a source of phylogenetic characters since the seminal study of Gaffney (1975). On a gross scale, differences have been found between early shelled stem-turtles such as Proganochelys quenstedtii from the Late Triassic (Gaffney, 1990) and crown-group turtles (Sterli & De la Fuente, 2010; Rabi et al., 2013). Within crown Testudines, the two major surviving clades, Pleurodira and Cryptodira, have been found to have different carotid circulation patterns (Albrecht, 1967, 1976; Gaffney, 1975, 1979). Additionally, systematic differences between cryptodires and pleurodires have been observed for the facial nerve system, which, for turtles, appears to have experienced greater evolutionary change over time than other cranial nerves (Gaffney, 1975, 1979). Parts of the facial nerve share the same course as parts of the carotid arterial system (Albrecht, 1967, 1976; Gaffney, 1979; Miyashita, 2012). The shared course of these systems potentially offers an explanation for the relatively greater variability of the facial nerve system. In the past, most descriptive turtle studies have focused on either carotid circulation (Sterli & De la Fuente, 2010) or cranial innervation (Evers, Barrett & Benson, 2019) of single species, but seldom on both (Miyashita, 2012). Additionally, the evolutionary understanding of the circulation system has been hampered by misidentifications resulting from interpreting the carotid arterial system without explicit reference to the facial nerve system, especially in fossils. This is exemplified by Eubaena cephalica, in which a canal for the palatine artery had been persistently identified, but for whom CT (computed tomography) scans have recently shown that this canal was actually absent and that the only canal present in that area is the canal for the vidian nerve (Rollot, Lyson & Joyce, 2018).

An overview of cranial carotid arterial circulation

The main branching patterns of the carotid arterial circulation are conserved across amniotes (see Fig. 1 for a basic scheme of the cranial circulation and facial innervation of turtles). The internal carotid artery arises from the common carotid artery and splits into a stapedial branch and a cerebral branch (e.g., Sedlmayr, 2002 for archosaurs including birds; Porter & Witmer, 2015 for squamates; Porter, Sedlmayr & Witmer, 2016 for crocodilians; and Müller, Sterli & Anquetin, 2011 for an amniote overview including stem-group taxa). In turtles, the cerebral branch retains the name “internal carotid artery ” (McDowell, 1961; Gaffney, 1979; Sterli et al., 2010; Rabi et al., 2013). The cerebral/internal carotid artery variously splits into anteriorly directed arterial branches (“palatine artery” in turtles: Gaffney, 1979) (Sedlmayr, 2002; Porter & Witmer, 2015), but otherwise enters and traverses the basicranium through the basisphenoid to exit into the pituitary fossa/sella turcica to supply blood to the brain (Sedlmayr, 2002; Müller, Sterli & Anquetin, 2011; Porter & Witmer, 2015; Porter, Sedlmayr & Witmer, 2016). Additional anteriorly directed arteries, such as the infraorbital artery, may also branch off the cerebral artery within the braincase (Porter & Witmer, 2015).

Figure 1 Basic scheme of internal carotid artery, lateral head vein, and facial nerve systems in turtles.

Despite this conserved pattern of splitting into subordinate arteries, homology has not been perfectly translated into nomenclature, which makes its understanding difficult. For instance, in squamates and turtles, the artery that supplies the dorsal head region splits extracranially into two major branches, one of which is the stapedial artery. The other branch enters the basicranium and splits again into two subordinate arteries, one that supplies blood to the brain, and one that typically supplies the facial region. In the turtle literature, the term “internal carotid artery” is used to refer to the arterial section that lies between the branching point of the stapedial artery, and the intracranial split (McDowell, 1961; Gaffney, 1979; Sterli et al., 2010; Rabi et al., 2013). Thus, in turtles, the “internal carotid artery” branches into (i) the “cerebral artery” that traverses the basisphenoid and enters the pituitary fossa to supply the brain with blood, and (ii) the “palatine artery” that traverses the basicranium anteriorly toward the facial region. In squamates, however, the term “internal carotid artery” is used for the carotid section prior to the divergence of the stapedial artery (Oelrich, 1956; Porter & Witmer, 2015). The second branch that forms at this divergence is called the “cerebral artery”. The “cerebral artery” of squamates (= “internal carotid artery” of turtles) then branches into the “sphenopalatine artery” (= “palatine artery” of turtles), and keeps its name for the remainder of its course into the brain (= “cerebral artery” of turtles).

In this study, we use the classic turtle nomenclature with regard to the splitting pattern of carotids, which gained influence primarily by the works of Albrecht (1967, 1976) and Gaffney (1972, 1979). Under this nomenclature, the internal carotid artery and stapedial artery diverge from one another in an extracranial position, with the stapedial artery entering the cavum acustico-jugulare. The mandibular artery is frequently developed as a branch of the stapedial artery, although many exceptions exist across turtles. For instance, the mandibular artery derives from the internal carotid artery in trionychids (Albrecht, 1967). The internal carotid artery extends medially toward the basicranium and commonly, but not ubiquitously, splits into the cerebral and palatine arteries. The cerebral artery is always present and always traverses the basisphenoid and enters the sella turcica/pituitary fossa.

A history of carotid research in turtles

Early work on the carotid arterial system in turtles was mainly limited to the documentation of carotid artery courses in single species (Bojanus, 1819–1821; Nick, 1912; Shindo, 1914), or a selection of related turtles (McDowell, 1961). Later, studies by Albrecht (1967, 1976) were the first synthetic approaches to summarize variation of features related to carotid circulation across clades. Albrecht’s findings were further publicized by the landmark studies of Gaffney, who introduced this character complex to turtle phylogenetics (Gaffney, 1975; Gaffney & Meylan, 1988), and who advocated for the use of osteological correlates of the arteries, namely the canals and foramina, to identify carotid features in fossil turtles, particularly by providing terminology (Gaffney, 1972) and comparisons across a large sample of turtles (Gaffney, 1979).

The importance of the cranial circulatory system in systematics has been highlighted now for more than half a century and many studies have added and/or revised characters related to carotids (Gaffney, 1975; Meylan & Gaffney, 1989; Jamniczky, 2008; Sterli & De la Fuente, 2010; Sterli et al., 2010; Rabi et al., 2013; Zhou & Rabi, 2015; Joyce, Volpato & Rollot, 2018; Evers & Benson, 2019; Hermanson et al., 2020), although methods of character implementation have significantly varied between studies (Joyce, 2007; Anquetin, 2012; Evers & Benson, 2019). Several “patterns” of circulation have been proposed, and other evolutionary scenarios have been put forward. For instance, Meylan & Gaffney (1989) postulated the cerebral and palatine arteries of turtles primitively had a similar size and were slightly smaller than the stapedial artery. Gaffney (1975) also proposed that the three major groups of turtles recognized at the time (i.e., pleurodires, cryptodires, and paracryptodires) each display a characteristic pattern for the circulation system, with the discriminating feature being the position of the foramen posterius canalis carotici interni, that is, the foramen through which the internal carotid artery enters the skull. However, most of these inferences were based on only a small sample of dissected taxa obtained during earlier studies (Albrecht, 1967, 1976), and patterns have been extrapolated from the few taxa studied to entire turtle groups (Gaffney, 1979; Jamniczky & Russell, 2007), without further verification.

Paleontologists have been reporting patterns of cranial arterial circulation and innervation for extinct species. These studies recognized that the observed features cannot easily be fit into the framework established on the few extant turtles studied in the past. For example, although all crown-group turtles have their carotid arterial system fully embedded within the basicranium, fully or partially ventrally-open carotid systems appear in stem-group turtles, but also on the stem-lineages of various crown-group clades (Meylan & Gaffney, 1989; Sterli et al., 2010; Rabi et al., 2013). Rabi et al. (2013) proposed an updated, comprehensive nomenclatural system for the diversity of foramina and canals associated with the carotid system, but this nomenclature was not universally applied in recent turtle literature, despite its advantages. However, although this contribution figuratively cut the Gordian knot of confusing terminology by suggesting an internally consistent nomenclature, we here suggest two minor modifications to the names of the canals that contain the cerebral and palatine arteries (see “Material and Methods” below). The advent of CT-based studies has further led to a re-investigation of variation on a broad taxonomic and sampling scale of extant and extinct turtles (Paulina-Carabajal et al., 2017; Evers & Benson, 2019; Hermanson et al., 2020), challenging some of the more classic character concepts regarding the carotid system.

Facial nerve

Much less recent research has been done on the facial nerve (CN VII), despite the availability of a broad historic literature describing nerve patterns in extant turtles (Bojanus, 1819–1821; Vogt, 1840; Kesteven, 1910; Gaupp, 1888; Hoffmann, 1890; Ruge, 1897; Siebenrock, 1897; Bender, 1906; Noack, 1906; Ogushi, 1911, 1913a, 1913b; Hanson, 1919; Shiino, 1913; Nick, 1912; Shindo, 1914; Fuchs, 1931; Van der Merwe, 1940; Soliman, 1964). The facial nerve branches off the brain of turtles and exits the braincase via the fossa acustico-facialis on the medial surface of the prootic (Fig. 1). From this fossa, it extends laterally through a canal. Further laterally, the facial nerve forms the geniculate ganglion, from which two major nerves emerge, the anteriorly directed vidian/palatine nerve, and the posteriorly directed hyomandibular nerve (Gaffney, 1979). Based on the comprehensive works of Siebenrock (1897), Shiino (1913), Soliman (1964), and Gaffney (1975, 1979) suggested that the position of the geniculate ganglion of the facial nerve differs systematically between cryptodires and pleurodires, with cryptodires having the geniculate ganglion positioned in the canal for the lateral head vein, that is, the canalis cavernosus, and pleurodires having the geniculate ganglion contact the internal carotid artery canal (=canalis caroticus internus). Gaffney (1979) interpretation was based on relatively few specimens, but has largely remained unchallenged and been used in many phylogenetic studies of turtles. Although deviations to Gaffney (1979) clear-cut patterns have been known for a long time (Gaffney, 1983), a more recent, systematic survey of CT scans reveals many more, including a potentially novel position for the geniculate ganglion (Evers & Benson, 2019). As historically terms were used interchangeably in the past, Rollot, Lyson & Joyce (2018) provided a consistent nomenclature for the canals left by the facial nerve system. Here, we investigate the facial nerve pattern on a broad taxonomic scale, and investigate how it is related to the carotid arterial pattern, with which it partially shares canals (Miyashita, 2012).

Aims and objectives

For this contribution, we aim to document the carotid circulation pattern, as well as the course of the facial nerve, for all major clades of extant turtles, with the explicit goal of wishing to clarify if these structures can be traced correctly based on osteological material alone, in particular fossils. Our work is primarily based on high-resolution micro-CT scans of 65 extant turtle species. The identity of canals was interpreted with reference to documented soft tissues (Albrecht, 1967, 1976) whenever this information was available. We focus on the position of foramina, the bones that form the canals and foramina, and the connections between these structures (i.e., canals). In addition to updating the standardized terminology for these features, we address the variation of these systems within and between clades and summarize patterns common to specific clades. We provide evidence for the repeated loss of some branches of the carotid arterial system and re-interpret canals that have been said to house the palatine artery in these turtles as actually housing the vidian nerve, a branch of the facial nerve. Our work provides an important step towards a comprehensive understanding of cranial arterial circulation and facial nerve patterns, which can aid in character construction for phylogenetic analyses. Furthermore, with the inclusion of fossil data, our work will provide the basis for a comprehensive evolutionary understanding of cranial arterial circulation and innervation in turtles.

Materials and Methods

CT-scanning and segmentation

For this study, we broadly sampled across the turtle tree to compile a set of 69 CT scans of the skulls of 65 extant turtle species (see Table 1). The relationships between the major clades of turtles are shown in Fig. 2. The vast majority of scans were previously published and are available at public repositories (Evers & Benson, 2018; Lautenschlager, Ferreira & Werneburg, 2018; Evers, 2019). The models generated are available on Morphobank (http://morphobank.org/permalink/?P3732). The (para)basisphenoid, the right pterygoid, the carotid system, the canalis cavernosus, and the facial nerve system of 14 skulls representing the primary clades (“families”) of extant turtles were segmented manually using the brush and lasso and interpolation tools of Amira 6.4.0 (https://www.fei.com/software) and final models were generated by exporting surfaces of the structures of interest.

Table 1 List of specimens used in this study.

Taxon	Specimen(s) used	Repository	Project	Media accession number	Citation/Source	
Carettochelyidae						
Carettochelys insculpta	NHMUK 1903.7.10.1	MorphoSource	P769	M42840	Evers (2019)	
Carettochelys insculpta	SMF 56626	none	NA	NA	shared by I. Werneburg	
Chelidae						
Chelodina oblonga	NHMUK 64.12.22	MorphoSource	P769	M42849	Evers (2019)	
Chelus fimbriatus	NHMUK 81.9.27.4	MorphoSource	P769	M42856	Evers (2019)	
Emydura subglobosa	PIMUZ lab# 2009.37	none	NA	NA	Lautenschlager, Ferreira & Werneburg (2018)	
Hydromedusa tectifera	SMF 70500	none	NA	NA	shared by I. Werneburg	
Phrynops geoffroanus	SMF 45470	MorphoSource	P462	M22118	Evers & Benson (2018)	
Phrynops hilarii	NHMUK 91.3.16.1	MorphoSource	P769	M42860	Evers (2019)	
Cheloniidae						
Caretta caretta	NHMUK 1938.1.9.1	MorphoSource	P769	M42843	Evers (2019)	
Caretta caretta	NHMUK 1940.3.15.1	MorphoSource	P769	M42844	Evers (2019)	
Chelonia mydas	NHMUK 1969.776	MorphoSource	P769	M42846	Evers (2019)	
Eretmochelys imbricata	FMNH 22242	MorphoSource	P769	M42863	Evers (2019)	
Lepidochelys olivacea	SMNS 11070	MorphoSource	P462	M22068	Evers & Benson (2018)	
Natator depressus	R112123	none	NA	NA	Jones et al. (2012)	
Chelydridae						
Chelydra serpentina	SMF 32846	none	NA	NA	shared by I. Werneburg	
Macrochelys temminckii	FMNH 22111	MorphoSource	P769	M42869	Evers (2019)	
Dermatemydidae						
Dermatemys mawii	SMF 59463	none	NA	NA	shared by I. Werneburg	
Dermochelyidae						
Dermochelys coriacea	FMNH 171756	MorphoSource	P769	M42864	Evers (2019)	
Dermochelys coriacea	UMZC R3031	MorphoSource	P462	M22024	Evers & Benson (2018)	
Emydidae						
Clemmys guttata	FMNH 22114	MorphoSource	P769	M42870	Evers (2019)	
Deirochelys reticularia	FMNH 98754	MorphoSource	P769	M42883	Evers (2019)	
Emydoidea blandingii	FMNH 22144	MorphoSource	P769	M42871	Evers (2019)	
Emys orbicularis	SMF 1987	none	NA	NA	Lautenschlager, Ferreira & Werneburg (2018)	
Glyptemys insculpta	FMNH 22240	MorphoSource	P769	M42873	Evers (2019)	
Glyptemys muhlenbergii	UF 85274	Digimorph	NA	NA	shared by J. Olori	
Graptemys geographica	NHMUK 55.12.6.11	MorphoSource	P769	M42847	Evers (2019)	
Pseudemys floridana	FMNH 8222	MorphoSource	P769	M42882	Evers (2019)	
Terrapene coahuila	FMNH 47372	MorphoSource	P769	M42881	Evers (2019)	
Terrapene ornata	FMNH 23014	MorphoSource	P769	M42875	Evers (2019)	
Geoemydidae						
Batagur baska	NHMUK 67.9.28.7	MorphoSource	P769	M42851	Evers (2019)	
Cuora amboinensis	NHMUK 69.42.145	MorphoSource	P769	M42852	Evers (2019)	
Cyclemys dentata	NHMUK 97.11.22.3	MorphoSource	P769	M42861	Evers (2019)	
Geoclemys hamiltonii	NHMUK 87.9.30.1	MorphoSource	P769	M42859	Evers (2019)	
Geoemyda spengleri	FMNH 260381	MorphoSource	P769	M42877	Evers (2019)	
Malayemys subtrijuga	NHMUK 1920.1.20.2545	MorphoSource	P769	M42841	Evers (2019)	
Mauremys leprosa	NHMUK unnumbered	MorphoSource	P769	M42862	Evers (2019)	
Morenia ocellata	NHMUK 87.3.11.7	MorphoSource	P769	M42858	Evers (2019)	
Pangshura tecta	NHMUK 1889.2.6.1	MorphoSource	P769	M42839	Evers (2019)	
Rhinoclemmys melanosterna	FMNH 44446	MorphoSource	P769	M42879	Evers (2019)	
Siebenrockiella crassicollis	NHMUK 1864.9.2.47	MorphoSource	P769	M42837	Evers (2019)	
Kinosternidae						
Kinosternon baurii	FMNH 211705	MorphoSource	P769	M42865	Evers (2019)	
Kinosternon scorpioides	SMF 71893	none	NA	NA	shared by I. Werneburg	
Kinosternon subrubrum hippocrepis	FMNH 211711	MorphoSource	P769	M42866	Evers (2019)	
Staurotypus salvinii	NHMUK 1879.1.7.5	MorphoSource	P769	M42838	Evers (2019)	
Sternotherus minor	FMNH 211696	MorphoSource	P462	M22123	Evers & Benson (2018)	
Pelomedusidae						
Pelomedusa subrufa	NMB 16229	none	NA	NA	scanned at University of Fribourg	
Pelusios subniger	NMB 16230	none	NA	NA	scanned at University of Fribourg	
Platysternidae						
Platysternon megacephalum	SMF 69684	none	NA	NA	shared by I. Werneburg	
Podocnemididae						
Podocnemis unifilis	NHMUK 60.4.16.9	MorphoSource	P769	M42848	Evers (2019)	
Podocnemis unifilis	FMNH 45657	MorphoSource	P769	M42880	Evers (2019)	
Testudinidae						
Agrionemys horsfieldii	PCHP 2929	Digimorph	NA	NA	shared by H. Jamniczky	
Aldabrachelys gigantea	NHMUK 77.11.12.2	MorphoSource	P769	M42855	Evers (2019)	
Gopherus agassizii	FMNH 216746	MorphoSource	P769	M42868	Evers (2019)	
Gopherus flavomarginatus	FMNH 98916	MorphoSource	P769	M42884	Evers (2019)	
Gopherus polyphemus	FMNH 211815	MorphoSource	P462	M22041	Evers & Benson (2018)	
Indotestudo elongata	SMF 71585	none	NA	NA	shared by I. Werneburg	
Indotestudo forstenii	SMF 73257	none	NA	NA	shared by I. Werneburg	
Kinixys erosa	SMF 40166	none	NA	NA	shared by I. Werneburg	
Malacochersus tornieri	SMF 58702	none	NA	NA	Lautenschlager, Ferreira & Werneburg (2018)	
Psammobates tentorius verroxii	SMF 57142	none	NA	NA	shared by I. Werneburg	
Testudo marginata	FMNH 51672	MorphoSource	P769	M42885	Evers (2019)	
Trionychidae						
Amyda cartilaginea	FMNH 244117	MorphoSource	P769	M42876	Evers (2019)	
Apalone mutica	PCHP 2746	Digimorph	NA	NA	shared by H. Jamniczky	
Apalone spinifera emoryi	FMNH 22178	MorphoSource	P462	M22038	Evers & Benson (2018)	
Chitra indica	NHMUK 1926.12.16.1	MorphoSource	P769	M42842	Evers (2019)	
Cyclanorbis senegalensis	NHMUK 65.5.9.21	MorphoSource	P769	M42850	Evers (2019)	
Cycloderma frenatum	NHMUK 84.2.4.1	MorphoSource	P769	M42857	Evers (2019)	
Lissemys punctata	SMF 74141	none	NA	NA	shared by I. Werneburg	
Pelodiscus sinensis	IW576-2b	none	NA	NA	Lautenschlager, Ferreira & Werneburg (2018)	
Note:

Abbreviations: FMNH, Field Museum of Natural History, Chicago, IL, USA; NHMUK, Natural History Museum London, London, England; NMB, Naturhistorisches Museum Basel, Basel, Switzerland; PCHP, Chelonian Research Institute, Oviedo, FL, USA; PIMUZ, Paläontologisches Museum Zürich, Zürich, Switzerland; SMF, Senckenberg Naturmuseum Frankfurt, Frankfurt am Main, Germany; SMNS, Staatliches Museum für Naturkunde Stuttgart, Stuttgart, Germany; UF, Florida Museum of Natural History, Gainesville, FL, USA; UMZC, University Museum of Zoology Cambridge, Cambridge, England.

Figure 2 Simplified phylogenetic tree showing the relationships between crown clades turtles, following Crawford et al. (2015).

Assessment of size relationships between “medial” and stapedial blood flow

Albrecht (1976) hypothesized that reductions in the size of the stapedial artery of turtles are counterbalanced by increases in size of the palatine or orbital arteries (“medial” blood flow) to ensure the arterial blood supply for the facial region of the head. In order to quantitatively and statistically assess this hypothesis, we measured the cross-sectional areas of the stapedial and carotid canals in Dragonfly 4.0 (https://www.theobjects.com/dragonfly/index.html) (see Table 2). For the internal carotid artery canal, the cross sectional area was measured close to the foramen posterius canalis carotici interni. For the stapedial, palatine, and cerebral artery canals, the cross sectional area was measured close to their respective exit foramina. In all cases, measurements were taken orthogonal to the orientation of the canal. These measurements provide approximate estimates of blood flow, as arterial canal diameter is proportional to arterial size in turtles (Albrecht, 1976; Jamniczky & Russell, 2004). To test if the reduction of the stapedial artery has an effect on the “medial” blood flow, we performed phylogenetic generalized least squares regression (pGLS; Grafen, 1989; Rohlf, 2001) of the log10-transformed internal carotid arterial cross-sectional area on the respective measurement for the stapedial artery. pGLS is a modification of ordinary least squares regression (OLS), which takes into account the expected co-variance structure of residuals that is the result of phylogenetic non-independence of the input data (Felsenstein, 1985; Grafen, 1989; Symonds & Blomberg, 2014). The expected variances and co-variances of traits among species follow a Brownian motion model of trait evolution in pGLS and related methods like phylogenetic independent contrasts (Felsenstein, 1985; Pagel, 1997, 1999), although different models of trait evolution can be used in pGLS. The strength of the phylogenetic correlation was estimated under maximum likelihood during the model fitting using variable Pagel’s lambda (Revell, 2010; Motani & Schmitz, 2011). Pagel’s lambda is a parameter that describes the model of evolution, with values of 1 indicating that trait evolution occurred under Brownian motion, whereas a value of 0 means there is no phylogenetic signal in the residuals of the regression. pGLS was performed in R with the gls() function of the package ape (Paradis & Schliep, 2018), and the corPagel option for setting the correlation structure of the model. Because of phylogenetic correlation in the error structure of a pGLS, the coefficient of correlation R2 cannot be easily defined for phylogenetic regressions (Ives, 2018). Here, we use a generalized form of R2 described by Nagelkerke (1991), which is derived by comparison of the log-likelihood of the model with that of a null model (log10(CCI) ~ 1). Because results from our initial pGLS analysis suggest that chelonioids, trionychians, and kinosternoids show differences in regression slope to other turtles (see “Results”), we performed generalized least-squares phylogenetic analysis of covariance (pANCOVA) using the gls.ancova function of the package evomap (Smaers & Rohlf, 2016; Smaers & Mongle, 2018) to test if our data can be better explained by a model with different intercepts or slopes or both for the visually separated groups. We defined a group with consistently positive residuals (i.e., chelonioids, kinosternoids, trionychians) and another with the remaining taxa for which models could be compared. As the phylogenetic input for the pGLS analyses, we used the dated phylogeny of Pereira et al. (2017), which we pruned to match our taxon sample. Two species in our sample, Kinixys erosa and Pseudemys floridana, were not included in the study of Pereira et al. (2017). We re-assigned the tip-labels of two closely related species, Kinixys belliana and Pseudemys texana, in the tree of Pereira et al. (2017) to those of our sampled species. This procedure assumes that Kinixys erosa (or Pseudemys floridana) is closer related to Kinixys belliana (or Pseudemys texana) than to any other turtle in the tree, which is justified by the circumstance that the monophyly of neither the species of Kinixys nor Pseudemys are disputed, and by the fact that we sampled only one species of each genus in our study. For the regression analysis, we chose the internal carotid artery as representing “medial” blood flow, as opposed to the combined values of the cerebral and palatine arteries. This allowed us to include taxa for which the palatine artery has no osteological correlate but is present (i.e., chelonioids; see “Results”). Additionally, this procedure allowed the inclusion of podocnemidids. Although podocnemidids provide no reliable estimation for the cross-sectional area of the internal carotid artery, as the artery extends through the enlarged cavum pterygoidei (see “Results”), these turtles could be included by taking the cross-sectional areas of the cerebral artery instead. This can be justified, as the palatine artery is absent in podocnemidids (see “Results”), and thus the cerebral artery canal represents the internal carotid artery in terms of cross-sectional area (=blood flow). Although interesting for questions we would have liked to address, we had to exclude Dermatemys mawii and Kinosternon scorpioides from the analysis (n = 2) because of their completely reduced stapedial arteries, as the logarithm of zero is undefined. Additionally, we excluded Kinosternon baurii after an initial run of analyses including it. K. baurii has a strongly reduced stapedial artery size in absolute numbers (two orders of magnitude smaller than in other kinosternids), and has a proportionally extremely large internal carotid artery (same order of magnitude as in other kinosternids), leading to an extreme positive residual when included into a regression. It can, therefore, be statistically defined as an outlier with large leverage on the regression line. Although the data for K. baurii strongly supports the notion that internal carotid artery size is strongly increased when the stapedial artery size is strongly reduced (see “Discussion”), we conservatively excluded the taxon. Our sample in total was thus n = 63.

Table 2 Cross-sectional area values (µm2) of the stapedial and carotid artery canals.

Taxon	Specimen number	Clade	CST (µm2)	CCI (µm2)	CCL (µm2)	CCB (µm2)	
Chelodina oblonga	NHMUK 64.12.22	Chelidae	1,059,000	151,000	0	226,000	
Chelus fimbriatus	NHMUK 81.9.27.4	Chelidae	4,085,000	892,000	0	736,000	
Emydura subglobosa	PIMUZ lab 2009.37	Chelidae	80,000	35,000	0	48,000	
Hydromedusa tectifera	SMF 70500	Chelidae	810,000	220,000	0	129,000	
Phrynops geoffroanus	SMF 45470	Chelidae	3,287,000	560,000	0	360,000	
Phrynops hilarii	NHMUK 91.3.16.1	Chelidae	6,240,000	591,000	0	646,000	
Podocnemis unifilis	NHMUK 60.4.16.9	Podocnemididae	5,716,000	0	0	1,171,000	
Podocnemis unifilis	FMNH 45657	Podocnemididae	622,000	0	0	366,000	
Pelomedusa subrufa	NMB 16229	Pelomedusidae	800,000	220,000	0	214,000	
Pelusios subniger	NMB 16230	Pelomedusidae	1,317,000	579,000	0	540,000	
Carettochelys insculpta	NHMUK 1903.7.10.1	Carettochelyidae	6,038,000	5,492,000	0	5,946,000	
Carettochelys insculpta	SMF 56626	Carettochelyidae	6,074,000	5,925,000	0	5,759,000	
Amyda cartilaginea	FMNH 244117	Trionychidae	4,051,000	6,345,000	2,473,000	8,543,000	
Apalone mutica	PCHP 2746	Trionychidae	4,294,000	15,625,000	5,532,000	15,084,000	
Apalone spinifera emoryi	FMNH 22178	Trionychidae	1,084,000	2,942,000	1,086,000	3,593,000	
Chitra indica	NHMUK 1926.12.16.1	Trionychidae	1,434,000	2,082,000	1,321,000	3,190,000	
Cyclanorbis senegalensis	NHMUK 65.5.9.21	Trionychidae	1,462,000	3,750,000	2,134,000	2,899,000	
Cycloderma frenatum	NHMUK 84.2.4.1	Trionychidae	5,057,000	8,859,000	3,025,000	8,682,000	
Lissemys punctata	SMF 74141	Trionychidae	599,000	1,228,000	662,000	1,106,000	
Pelodiscus sinensis	IW576-2b	Trionychidae	349,000	911,000	538,000	1,070,000	
Kinosternon baurii	FMNH 211705	Kinosternoidea	7,000	296,000	284,000	204,000	
Kinosternon scorpioides	SMF 71893	Kinosternoidea	0	389,000	325,000	253,000	
Kinosternon subrubrum hippocrepis	FMNH 211711	Kinosternoidea	154,000	354,000	426,000	193,000	
Staurotypus salvinii	NHMUK 1879.1.7.5	Kinosternoidea	290,000	1,038,000	1,110,000	431,000	
Sternotherus minor	FMNH 211696	Kinosternoidea	120,000	428,000	617,000	254,000	
Dermatemys mawii	SMF 59463	Kinosternoidea	0	1,957,000	2,431,000	619,000	
Chelydra serpentina	SMF 32846	Chelydridae	4,474,000	1,586,000	299,000	893,000	
Macrochelys temminckii	FMNH 22111	Chelydridae	5,302,000	716,000	331,000	527,000	
Caretta caretta	NHMUK 1938.1.9.1	Cheloniidae	26,309,000	11,101,000	NA	5,080,000	
Caretta caretta	NHMUK 1940.3.15.1	Cheloniidae	20,114,000	11,618,000	NA	5,565,000	
Chelonia mydas	NHMUK 1969.776	Cheloniidae	22,229,000	11,715,000	NA	3,359,000	
Eretmochelys imbricata	FMNH 22242	Cheloniidae	5,324,000	2,387,000	NA	1,457,000	
Lepidochelys olivacea	SMNS 11070	Cheloniidae	8,862,000	3,737,000	NA	1,944,000	
Natator depressus	R112123	Cheloniidae	20,562,000	10,457,000	NA	3,624,000	
Dermochelys coriacea	FMNH 171756	Dermochelyidae	114,072,000	32,496,000	NA	NA	
Dermochelys coriacea	UMZC R3031	Dermochelyidae	94,064,000	32,013,000	NA	NA	
Platysternon megacephalum	SMF 69684	Emysternia	1,327,000	404,000	0	424,000	
Clemmys guttata	FMNH 22114	Emysternia	316,000	77,000	21,000	69,000	
Deirochelys reticularia	FMNH 98754	Emysternia	573,000	296,000	25,000	240,000	
Emydoidea blandingii	FMNH 22144	Emysternia	845,000	228,000	109,000	167,000	
Emys orbicularis	SMF 1987	Emysternia	547,000	181,000	51,000	176,000	
Glyptemys insculpta	FMNH 22240	Emysternia	1,145,000	358,000	109,000	197,000	
Glyptemys muhlenbergii	UF 85274	Emysternia	1,102,000	501,000	100,000	590,000	
Graptemys geographica	NHMUK 55.12.6.11	Emysternia	404,000	142,000	60,000	74,000	
Pseudemys floridana	FMNH 8222	Emysternia	898,000	216,000	35,000	171,000	
Terrapene coahuila	FMNH 47372	Emysternia	474,000	181,000	48,000	165,000	
Terrapene ornata	FMNH 23014	Emysternia	437,000	88,000	14,000	74,000	
Agrionemys horsfieldii	PCHP 2929	Testudinidae	3,165,000	1,403,000	0	620,000	
Aldabrachelys gigantea	NHMUK 77.11.12.2	Testudinidae	19,120,000	4,601,000	0	1,804,000	
Gopherus agassizii	FMNH 216746	Testudinidae	4,030,000	553,000	0	313,000	
Gopherus flavomarginatus	FMNH 98916	Testudinidae	3,879,000	776,000	0	628,000	
Gopherus polyphemus	FMNH 211815	Testudinidae	1,460,000	604,000	0	293,000	
Indotestudo elongata	SMF 71585	Testudinidae	1,611,000	874,000	0	605,000	
Indotestudo forstenii	SMF 73257	Testudinidae	626,000	476,000	0	525,000	
Kinixys erosa	SMF 40166	Testudinidae	1,027,000	451,000	0	449,000	
Malacochersus tornieri	SMF 58702	Testudinidae	965,000	447,000	0	224,000	
Psammobates tentorius verroxii	SMF 57142	Testudinidae	321,000	309,000	0	279,000	
Testudo marginata	FMNH 51672	Testudinidae	1,754,000	504,000	0	290,000	
Batagur baska	NHMUK 67.9.28.7	Geoemydidae	5,452,000	1,966,000	755,000	1,059,000	
Cuora amboinensis	NHMUK 69.42.145	Geoemydidae	733,000	188,000	93,000	132,000	
Cyclemys dentata	NHMUK 97.11.22.3	Geoemydidae	706,000	205,000	12,000	251,000	
Geoclemys hamiltonii	NHMUK 87.9.30.1	Geoemydidae	1,285,000	201,000	101,000	232,000	
Geoemyda spengleri	FMNH 260381	Geoemydidae	197,000	47,000	3,000	46,000	
Malayemys subtrijuga	NHMUK 1920.1.20.2545	Geoemydidae	784,000	404,000	0	413,000	
Mauremys leprosa	NHMUK unnumbered	Geoemydidae	767,000	291,000	67,000	180,000	
Morenia ocellata	NHMUK 87.3.11.7	Geoemydidae	1,452,000	335,000	109,000	202,000	
Pangshura tecta	NHMUK 1889.2.6.1	Geoemydidae	2,049,000	409,000	0	354,000	
Rhinoclemmys melanosterna	FMNH 44446	Geoemydidae	440,000	303,000	0	180,000	
Siebenrockiella crassicollis	NHMUK 1864.9.2.47	Geoemydidae	361,000	206,000	80,000	128,000	
Note:

Abbreviations: CCB, canalis caroticus basisphenoidalis; CCI, canalis caroticus internus; CCL, canalis caroticus lateralis; CST, canalis stapedio-temporalis. Note that value 0 is used when the canalis caroticus lateralis and arteria palatina are known to be absent, whereas NA (non-applicable) is only used for chelonioids, in which the palatine artery is present but does not extend through its own canal, preventing us from making any measurement of the latter.

Nomenclature and homology

As our primary sources of information are skulls, we were not able to observe the cranial circulation and innervation systems directly, but rather could only assess their former presence by explicit reference to the canals they left in bone in combination with published dissections, in particular Albrecht (1967, 1976). The vast majority of our assessments are, nevertheless, uncontroversial. We therefore shorten our descriptions by not justifying explicitly why we believe a particular canal to be filled by a particular soft structure. Instead, we only provide separate justifications in the text for the rare cases where our conclusions differ from previously published ones.

The dissections of Albrecht (1967, 1976) documented strong variation in the organization of the cranial blood irrigation of turtles, which are interchangeably supplied by branches of the internal carotid artery and the stapedial artery. For instance, the sensory organs and tissues of the snout region (i.e., “the palate”) are mostly supplied by the stapedial branch in chelydrids, testudinoids, and pleurodires, by the palatine branch in kinosternoids, by a mixture of the stapedial and palatine branches in chelonioids, but by the orbital branches of the cerebral branch in trionychids. Conversely, the mandible is supplied by the stapedial branch in chelydrids, pleurodires, and testudinoids, but by the palatine branch in kinosternoids, chelonioids, and trionychids. So, while the terms “cerebral” and “stapedial” are somewhat misleading, as they disguise the fact that tissues other than the brain and stapes are often supplied by them, the term “palatine” is positively misleading, as the relevant artery, if present, does not supply the palate at all in many turtles. The situation is especially problematic in trionychids, where the “palatine artery” only supplies the mandible, while the palate is entirely supplied by a large subordinate branch of the “cerebral artery.”

The nomenclatural system of Rabi et al. (2013) created nomenclatural clarity by providing separate names for all possible canals and openings formed by the internal carotid system, but accidentally introduced a new source of confusion by selecting names for the canals that negate the homology of the vessels that they contain. As the system is still relatively new, we here take the opportunity to provide minor adjustments, by replacing the terms canalis caroticus cerebralis with canalis caroticus basisphenoidalis and canalis caroticus palatinum with canalis caroticus lateralis. We select these terms as they have historic precedence. The term canalis caroticus basisphenoidalis connects to the term “foramen caroticum basisphenoidale,” which was introduced by Gaffney (1983) for the posterior opening of the carotid canal that punctures the basisphenoid (e.g., the internal carotid canal of Gaffney, 1983, but the cerebral canal of Rabi et al., 2013). Although this term was sometimes applied to the fenestra caroticus of Rabi et al. (2013) (Meylan & Gaffney, 1989; Brinkman & Peng, 1996; Sukhanov, 2000), it was mostly used for the aforementioned posterior opening in subsequent descriptions of basal turtles (Gaffney, 1990, 1996; Gaffney & Meylan, 1992; Gaffney & Ye, 1992; Brinkman & Peng, 1993; Sukhanov, 2000). The term canalis caroticus lateralis, on the other hand, had been used consistently over the course of the entire 20th century (Gaffney, 1972, 1979), but had been replaced purposefully by Rabi et al. (2013), as the name of the canal did not make reference to the arteries it contained. However, as highlighted above, the replacement term “palatine canal” is just as confusing, as the vessels held in this canal do not supply the palate in many turtles. In addition to reconnecting to the historic literature, these newly established names have the advantage of being purely positional and not implying what vessels they contain. This nomenclature therefore only highlights the homology that exists to the basic branching pattern seen in the carotid system of turtles. As the artery held in the canalis caroticus basisphenoidalis universally supplies the brain, we here retain the term “cerebral artery,” even though other structures may be supplied as well by subordinate branches that diverge off the cerebral artery within the sella turcica. For the artery held in the canalis caroticus lateralis, we will try to be specific, where possible, by reference to Albrecht (1967, 1976). However, in anticipation of the needs of paleontologists, who cannot know what tissues are supplied by the arteries passing through the lateral canal, we suggest the informal term “lateral branch of the internal carotid canal.”

In addition to the modifications provided for the nomenclature of the cranial arterial circulation system, we also add the terms “proximal” and “distal” to some aspects of the innervation system. Although the usage of “proximal” and “distal” are unusual in a cranial context, we find these terms useful to refer to distinct sections of nerves within the cranium. The “proximal” part of a nerve is close to the brain or its ganglion, whereas “distal” portions are close to the innervated tissue. Although the sensory nerves discussed herein pass information from the sensory organs or innervated tissues to the brain, we follow Gaffney (1972, 1979) by describing sensory nerve canals as moving away from the brain, to remain consistent with the literature.

Our modified list for the definitions of all canals we discuss and criteria on how to recognize them is as follows:

Cavum cranii—We here restrict the meaning of the term cavum cranii from referring to the entire space from the fossa nasalis to the foramen magnum (Gaffney, 1972) to the endocranial space limited anterodorsally by the sulcus olfactorius and posteriorly by the foramen magnum.

Canalis caroticus internus—The bony canal that contains the main branch of the internal carotid artery from the foramen posterius canalis carotici interni to the fenestra caroticus (only developed in some fossil turtles) or its split into the canalis caroticus lateralis and canalis caroticus basisphenoidalis (Rabi et al., 2013). Whenever the split of the internal carotid artery cannot be identified, typically because of the absence of a formed canalis caroticus lateralis, the canalis caroticus internus is defined as the portion of the canal that does not yet traverse the basisphenoid, as opposed to the canalis caroticus basisphenoidalis, which is defined as the portion of the canal that is located inside the basisphenoid. Additional structures might be contained in the canalis caroticus internus, in particular the vidian nerve, but this is not reflected in the name. The canalis caroticus internus typically traverses the skull at the junction of the basisphenoid with the pterygoid and/or prootic in the posterior prolongation of the canalis caroticus basisphenoidalis.

Foramen posterius canalis carotici interni—The posterior foramen to the canalis caroticus internus, which serves as the entry of the internal carotid artery into the skull (Rabi et al., 2013).

Canalis caroticus basisphenoidalis—The canal that contains the cerebral artery from the fenestra caroticus (only developed in some fossil turtles) or from the split of the canalis caroticus internus into its primary branches to the foramen anterius canalis carotici basisphenoidalis (canalis caroticus cerebralis of Rabi et al., 2013). Whenever a canalis caroticus lateralis cannot be identified, the canalis caroticus basisphenoidalis is defined as the portion of the carotid canal that traverses the basisphenoid, as opposed to the more posteriorly located canalis caroticus internus, which is located outside of the basisphenoid. No other structures are known to traverse this canal. The canalis caroticus basisphenoidalis typically penetrates the basisphenoid at mid-length and universally exits the basisphenoid within the sella turcica. The canalis caroticus basisphenoidalis is the canal with the greatest diameter to pierce the basisphenoid.

Foramen anterius canalis carotici basisphenoidalis—The anterior opening of the canalis caroticus basisphenoidalis (foramen anterius canalis carotici cerebralis of Rabi et al., 2013), universally located within the sella turcica of the basisphenoid.

Canalis caroticus lateralis—The bony canal that contains the lateral branch of the internal carotid canal, typically the palatine artery, from the fenestra caroticus (only developed in some fossil turtles) or from the the split of the canalis caroticus internus into its two primary branches to the foramen anterius canalis carotici lateralis (canalis caroticus palatinum of Rabi et al., 2013). The vidian nerve may traverse part of this canal. The canalis caroticus lateralis typically punctures the skull at the junction of the basisphenoid with the pterygoid and exits within the sulcus cavernosus and near the anterior end of the latter.

Foramen anterius canalis carotici lateralis—The anterior exit of the canalis caroticus lateralis, which typically contains the palatine artery, located near the anterior end of the sulcus cavernosus (foramen anterius canalis carotici palatinum of Rabi et al., 2013).

Canalis carotico-pharyngealis—A single or series of ventrally-directed bony canals in the pterygoid that contain the arteria carotico-pharyngealis. The arteria carotico-pharyngealis splits from the palatine artery, enters the canalis carotico-pharyngealis to extend ventrally through the pterygoid, and exits the skull through the foramen carotico-pharyngeale (Albrecht, 1967). We only infer the presence of a canalis carotico-pharyngealis whenever a canal connects the lateral canal with the ventral surface of the pterygoid.

Foramen carotico-pharyngeale—The ventral exit of the canalis carotico-pharyngealis located on the ventral surface of the pterygoid.

Canalis cavernosus—The bony canal that contains the lateral head vein and extends between the foramen cavernosum anteriorly and the cavum acustico-jugulare posteriorly (Gaffney, 1972). The canalis cavernosus is usually formed by the quadrate, prootic, and pterygoid. The distal portion of the hyomandibular nerve may traverse this canal as well. Additionally, the mandibular artery may traverse this canal in some turtle clades. The canalis cavernosus is generally a straight canal with a large diameter that obliquely traverses the skull lateral to the braincase.

Canalis nervus facialis—The bony canal that contains the facial nerve medial to its split at the geniculate ganglion into the vidian and hyomandibular branches (Rollot, Lyson & Joyce, 2018). The canal is recognized as the structure laterally penetrating the prootic from the fossa acustico-facialis. It may end within the canalis cavernosus, the canalis caroticus internus, or at its split into the vidian and hyomandibular canals.

Canalis nervus hyomandibularis proximalis—The bony canal that is laterally continuous with the canalis nervus facialis, and joins the canalis cavernosus. This canal contains the proximal part of the hyomandibular branch of the facial nerve, but is only present in turtles in which the geniculate ganglion is not positioned within the canalis cavernosus.

Canalis nervus hyomandibularis distalis—The bony canal that originates within the wall of the canalis cavernosus posterior to the connection of the canalis nervus facialis or canalis nervus hyomandibularis proximalis. The canal extends posteriorly, paralleling the canalis cavernosus, and joins the cavum acustico-jugulare. It contains the distal portion of the hyomandibular branch of the facial nerve, and is only present in some turtles.

Canalis pro ramo nervi vidiani—The bony canal that contains the posterior (=proximal) portion of the vidian nerve to the exclusion of the other primary nerves and blood vessels defined herein (Rollot, Lyson & Joyce, 2018). In living turtles, the canal, when present, connects the geniculate ganglion to the canalis caroticus internus and traverses the prootic and pterygoid along its way.

Canalis nervus vidianus—The bony canal that contains the anterior (=distal) portion of the vidian nerve to the exclusion of the other primary nerves and blood vessels defined herein (Rollot, Lyson & Joyce, 2018). The canal typically originates from the canalis caroticus lateralis and anteroposteriorly traverses the pterygoid or palatine to open towards the foramen palatinum posterius. Numerous accessory canals may branch off this canal that innervate the dorsal or ventral side of the pterygoid or palatine.

Foramen anterius canalis nervi vidiani—The anterior exit of the canalis nervus vidianus (Rollot, Lyson & Joyce, 2018). The foramen is typically located posterior to the foramen palatinum posterius, but can be split into multiple foramina arranged on the dorsal or ventral surfaces of the pterygoid or palatine.

Foramen arteriomandibulare—A separate, anterior exit for the mandibular artery from the canalis cavernosus into the temporal fossa. It is located just posterior to the foramen nervi trigemini.

Results

Chelidae (Figs. 3)

Canalis caroticus internus—In all chelids studied, the internal carotid artery enters the skull ventrally. The foramen posterius canalis carotici interni (Figs. 3A and 3B) is formed by the prootic in Phrynops hilarii and Hydromedusa tectifera, by the basisphenoid and the prootic in Chelus fimbriatus, by the prootic and quadrate in Chelodina oblonga and Phrynops geoffroanus, and by the basisphenoid, prootic and quadrate in Emydura subglobosa. Historically, only the proootic and basisphenoid were recognized as contributing to the foramen in chelids (Siebenrock, 1897; Albrecht, 1976). However, recent phylogenetic studies have documented further variation, such as the contribution of the quadrate (Evers & Benson, 2019). Albrecht (1976) similarly described the foramen posterius canalis carotici interni as only being formed by the prootic and basisphenoid in the chelids he studied. The apparent variation in bone contributions to this foramen was recently coded by Evers & Benson (2019) as separate characters. The canalis caroticus internus is directed anteromedially and formed by the prootic and basisphenoid in Chelus fimbriatus (Figs. 3A and 3B), P. hilarii, and H. tectifera, by the quadrate, prootic, and basisphenoid in Chelodina oblonga and P. geoffroanus, and by the quadrate, prootic, basisphenoid and pterygoid in E. subglobosa. Again, this contrasts with Albrecht (1976), who noted that the canalis caroticus internus is only formed by the prootic and basisphenoid. We are able to observe a canal extending anteriorly along the pterygoid-basisphenoid suture in Chelodina oblonga and H. tectifera, which fits with the expected position of a canalis caroticus lateralis. A canal in a more lateral position is found in Chelus fimbriatus (Figs. 3A and 3B), E. subglobosa, P. geoffroanus, and P. hilarii. Albrecht (1976) stated that the canalis caroticus lateralis is present in all the chelids he observed, except for Chelus fimbriatus, and Hermanson et al. (2020) also identified a canalis caroticus lateralis in chelids. A close reading of Albrecht (1976) reveals that he did not observe the palatine or any other artery in any chelid in his sample and that the vidian nerve is the sole structure to occupy the canalis caroticus lateralis instead. These observations are indirectly confirmed by our sample, as the canal in question does not split from the canalis caroticus internus in Chelodina oblonga (right side) and H. tectifera, but rather directly from the inferred position of the geniculate ganglion itself, following on from the position of the latter and extending anteriorly mostly along the pterygoid-basisphenoid suture (see Fig. 3A for illustration as Chelus fimbriatus exhibits a similar configuration). The dissections of Albrecht (1976), clearly showing the presence of the vidian nerve and absence of the palatine artery, and our observations for Chelodina oblonga and H. tectifera, support the fact that the canal of interest very likely corresponds to the canalis nervus vidianus and should not be termed canalis caroticus lateralis. The case of chelids is a good example to highlight how identification of canals can be tricky and misleading when osteological correlates are the only source of information available. As Albrecht (1976) provided dissection-based descriptions, we were able to decide on the identity of the canals with confidence, without which we would have identified the canal extending along the pterygoid-basisphenoid suture as the canalis caroticus lateralis, as the latter is expected to be located at this position in most turtles. The case highlighted herein shows how much carefulness is required when working on the cranial circulation, especially in fossils, for which soft tissues are only rarely preserved (see “Discussion”). The canalis carotico-pharyngealis is absent. The canalis caroticus basisphenoidalis, herein defined by its entry into the basisphenoid due to the absence of the canalis caroticus lateralis, and the foramen anterius canalis carotici basisphenoidalis are formed by the basisphenoid (Figs. 3A–3C). The foramina anterius canalis carotici basisphenoidalis are widely-spaced (Figs. 3A and 3B). The canalis stapedio-temporalis is the largest canal in all observed chelids, ranging from ten times bigger than any carotid canal in P. hilarii, to two times the size of them in E. subglobosa. The canalis caroticus internus is the smallest canal in Chelodina oblonga, E. subglobosa, and P. hilarii, whereas the canalis caroticus basisphenoidalis is the smallest in Chelus fimbriatus, H. tectifera, and P. geoffroanus (Table 2).

Figure 3 The carotid circulation and vidian canal system of Chelus fimbriatus (NHMUK 81.9.27.4).

Three-dimensional reconstructions of the basisphenoid, right pterygoid, in (A) dorsal, (B) ventral, and (C) left lateral view. Illustration in dorsal view (D) highlighting the placement of relevant arteries, nerves, and veins. Dark colors highlight sections fully covered by bone, light colors partially or fully uncovered sections. Abbreviations: ac, arteria carotis cerebralis; aci, arteria carotis interna; bs, basisphenoid; “cc”, secondary canalis cavernosus; ccb, canalis caroticus basisphenoidalis; cci, canalis caroticus internus; ccv, canalis cavernosus; cnf, canalis nervus facialis; cnhp, canalis nervus hyomandibularis proximalis; cnv, canalis nervus vidianus; faccb, foramen anterius canalis carotici basisphenoidalis; facnv, foramen anterius canalis nervi vidiani; fpcci, foramen posterius canalis carotici interni; gg, geniculate ganglion; pt, pterygoid; vcl, vena capitis lateralis; VII, nervus facialis; VIIhy, nervus hyomandibularis; VIIvi, nervus vidiani.

Mandibular artery—There is no trace in bone of the mandibular artery in our sample. This is consistent with Albrecht (1976) conclusion that the (internal) mandibular artery of chelids branches from the stapedial artery after its exit from the foramen stapedio-temporale.

Canalis cavernosus—The canalis cavernosus (Figs. 3A–3D) extends from the foramen cavernosum to the level of the foramen stapedio-temporale in Chelus fimbriatus, P. hilarii and P. geoffroanus, but posterior to the level of the foramen stapedio-temporale in Chelodina oblonga, E. subglobosa, and H. tectifera. The canalis cavernosus is formed by the quadrate, prootic, and basisphenoid in Chelodina oblonga, by the prootic, pterygoid, basisphenoid, and parietal in Chelus fimbriatus, by the quadrate and prootic in E. subglobosa, and by the quadrate, prootic, and pterygoid in P. hilarii, P. geoffroanus, and H. tectifera. The foramen cavernosum itself is formed by the prootic and pterygoid. In Chelus fimbriatus, the lateral head vein is fully enclosed by bone a second time anterior to the foramen nervi trigemini, a feature not observed in other chelids (Figs. 3A–3D). In all observed chelids but E. subglobosa, the canalis cavernosus connects to a large ventrolaterally directed canal that opens on the ventral surface of the pterygoid, and which we call the “chelid canal” herein (Figs. 3A and 3B). Siebenrock (1897) observed this canal in all chelids and interpreted it as containing a branch of the carotid artery, but dissections made by Albrecht (1976) on Chelodina sp. and Mesoclemmys nasuta showed that the canal only contains connective tissue.

Facial nerve canal system—The facial nerve extends laterally from the cavum cranii to the canalis caroticus internus through the canalis nervus facialis, which is formed by the prootic (Figs. 3A and 3D). The geniculate ganglion is inferred to be in contact with the canalis caroticus internus in all observed chelids (Figs. 3A and 3D). The hyomandibular branch extends posterolaterally through the prootic in its own canal, the canalis nervus hyomandibularis proximalis, for most of its length (Figs. 3A–3D). In Chelodina oblonga and E. subglobosa, the quadrate contributes to the canalis nervus hyomandibularis proximalis posteriorly. In all observed taxa but E. subglobosa, the canalis nervus hyomandibularis proximalis connects to the canalis cavernosus posteriorly, close to the position of the columella auris. In E. subglobosa, however, the canalis nervus hyomandibularis proximalis does not connect to the canalis cavernosus and directly joins the cavum acustico-jugulare. As already discussed above, Albrecht (1976) suggested that the palatine artery is absent and that the vidian nerve extends through the canalis caroticus lateralis, but we here conclude based on the same observations that the lateral carotid canal is absent and that the vidian nerve traverses the canalis nervus vidianus. In Chelus fimbriatus, the vidian nerve emerges from the geniculate ganglion and does not enter the canalis caroticus internus, but rather directly enters the canalis nervus vidianus which crosses the prootic and pterygoid to join the sulcus cavernosus at the level of the foramen nervi trigemini (Figs. 3A and 3D). The path is generally similar in other chelids, but in P. hilarii, P. geoffroanus, and H. tectifera, the vidian nerve partially enters the canalis caroticus internus, whereas in Chelodina oblonga and E. subglobosa the vidian nerve fully enters the canalis caroticus internus before splitting from it again further anteriorly. The vidian nerve passes for this part of the vidian canal through the prootic, pterygoid, and basisphenoid in Chelodina oblonga and H. tectifera, through the prootic and pterygoid in Chelus fimbriatus, E. subglobosa, P. hilarii, and P. geoffroanus. The basisphenoid contribution to this canal had previously been noted for Chelus fimbriatus and Chelodina longicollis by Siebenrock (1897). In E. subglobosa, P. hilarii, P. geoffroanus, and H. tectifera, no trace of the vidian nerve is visible inside the bone anterior to the foramen nervi trigemini, so the vidian nerve probably lies inside the sulcus cavernosus. In Chelus fimbriatus, the vidian nerve re-enters the skull anteriorly through the lateral wall of the sulcus cavernosus and extends anteriorly along the pterygoid-parietal suture for a long distance and at its anterior end, splits into two short canals that leave the skull posterolaterally to the descending process of the parietal (Figs. 3A, 3B and 3D). The foramen anterius canalis nervi vidiani is formed by the pterygoid and parietal. A similar pattern is visible in Chelodina oblonga, except that the portion of the vidian nerve that is enclosed by bone is much shorter.

Podocnemididae (Fig. 4)

Canalis caroticus internus—The main branch of the internal carotid artery is never tightly enclosed by bone in the observed podocnemidids, so a true foramen posterius canalis carotici interni is absent, at least if a tight enclosure is used as a definitional criterion (Fig. 4A). Instead, podocnemidids possess an enlarged cavity, the podocnemidoid fossa (Lapparent de Broin & Werner, 1998) or cavum pterygoidei (Gaffney, Tong & Meylan, 2006), through which the carotid artery extends and that is additionally filled by muscle tissue (Albrecht, 1976). As the internal carotid artery is not enclosed by bone, it is difficult to make clear statements regarding the circulation pattern of the carotid artery in our specimens. We can only identify with confidence a short canalis caroticus basisphenoidalis that is formed by the basisphenoid (Fig. 4A). These observations broadly agree with those of Siebenrock (1897), though with different nomenclature, as he interprets the entry of the cavum pterygoidei as the foramen posterius canalis carotici interni. Albrecht (1976) described a small artery splitting from the internal carotid artery in Podocnemis sextuberculata that he hypothesized to be a possible, vestigial palatine artery, but he was not able to identify similar structures in the other podocnemidids he dissected. This small artery splits into two branches, one supplying the muscle inside the cavum pterygoidei, one joining the canalis cavernosus to supply tissue within the sulcus cavernosus. Albrecht (1976) also noted the presence of the foramen anterius canalis carotici lateralis in all of the podocnemidids he observed. We identify a large, irregularly shaped fenestra between the cavum pterygoidei and the sulcus cavernosus in both specimens of Podocnemis unifilis that may correspond to the foramen anterius canalis carotici lateralis of Albrecht (1976). Given that Albrecht (1976) only observed a blood vessel that passes this fenestra in one out of four podocnemidids in his sample and that this vessel does not supply tissues related to the palate, we conclude that the palatine artery is not present in extant podocnemidids. The canalis carotico-pharyngealis are absent. The foramina anterius canalis carotici basisphenoidalis are widely-spaced (Figs. 4A and 4B). The canalis stapedio-temporalis is the largest canal, being five times larger than the canalis caroticus basisphenoidalis in one of our specimens of Podocnemis unifilis (NHMUK 60.4.16.9), but only twice as large in the other (FMNH 45657) (Table 2).

Figure 4 The carotid circulation and vidian canal system of Podocnemis unifilis (NHMUK 60.4.16.9).

Three-dimensional reconstructions of the basisphenoid, right pterygoid, in (A) dorsal, (B) ventral, and (C) left lateral view. Illustration in dorsal view (D) highlighting the placement of relevant arteries, nerves, and veins. Dark colors highlight sections fully covered by bone, light colors partially or fully uncovered sections. Abbreviations: ac, arteria carotis cerebralis; aci, arteria carotis interna; bs, basisphenoid; ccb, canalis caroticus basisphenoidalis; ccv, canalis cavernosus; cnf, canalis nervus facialis; cnhd, canalis nervus hyomandibularis distalis; cnhp, canalis nervus hyomandibularis proximalis; cnv, canalis nervus vidianus; cprnv, canalis pro ramo nervi vidiani; faccb, foramen anterius canalis carotici basisphenoidalis; facnv, foramen anterius canalis nervi vidiani; gg, geniculate ganglion; pt, pterygoid; vcl, vena capitis lateralis; VII, nervus facialis; VIIhy, nervus hyomandibularis; VIIvi, nervus vidiani.

Mandibular artery—We are not able to find any traces of a mandibular artery in our sample of podocnemidids. This is consistent with Albrecht (1976) observation that the external and internal mandibular arteries of podocnemidids branch off the stapedial artery prior to and after its passage through the canalis stapedio-temporalis, respectively.

Canalis cavernosus—The canalis cavernosus (Figs. 4A–4D) extends posteriorly from the foramen cavernosum to the level of the foramen stapedio-temporale in all podocnemidids and is formed by the quadrate, prootic, and pterygoid. The foramen cavernosum is formed by the prootic and pterygoid.

Facial nerve canal system—The facial nerve canal is short and extends mediolaterally through the prootic (Figs. 4A and 4D). At mid-distance between the canalis cavernosus and the fossa acustico-facialis, that is, within the prootic, the facial nerve canal bifurcates into two branches, indicating the position of the geniculate ganglion and the split of the facial nerve into the hyomandibular and vidian branches (Figs. 4A and 4D). In our first specimen of Podocnemis unifilis (NHMUK 60.4.16.9), the hyomandibular branch extends posterolaterally through the prootic through a canalis nervus hyomandibularis proximalis. This canal coalesces with the canalis cavernosus (Figs. 4A and 4D). For most of its distal course along the canalis cavernosus, the hyomandibular nerve extends along a sulcus in the wall of the canalis cavernosus. However, the sulcus becomes a proper canalis nervus hyomandibularis distalis in the posterior-most section of the canalis cavernosus, just before entering the cavum acustico-jugulare (Fig. 4B). In our second specimen of Podocnemis unifilis (FMNH 45657), a short but distinct canalis nervus hyomandibularis distalis joining the cavum acustico-jugulare is present. In our first specimen of Podocnemis unifilis (NHMUK 60.4.16.9), the vidian nerve extends ventrally through the canalis pro ramo nervi vidiani, which is formed by the prootic, to enter the cavum pterygoidei (Figs. 4A, 4C and 4D). The vidian nerve is then inferred to first extend through the cavum pterygoidei to then pierce the full anteroposterior length of the pterygoid, where it splits into several branches (Fig. 4D). These branches extend through the pterygoid, palatine, and parietal, and exit the skull through various foramina formed by these bones (Figs. 4A–4D). The path is inferred to be similar for the other specimen of Podocnemis unifilis (FMNH 45657), but the vidian nerve first crosses the above-mentioned fenestra to enter the sulcus cavernosus before entering the pterygoid and only splits into two branches within that bone. The lateral one extends through the pterygoid and its anterior foramen is formed by the pterygoid and the palatine. The medial one extends through the pterygoid, palatine, and parietal, and its anterior foramen is formed by the palatine and parietal. Siebenrock (1897) noted for Erymnochelys madagascariensis that the vidian nerve exits the prootic posteromedially to cross the cavum pterygoidei (his canalis caroticus internus), and then enters the pterygoid, traverses it in its own canal, and exits the pterygoid on its dorsal surface, posterior to the orbit and medial to foramen palatinum posterius. Albrecht (1976) noticed that the vidian nerve mainly extends through the pterygoid as well.

Pelomedusidae (Fig. 5)

Canalis caroticus internus—In Pelomedusa subrufa and Pelusios subniger, the carotid artery enters the skull from ventral and the foramen posterius canalis carotici interni (Fig. 5A–5C) is formed by the basisphenoid and the prootic. The internal carotid artery is directed anteromedially and extends through the prootic and the basisphenoid. The canalis caroticus internus intersects with the facial nerve system at the inferred position of the geniculate ganglion dorsally in the prootic close to the foramen posterius canalis carotici interni (Figs. 5A and 5D). This intersection with the facial nerve system is superficial so that the vidian nerve never fully enters the canalis caroticus internus (Figs. 5A and 5D). The intersection occurs in the prootic in Pelusios subniger and along the prootic-basisphenoid suture in Pelomedusa subrufa. This slightly differs from Albrecht (1976) observations, who noted that this intersection occurred only in the prootic in both species. In Pelomedusa subrufa, the intersection is exposed ventrally. Albrecht (1976) also noted the absence of the palatine artery and foramen anterius canalis carotici lateralis in dissected specimens of Pelomedusa subrufa and Pelusios subniger. As we do not observe any canal in the expected position of the canalis caroticus lateralis, we here confirm observations of Albrecht (1976) and conclude that the canalis caroticus lateralis is absent in pelomedusids due to the absence of the palatine artery. For the same reason, the canalis carotico-pharyngealis is absent as well. The canalis caroticus basisphenoidalis, herein defined as beginning with the entry of the carotid system into the basisphenoid, and foramen anterius canalis carotici basisphenoidalis are formed by the basisphenoid (Fig. 5A). The foramina anterius canalis carotici basisphenoidalis are widely-spaced (Fig. 5A). The canalis stapedio-temporalis is the largest canal, being four times larger than the canalis caroticus internus and canalis caroticus basisphenoidalis in Pelomedusa subrufa, and about three times larger than them in Pelusios subniger (Table 2).

Figure 5 The carotid circulation and vidian canal system of Pelusios subniger (NMB 16230).

Three-dimensional reconstructions of the basisphenoid, right pterygoid, in (A) dorsal, (B) ventral, and (C) left lateral view. Illustration in dorsal view (D) highlighting the placement of relevant arteries, nerves, and veins. Dark colors highlight sections fully covered by bone, light colors partially or fully uncovered sections. Abbreviations: ac, arteria carotis cerebralis; aci, arteria carotis interna; bs, basisphenoid; ccb, canalis caroticus basisphenoidalis; cnh, canalis nervus hyomandibularis; cci, canalis caroticus internus; ccv, canalis cavernosus; cnf, canalis nervus facialis; cnhp, canalis nervus hyomandibularis proximalis; cnv, canalis nervus vidianus; faccb, foramen anterius canalis carotici basisphenoidalis; facnv, foramen anterius canalis nervi vidiani; fpcci, foramen posterius canalis carotici interni; gg, geniculate ganglion; pt, pterygoid; vcl, vena capitis lateralis; VII, nervus facialis; VIIhy, nervus hyomandibularis; VIIvi, nervus vidiani.

Mandibular artery—Albrecht (1976) documented that the mandibular artery of pelomedusids branches from the stapedial artery after its passage through the stapedio-temporal canal. Evers & Benson (2019), on the other hand, reported the presence of a foramen arteriomandibulare for Pelomedusa subrufa, which suggests passage of the mandibular artery through the sulcus cavernosus. After restudying the same CT scans as used by Evers & Benson (2019), however, we support the initial conclusion of Albrecht (1976), as the reported foramen arteriomandibulare appears to be an ontogenetic artifact (see “Discussion” for more details).

Canalis cavernosus—The canalis cavernosus (Figs. 5A–5D) extends posteriorly from the foramen cavernosum to the level of the foramen stapedio-temporale in Pelomedusa subrufa, but posterior to the foramen stapedio-temporale in Pelusios subniger. In Pelusios subniger, the canalis cavernosus extends posterolaterally through the prootic, whereas in Pelomedusa subrufa, the quadrate contributes to the lateral border of the canalis cavernosus for much of its path. The foramen cavernosum is formed by the prootic in both species.

Facial nerve canal system—The facial nerve extends laterally from the braincase toward the canalis caroticus internus through the prootic (Figs. 5A and 5D). The geniculate ganglion is inferred to contact the canalis caroticus internus in both species and the canalis pro ramo nervi vidiani is consequently absent (Figs. 5A and 5D). In Pelusios subniger, a long canalis nervus hyomandibularis proximalis is present and directed posterolaterally through the prootic (Figs. 5A–5D). The canalis nervus hyomandibularis proximalis joins the ventral part of the canalis cavernosus at the level of the bifurcation between the canalis cavernosus and canalis stapedio-temporalis. In Pelomedusa subrufa, the hyomandibular branch almost directly enters the canalis cavernosus and is therefore mainly visible as a sulcus alongside the wall of the latter. At the level of the contact between the geniculate ganglion and the internal carotid artery, the vidian nerve leaves the canalis caroticus internus. In Pelusios subniger, the split of the facial nerve occurs near the dorsal part of the canalis caroticus internus and the vidian nerve directly enters a separate canalis nervus vidianus (Figs. 5A and 5D). This canal first shortly extends through the prootic and along the suture formed between the prootic, pterygoid, and basisphenoid. The vidian nerve then continues anteriorly through the pterygoid and the foramen anterius canalis nervi vidiani is formed by the pterygoid (Figs. 5A–5D). In Pelomedusa subrufa, the split of the facial nerve occurs on the dorsolateral part of the canalis caroticus internus and for a short distance, the vidian nerve extends through the canalis caroticus internus. The vidian nerve then branches off the canalis caroticus internus and extends out of the skull, ventral to the prootic and basisphenoid, before re-entering the skull through the pterygoid via a canalis nervus vidianus. The vidian nerve then extends anteriorly through the pterygoid, and the foramen anterius canalis nervi vidiani is formed by the pterygoid.

Carettochelyidae (Fig. 6)

Canalis caroticus internus—The internal carotid artery enters the skull close to the posterior end of the basisphenoid and the foramen posterius canalis carotici interni (Figs. 6A and 6B) is formed by the pterygoid only. The canalis caroticus internus extends anteromedially mostly through the pterygoid, and enters the basisphenoid, at which point it transforms into the canalis caroticus basisphenoidalis, considering that the lateral carotid canal is absent (Figs. 6A and 6B). Albrecht (1976) reported that the foramen anterius canalis carotici lateralis is either absent or small, but did not draw any firm conclusions about a potentially lacking palatine artery as he only had access to skeletal material. CT scans of a stained specimen available to us however show that the only canal splitting from the canalis caroticus internus contains the vidian nerve, which allows us to conclude that the palatine artery and canalis caroticus lateralis are absent. The canalis pro ramo nervi vidiani joins the canalis caroticus internus halfway along its path (Figs. 6B and 6D). Somewhat anterior to this connection, just before the origin of the canalis caroticus basisphenoidalis, and along the pterygoid-basisphenoid suture, the vidian nerve leaves the canalis caroticus internus laterally to form the vidian canal (Figs. 6A, 6B and 6D). The canalis carotico-pharyngealis is absent. The canalis caroticus basisphenoidalis is formed by the basisphenoid. The foramina anterius canalis carotici basisphenoidalis are widely separated from each other and formed by the basisphenoid (Fig. 6A). The canalis stapedio-temporalis, canalis caroticus internus, and canalis caroticus basisphenoidalis are relatively large and roughly equal-sized (Table 2).

Figure 6 The carotid circulation and vidian canal system of Carettochelys insculpta (NHMUK 1903.7.10.1).

Three-dimensional reconstructions of the basisphenoid, right pterygoid, in (A) dorsal, (B) ventral, and (C) left lateral view. Illustration in dorsal view (D) highlighting the placement of relevant arteries, nerves, and veins. Dark colors highlight sections fully covered by bone, light colors partially or fully uncovered sections. The red portion of the canalis cavernosus shows the inferred position of the mandibular artery. Abbreviations: ac, arteria carotis cerebralis; aci, arteria carotis interna; am, arteria mandibularis; bs, basisphenoid; ccb, canalis caroticus basisphenoidalis; cci, canalis caroticus internus; ccv, canalis cavernosus; cnf, canalis nervus facialis; cnv, canalis nervus vidianus; cprnv, canalis pro ramo nervi vidiani; faccb, foramen anterius canalis carotici basisphenoidalis; facnv, foramen anterius canalis nervi vidiani; fpcci, foramen posterius canalis carotici interni; gg, geniculate ganglion; pt, pterygoid; vcl, vena capitis lateralis; VII, nervus facialis; VIIhy, nervus hyomandibularis; VIIvi, nervus vidiani.

Mandibular artery—Joyce, Volpato & Rollot (2018) previously noted the absence of the canalis caroticus lateralis and speculated by reference to trionychids, where the mandibular artery passes through the lateral carotid canal, that the mandible of Carettochelys insculpta may be supplied by the cerebral artery instead. CT scans of the aforementioned stained specimen, however, clarify that the mandible of Carettochelys insculpta is supplied by the mandibular artery, which branches off the stapedial artery and extends through the canalis cavernosus and the posteriorly elongated trigeminal foramen (Fig. 6D). The only recently reported foramen arteriomandibulare apparent in carettochelyids (Evers & Benson, 2019) serves as an excellent osteological correlate.

Canalis cavernosus—The canalis cavernosus (Figs. 6A–6D) is very short and extends posteriorly from the foramen cavernosum to the level of the foramen stapedio-temporale. For most of its path, the canalis cavernosus is bordered by the quadrate laterally and ventrolaterally, and by the prootic dorsally, medially and ventromedially. Slightly posterior to the foramen cavernosum, the pterygoid minorly contributes to the ventral limit of the canalis cavernosus. The foramen cavernosum is formed by the quadrate, prootic, and pterygoid.

Facial nerve canal system—The facial nerve course of Carettochelys insculpta was already described in detail by Joyce, Volpato & Rollot (2018) based on the same specimen and CT scans. The facial nerve extends laterally from the cavum cranii through the prootic. At two-thirds of the distance between the fossa acustico-facialis and the canalis cavernosus, the canalis nervus facialis splits into canals for the vidian and hyomandibular branches (Figs. 6A and 6D). The aforementioned stained specimen confirms a geniculate ganglion position within the prootic, although the position is close to the wall of the canalis cavernosus and not as deeply within the prootic as in many pleurodires. The hyomandibular nerve extends laterally along its own proximal canal through the prootic for a short distance and joins the canalis cavernosus, from which it further extends posteriorly within a sulcus (Figs. 6A and 6D). The canalis pro ramo nervi vidiani is small, extends through the prootic and pterygoid, and connects to the canalis caroticus internus halfway along its length (Figs. 6B and 6D). The vidian nerve leaves the canalis caroticus internus laterally at the basisphenoid-pterygoid suture (Figs. 6A, 6B and 6D). Its proximal part is located in the pterygoid, the most distal portion fully in the palatine. The canalis nervus vidianus ends in a series of various small canals that open on the dorsal and ventral surfaces of the palatine (Figs. 6A, 6B and 6D).

Trionychidae (Fig. 7)

Canalis caroticus internus—In all observed taxa, the internal carotid artery enters the skull from ventral and the foramen posterius canalis carotici interni (Figs. 7A–7C) is formed by the pterygoid only. The canalis caroticus internus (Figs. 7A–7D) is formed by the pterygoid and basisphenoid in all trionychid specimens with minor contributions of the prootic in Amyda cartilaginea, Apalone mutica, Apalone spinifera, Chitra indica, Cyclanorbis senegalensis, and Pelodiscus sinensis. Jamniczky & Russell (2007) only identified the basisphenoid and pterygoid as contributing to the canalis caroticus internus in Ap. mutica and Trionyx triunguis. The differences in observation for Ap. mutica may be due to intraspecific variation or error in observation. Many other such differences exist for other taxa and structures. As these possibilities can only be tested in the context of much greater sampling, we here assume that all such differences pertain to intraspecific variation, while favoring our observations in the tables. In Am. cartilaginea, Chitra indica, P. sinensis, and Cycloderma frenatum, the internal carotid artery is exposed dorsally, that is, in the floor of the cavum acustico-jugulare, for a short distance between the foramen posterius canalis carotici interni and the canalis pro ramo nervi vidiani. The canalis pro ramo nervi vidiani enters the canalis caroticus internus midway and the vidian nerve follows the path of the canalis caroticus lateralis (Fig. 7D). In Cycloderma frenatum, the vidian nerve splits directly from the canalis caroticus internus instead of following the path of the canalis caroticus lateralis. The split of the internal carotid artery into its two subbranches, in this case the cerebral and mandibular arteries (Albrecht, 1967), occurs at the same level as the sella turcica (Figs. 7A and 7B). The canalis caroticus lateralis and canalis caroticus basisphenoidalis are short. Indeed, the canalis caroticus lateralis is shortened to the length of a fenestra. The mandibular artery joins the sulcus cavernosus through this fenestra, which is formed by the pterygoid and basisphenoid in Am. cartilaginea, Ap. mutica, Ap. spinifera, Chitra indica, and P. sinensis, but only by the pterygoid in Cyclanorbis senegalensis, Cycloderma frenatum and L. punctata. The canalis carotico-pharyngealis is absent, as observed by Albrecht (1967) in Trionyx spp. In all examined trionychids, the cerebral branch exits through the sella turcica. The foramina anterius canalis carotici basisphenoidalis are widely-spaced and formed by the basisphenoid (Fig. 7B). The canalis caroticus basisphenoidalis is larger than the canalis stapedio-temporalis and canalis caroticus lateralis in all observed trionychids. However, some differences as to which canal is the smallest are apparent across taxa. The canalis caroticus lateralis is the smallest in Am. cartilaginea, C. indica, and Cycloderma frenatum, whereas the canalis stapedio-temporalis is the smallest in Ap. mutica, Cyclanorbis senegalensis, L. punctata, and P. sinensis. The canalis stapedio-temporalis and canalis caroticus lateralis are equal-sized in Ap. spinifera (Table 2). This corroborates Albrecht (1976), who noted that trionychids have a small stapedial artery and a large carotid artery, but slightly differs from Jamniczky & Russell (2007), who identified a foramen anterius canalis carotici lateralis that is larger than the foramen anterius canalis carotici basisphenoidalis in Ap. mutica and Trionyx triunguis. The structures we identify in trionychids are identical to the ones identified by Albrecht (1967, 1976).

Figure 7 The carotid circulation and vidian canal system of Apalone spinifera (FMNH 22178).

Three-dimensional reconstructions of the basisphenoid, right pterygoid, in (A) dorsal, (B) ventral, and (C) left lateral view. Illustration in dorsal view (D) highlighting the placement of relevant arteries, nerves, and veins. Dark colors highlight sections fully covered by bone, light colors partially or fully uncovered sections. Abbreviations: ac, arteria carotis cerebralis; aci, arteria carotis interna; am, arteria mandibularis; bs, basisphenoid; ccb, canalis caroticus basisphenoidalis; cci, canalis caroticus internus; ccl, canalis caroticus lateralis; ccv, canalis cavernosus; cnf, canalis nervus facialis; cnv, canalis nervus vidianus; faccb, foramen anterius canalis carotici basisphenoidalis; faccl, foramen anterius canalis caroticus lateralis; facnv, foramen anterius canalis nervi vidiani; fpcci, foramen posterius canalis carotici interni; gg, geniculate ganglion; pt, pterygoid; vcl, vena capitis lateralis; VII, nervus facialis; VIIhy, nervus hyomandibularis; VIIvi, nervus vidiani.

Canalis cavernosus—The canalis cavernosus (Figs. 7A–7D) extends from the foramen cavernosum to the level of the foramen stapedio-temporale or slightly posterior to it. The canalis cavernosus is formed by the pterygoid, prootic, and quadrate in all observed taxa except Cycloderma frenatum, in which the canalis cavernosus is only bordered by the prootic and pterygoid. The foramen cavernosum is formed by the prootic and pterygoid.

Facial nerve canal system—The facial nerve extends laterally from the fossa acustico-facialis through the prootic (Figs. 7A and 7D). The geniculate ganglion is inferred to be located in the canalis cavernosus in Am. cartilaginea, Ap. mutica, Ap. spinifera (Figs. 7A and 7D), and P. sinensis, but in between the canalis cavernosus and canalis caroticus internus in Chitra indica, Cyclanorbis senegalensis, Cycloderma frenatum, and L. punctata. This observation contradicts Gaffney (1979), who stated that the geniculate ganglion is always in contact with the canalis cavernosus in cryptodires. The facial nerve splits into two branches in all observed trionychids but Cyclanorbis senegalensis, the ventral one being the vidian nerve, and the lateral one being the hyomandibular branch (Fig. 7D). In Cyclanorbis senegalensis, a third branch is present that extends anteriorly along the prootic-pterygoid suture to join the sulcus cavernosus. The distal portion of the hyomandibular branch is fully confluent with the canalis cavernosus in Am. cartilaginea, is present as a sulcus connected to the canalis cavernosus in Ap. spinifera, Ap. mutica, and P. sinensis, and extends posterolaterally through its own canalis nervus hyomandibularis proximalis along the prootic-pterygoid suture in Chitra indica, Cyclanorbis senegalensis, Cycloderma frenatum and L. punctata. In all observed trionychids, the vidian nerve is contained in the canalis pro ramo nervi vidiani and extends ventrally from the geniculate ganglion through the prootic and pterygoid to join the canalis caroticus internus midway (Fig. 7D). In all taxa but Cycloderma frenatum, the vidian branch likely initially follows the course of the canalis caroticus internus, but then passes through the fenestra-like canalis caroticus lateralis to enter the sulcus cavernosus. From there, the nerve pierces either the pterygoid or the palatine to form the canalis nervus vidianus (Figs. 7A and 7D). In Cycloderma frenatum, the vidian branch leaves the canalis caroticus internus laterally between the canalis pro ramo nervi vidiani and canalis caroticus lateralis and then joins the sulcus cavernosus. Bender (1906) described a similar pattern in Apalone ferox, noting that the vidian nerve enters a canal formed by the basisphenoid and pterygoid (our canalis caroticus internus) and extends anteriorly along the suture made by these two bones. The vidian nerve then bifurcates, extends anteriorly along the pterygoid-palatine suture, and connects with the maxillary branch of the trigeminal nerve (V2) outside of the bone to form the sphenopalatine ganglion, at the level of the contact between the pterygoid and the maxilla. Our observations for trionychids in general correspond to those of Ogushi (1911) for P. sinensis and Shiino (1913) for the vidian nerve in Am. cartilaginea. The canalis nervus vidianus is formed by the palatine in Ap. spinifera, Chitra indica, P. sinensis, and Cyclanorbis senegalensis, by the pterygoid and palatine in Ap. mutica, Am. cartilaginea and L. punctata, and by the pterygoid, palatine and parietal in Cycloderma frenatum. In Ap. spinifera, Am. cartilaginea, and P. sinensis, the canalis nervus vidianus splits into several smaller canals that exit the skull on the dorsal and ventral surfaces of the palatine. Much of this variation was previously noted by Meylan (1987) for a broader sample. The foramen anterius canalis nervi vidiani is formed by the palatine in all observed trionychids.

Kinosternidae (Fig. 8)

Canalis caroticus internus—In all observed kinosternids, the internal carotid artery initially enters the skull through the fenestra postotica and then passes through a groove within the cavum acustico-jugulare that is roofed by the prootic, as previously noted by Evers & Benson (2019). The actual foramen posterius canalis carotici interni (Figs. 8A–8C) is formed by the pterygoid and prootic, and the canalis caroticus internus (Figs. 8A and 8B) is formed by the pterygoid, prootic, and basisphenoid. This slightly differs from the observations of Siebenrock (1897), who did not mention the involvement of the basisphenoid in the formation of the canalis caroticus internus. Jamniczky & Russell (2007) only mentioned the pterygoid as forming the foramen posterius canalis carotici interni, and the basisphenoid and pterygoid as forming the canalis caroticus internus in Kinosternon baurii and Staurotypus salvinii. The canalis pro ramo nervi vidiani connects to the canalis caroticus internus just before its split into its palatine and cerebral branches (Albrecht, 1976; Figs. 8B and 8D). The canalis caroticus lateralis and canalis caroticus basisphenoidalis are short. The canalis caroticus lateralis joins the sulcus cavernosus. The canalis caroticus lateralis and foramen anterius canalis carotici lateralis are formed by the prootic, pterygoid, and basisphenoid in Sternotherus minor, Kinosternon subrubrum, and Staurotypus salvinii, by the pterygoid and basisphenoid in K. baurii, and only by the pterygoid in Kinosternon scorpioides. Two canalis carotico-pharyngealis are present in K. scorpioides and Staurotypus salvinii. The cerebral branch exits at the sella turcica. The canalis caroticus basisphenoidalis and foramen anterius canalis carotici basisphenoidalis are formed by the basisphenoid. The foramina anterius canalis carotici basisphenoidalis are widely separated (Fig. 8A). The canalis caroticus lateralis has a greater cross-section than the canalis caroticus basisphenoidalis and canalis stapedio-temporalis in all observed kinosternids, and the canalis stapedio-temporalis, which is absent in K. scorpioides, is the smallest canal in K. baurii, K. subrubrum, Staurotypus salvinii, and Sternotherus minor (Table 2). With the exception of K. baurii, for which Jamniczky & Russell (2007) were not able to identify the canalis stapedio-temporalis and foramen stapedio-temporalis, the relative canal sizes we observe in our specimens confirm the observations of McDowell (1961), Albrecht (1967, 1976), and Jamniczky & Russell (2007).

Figure 8 The carotid circulation and vidian canal system of Sternotherus minor (FMNH 211696).

Three-dimensional reconstructions of the basisphenoid, right pterygoid, in (A) dorsal, (B) ventral, and (C) left lateral view. Illustration in dorsal view (D) highlighting the placement of relevant arteries, nerves, and veins. Dark colors highlight sections fully covered by bone, light colors partially or fully uncovered sections. Abbreviations: ac, arteria carotis cerebralis; aci, arteria carotis interna; ap, arteria palatina; am, arteria mandibularis; bs, basisphenoid; ccb, canalis caroticus basisphenoidalis; cci, canalis caroticus internus; ccl, canalis caroticus lateralis; ccv, canalis cavernosus; cnf, canalis nervus facialis; cnv, canalis nervus vidianus; cprnv, canalis pro ramo nervi vidiani; faccb, foramen anterius canalis carotici basisphenoidalis; faccl, foramen anterius canalis caroticus lateralis; facnv, foramen anterius canalis nervi vidiani; fpcci, foramen posterius canalis carotici interni; gg, geniculate ganglion; pt, pterygoid; vcl, vena capitis lateralis; VII, nervus facialis; VIIhy, nervus hyomandibularis; VIIvi, nervus vidiani.

Mandibular artery—The mandibular artery (Fig. 8D) branches from the palatine artery within the sulcus cavernosus and exits through the trigeminal foramen (Albrecht, 1967). It is therefore not surprising that we cannot find any osteological correlates for this artery beyond the enormous size of the anterior foramen of the lateral carotid canal.

Canalis cavernosus—The canalis cavernosus (Figs. 8A–8D) extends posteriorly from the foramen cavernosum and joins the cavum acustico-jugulare slightly posterior to the level of the foramen stapedio-temporale. It is formed by the quadrate, prootic, and pterygoid. The foramen cavernosum is formed by the prootic and pterygoid.

Facial nerve canal system—The facial nerve extends ventrolaterally from the cavum cranii to the canalis cavernosus through the prootic (Figs. 8A and 8D). The geniculate ganglion is inferred to be located in the canalis cavernosus and splits into two branches in all observed kinosternids (Figs. 8A and 8D). The hyomandibular nerve extends posterolaterally within the canalis cavernosus, visible as a sulcus in K. subrubrum, K. baurii, K. scorpioides, and Sternotherus minor (Figs. 8A and 8D), but undistinguishable from the canalis cavernosus in Staurotypus salvinii. In all observed kinosternids, the vidian nerve is contained in a short canalis pro ramo nervi vidiani that extends ventrally through the prootic and pterygoid to join the canalis caroticus internus (Figs. 8B and 8D). In Sternotherus minor, K. subrubrum, K. baurii, and K. scorpioides, the vidian nerve extends within the canalis caroticus internus and then anteriorly within the canalis caroticus lateralis, from which it exits into the sulcus cavernosus. Slightly more anteriorly in the sulcus cavernosus, approximately at the level of the dorsum sellae of the basisphenoid, the vidian nerve enters its own canal (Figs. 8A–8D). This canal, the canalis nervus vidianus, is formed by the pterygoid and palatine, with some contributions of the parietal in K. baurii, and of the epipterygoid in K. scorpioides. Within the palatine, the vidian nerve splits into several branches. One of these branches extends to the foramen palatinum posterius, while the others either become untraceable in the porosity of the bone, or connect to the dorsal or ventral surfaces of the palatine (likely forming, respectively, the foramina arteriae anteriovidianae and foramina arteriaevidianae of Albrecht, 1967). In Staurotypus salvinii, a canal leaves the canalis caroticus internus ventrolaterally and merges with a canal coming from ventral. This complex then becomes confluent with another canal coming from the canalis caroticus internus, and the resulting canal divides more anteriorly into two branches. The largest one extends anterodorsally and joins the sulcus cavernosus anterior to the foramen anterius canalis carotici lateralis. The smallest one extends anteriorly through the pterygoid and palatine, likely corresponding to the canal for the vidian nerve. The vidian nerve exits the skull on the ventromedial side of the palatine, posterior to the anterior end of the pterygoid. More anteriorly, the vidian nerve likely re-enters the skull, merges with unidentified canals, and exits near the foramen palatinum posterius inside the palatine. The pattern we observe for the vidian nerve differs in two respects from the descriptions of McDowell (1961) and Albrecht (1967). Firstly, McDowell (1961) only mentioned the pterygoid as the bone forming the vidian canal, whereas we observed a higher complexity of the innervation pattern, with the palatine, parietal, and epipterygoid contributing to the formation of the canalis nervus vidianus. Secondly, Albrecht (1967) noted that some branches of the vidian nerve leave the canalis pro ramo nervi vidiani to extend anteriorly through the pterygoid, and anteriorly connect with other branches of the vidian nerve that exit the canalis caroticus lateralis. Although we cannot exclude that Albrecht (1967) indeed observed that in the specimens he dissected, we have not been able to observe this pattern in our specimens and the canalis nervus vidianus is either exclusively connected to the sulcus cavernosus (Sternotherus minor, K. subrubrum, K. baurii, and K. scorpioides), or the canalis caroticus internus (Staurotypus salvinii).

Dermatemydidae (Fig. 9)

Canalis caroticus internus—In Dermatemys mawii, the internal carotid artery enters the skull through the fenestra postotica. The posterior course of the internal carotid artery can be inferred from a dorsally open trough in the pterygoid. The internal carotid artery only becomes fully enclosed by bone within the cavum acustico-jugulare, where the prootic covers the pterygoid to form the actual foramen posterius canalis carotici interni (Fig. 9A). The internal carotid artery is bordered by the prootic, pterygoid, and basisphenoid. The canalis pro ramo nervi vidiani enters the internal carotid canal about mid-length. Slightly posterior to the split into the canalis caroticus lateralis and canalis caroticus basisphenoidalis, a small canal of unclear function or homology connects the canalis caroticus internus to the canalis cavernosus on the right side of the available specimen. The split of the canalis caroticus internus into the canalis caroticus basisphenoidalis and canalis caroticus lateralis occurs along the pterygoid-basisphenoid suture (Figs. 9A and 9B). The canalis caroticus lateralis is short and formed by the prootic, pterygoid, and basisphenoid. The herein contained palatine artery (Albrecht, 1976) joins the sulcus cavernosus by the way of the foramen anterius canalis carotici lateralis, which is formed by the pterygoid and basisphenoid (Figs. 9A and 9B). A canalis carotico-pharyngealis is absent. The canalis caroticus basisphenoidalis extends anteromedially through the basisphenoid and joins the braincase via the widely-spaced foramina anterius canalis carotici basisphenoidalis, which are formed by the basisphenoid (Fig. 9A). The canalis caroticus lateralis is about four times bigger in cross section than the canalis caroticus basisphenoidalis (Table 2), and the canalis stapedio-temporalis and foramen stapedio-temporale are absent, possibly explaining the particularly large diameter of the canalis caroticus lateralis.

Figure 9 The carotid circulation and vidian canal system of Dermatemys mawii (SMF 59463).

Three-dimensional reconstructions of the basisphenoid, right pterygoid, in (A) dorsal, (B) ventral, and (C) left lateral view. Illustration in dorsal view (D) highlighting the placement of relevant arteries, nerves, and veins. Dark colors highlight sections fully covered by bone, light colors partially or fully uncovered sections. The red portion of the canalis cavernosus shows the inferred position of the mandibular artery. Abbreviations: ac, arteria carotis cerebralis; aci, arteria carotis interna; am-a, anterior arteria mandibularis; am-p, posterior arteria mandibularis; ap, arteria palatina; bs, basisphenoid; ccb, canalis caroticus basisphenoidalis; cci, canalis caroticus internus; ccl, canalis caroticus lateralis; ccv, canalis cavernosus; cnf, canalis nervus facialis; cnv, canalis nervus vidianus; faccb, foramen anterius canalis carotici basisphenoidalis; faccl, foramen anterius canalis caroticus lateralis; facnv, foramen anterius canalis nervi vidiani; fpcci, foramen posterius canalis carotici interni; gg, geniculate ganglion; pt, pterygoid; vcl, vena capitis lateralis; VII, nervus facialis; VIIhy, nervus hyomandibularis; VIIvi, nervus vidiani.

Mandibular artery—We are unaware of any literature that explicitly discusses the placement of the mandibular artery in Dermatemys mawii. Its branching from the stapedial artery posterior to the stapedial canal and exit through the trigeminal foramen is suggested by a lateral sulcus within the canalis cavernosus. This sulcus is otherwise only apparent in turtles for which the mandibular artery extends through the canalis cavernosus. However, the large diameter of the lateral carotid canal, on the other side, suggests that the mandible is perhaps partially or fully supplied by the palatine artery as well. The taxa that phylogenetically bracket Dermatemys mawii do not provide any further evidence, as the mandible is supplied by the stapedial artery in chelydrids, but by the palatine artery in kinosternids. As we believe the division of the canalis cavernosus and the relative size of the lateral carotid canal to be useful osteological correlates, we reconstruct Dermatemys mawii as having a split mandibular artery (Fig. 9D), as seen in Cheloniidae. This issue will need to be resolved in the future, however, by reference to wet specimens.

Canalis cavernosus—The canalis cavernosus (Figs. 9A–9D) extends posteriorly from the foramen cavernosum and joins the cavum acustico-jugulare slightly anterior to the level of the columella auris. The canalis cavernosus is formed by the quadrate, prootic, and pterygoid. The foramen cavernosum is formed by the prootic and pterygoid.

Facial nerve canal system—The facial nerve canal extends ventrolaterally from the cavum cranii to the canalis cavernosus through the prootic (Figs. 9A and 9D). As there are no osteological indications for the split of the facial nerve to occur prior to the facial nerve canal joining the canalis cavernosus, the position of the geniculate ganglion is inferred to be within the latter (Figs. 9A and 9D). And as no osteological correlate for the hyomandibular branch of the facial nerve could be identified within the canalis cavernosus, its course is inferred to be posteriorly directed within that structure (Fig. 9D). A small canalis pro ramo nervi vidiani is visible on the right side only in the available specimen that extends along the prootic-pterygoid suture and joins the canalis caroticus internus mid-length. The vidian nerve is then inferred to follow the path of the canalis caroticus internus and the canalis caroticus lateralis into the sulcus cavernosus (Fig. 9D). Anterior to the rostrum basisphenoidale and posteroventromedially to the processus inferior parietalis, the vidian nerve leaves the sulcus cavernosus ventrolaterally to enter the canalis nervus vidianus, extends anteriorly through the palatine, and ends at the foramen palatinum posterius (Figs. 9A–9D).

Chelydridae (Fig. 10)

Canalis caroticus internus—The internal carotid artery enters the skull slightly anterior to the level of the columella auris through the fenestra postotica. The artery forms a dorsally exposed sulcus within the cavum acustico-jugulare in Macrochelys temminckii, but not in Chelydra serpentina. The foramen posterius canalis carotici interni (Fig. 10A) within the cavum acustico-jugulare is formed by the prootic and pterygoid in C. serpentina, but only by the pterygoid in M. temminckii, as noted by Siebenrock (1897). The canalis caroticus internus (Fig. 10B) is formed by the prootic, pterygoid, and basisphenoid. This slightly differs from Siebenrock (1897) who only described the prootic and pterygoid as the bones forming this canal in C. serpentina, and only the pterygoid in M. temminckii, and from Jamniczky & Russell (2007), who noted that the canalis caroticus internus traverses the pterygoid and basisphenoid in C. serpentina. The canalis pro ramo nervi vidiani connects the canalis cavernosus to the canalis caroticus internus about mid-length of the latter. In both observed chelydrids, the split of the canalis caroticus internus into the canalis caroticus lateralis and canalis caroticus basisphenoidalis occurs at the pterygoid-basisphenoid suture (Figs. 10A and 10B). The canalis caroticus lateralis is much longer than the canalis caroticus basisphenoidalis. The palatine artery (Albrecht, 1976) extends anteriorly through the canalis caroticus lateralis, first through the pterygoid and then along the pterygoid-basisphenoid suture in M. temminckii, but only along the pterygoid-basisphenoid suture in C. serpentina (Figs. 10A–10D). The palatine artery then exits its canal through the foramen anterius canalis carotici lateralis, formed by the pterygoid and basisphenoid (Figs. 10A and 10B). This contrasts with Albrecht (1976), who noted that the foramen anterius canalis carotici lateralis is only formed by the pterygoid in chelydrids. A canalis carotico-pharyngealis is present in both taxa. The canalis caroticus basisphenoidalis extends anteromedially through the basisphenoid and opens within the sella turcica. The foramina anterius canalis carotici basisphenoidalis are widely-spaced in C. serpentina (Fig. 10A) but close together in M. temminckii. The canalis stapedio-temporalis is the largest canal and the canalis caroticus lateralis the smallest in terms of diameters in both specimens. The canalis stapedio-temporalis is about five times larger than the canalis caroticus basisphenoidalis in C. serpentina, and ten times larger in M. temminckii, and more than ten times larger than the canalis caroticus lateralis in both specimens (Table 2). These canal proportions corroborate the results of McDowell (1961), Albrecht (1976), and Jamniczky & Russell (2007).

Figure 10 The carotid circulation and vidian canal system of Chelydra serpentina (SMF 32846).

Three-dimensional reconstructions of the basisphenoid, right pterygoid, in (A) dorsal, (B) ventral, and (C) left lateral view. Illustration in dorsal view (D) highlighting the placement of relevant arteries, nerves, and veins. Dark colors highlight sections fully covered by bone, light colors partially or fully uncovered sections. The red portion of the canalis cavernosus shows the inferred position of the mandibular artery. Abbreviations: ac, arteria carotis cerebralis; aci, arteria carotis interna; ap, arteria palatina; am, arteria mandibularis; bs, basisphenoid; ccb, canalis caroticus basisphenoidalis; cci, canalis caroticus internus; ccl, canalis caroticus lateralis; ccv, canalis cavernosus; cnf, canalis nervus facialis; cnv, canalis nervus vidianus; faccb, foramen anterius canalis carotici basisphenoidalis; faccl, foramen anterius canalis caroticus lateralis; facnv, foramen anterius canalis nervi vidiani; fpcci, foramen posterius canalis carotici interni; gg, geniculate ganglion; pt, pterygoid; vcl, vena capitis lateralis; VII, nervus facialis; VIIhy, nervus hyomandibularis; VIIvi, nervus vidiani.

Mandibular artery—A dorsal groove within the large canalis cavernosus provides an osteological correlate for the presence of two parallel lying vessels, which we interpret as the ventromedially located lateral head vein and the dorsolaterally located mandibular artery, which exits the canalis cavernosus through the trigeminal foramen (Albrecht, 1976; Figs. 10A, 10B and 10D).

Canalis cavernosus—The canalis cavernosus (Figs. 10A–10D) extends from the foramen cavernosum to the level of the foramen stapedio-temporale. It is short in C. serpentina, but long in M. temminckii. The canalis cavernosus is formed by the quadrate, prootic, and pterygoid, and houses the geniculate ganglion. The foramen cavernosum is formed by the prootic and pterygoid.

Facial nerve canal system—The facial nerve canal extends ventrolaterally from the fossa acustico-facialis to the canalis cavernosus through the prootic (Figs. 10A and 10D). The geniculate ganglion is inferred to be positioned within the canalis cavernosus and splits into two branches (Figs. 10A and 10D). The hyomandibular branch has no osteological correlate in M. temminckii, but its course remains visible in C. serpentina as a sulcus in the wall of the canalis cavernosus (Figs. 10A and 10D). In C. serpentina, the canalis pro ramo nervi vidiani is short and located along the prootic-pterygoid suture, whereas it is long in M. temminckii and formed by the pterygoid only. The vidian nerve enters the internal carotid canal at mid-length in both taxa. In C. serpentina, the vidian nerve leaves the canalis caroticus internus just posterior to the split of the canalis caroticus internus into the canalis caroticus lateralis and canalis caroticus basisphenoidalis (Figs. 10A and 10D). The canalis nervus vidianus is long but narrow in diameter, extends anteriorly through the pterygoid, and connects to small exiting canals along its path (Figs. 10A–10D). The main vidian canal opens onto the dorsal surface of the pterygoid, anterior to the foramen anterius canalis carotici lateralis, and posteromedially to the anterior limit of the processus inferior parietalis (Figs. 10A–10D). Siebenrock (1897) described the vidian nerve of C. serpentina as extending through a groove lateral to the internal carotid artery, being posterodorsally covered by the prootic, and posteriorly starting from the foramen jugulare anterius, and that the vidian nerve extends through the canalis caroticus lateralis and foramen anterius canalis carotici lateralis. Albrecht (1976) similarly stated that the canal we identify as the vidian canal does not connect to the canalis caroticus internus and that no nerve is contained in it. Our observations differ from these descriptions as our specimen clearly exhibits a canalis nervus vidianus distinct from the canalis caroticus lateralis and that is connected to the canalis caroticus internus, as previously suggested by Gaffney (1972) for this taxon, even though it is hard to distinguish and sometimes barely visible in the CT scans. In M. temminckii, two small canals ventrally leave the canalis caroticus internus near its posterior end and pass through the pterygoid to eventually merge through detours with a canal that extends anteriorly through the pterygoid, lateral to the canalis caroticus internus, to connect with a branch of the canalis carotico-pharyngealis, and to merge with the canalis caroticus lateralis close to the foramen anterius canalis carotici lateralis. This system of canals may have held the vidian nerve. This conclusion differs from Albrecht (1976) who described the canalis nervus vidianus in M. temminckii as connecting to the canalis caroticus internus and bifurcating from it slightly posterior to the split of the canalis caroticus internus into the canalis caroticus lateralis and canalis caroticus basisphenoidalis. His canalis nervus vidianus then extends anteriorly through the pterygoid and opens into the orbit at the anterior end of the crista pterygoidea.

Cheloniidae (Fig. 11)

Canalis caroticus internus—In all observed cheloniids, the internal carotid artery enters the skull from posterior at the posteroventral part of the pterygoid. The foramen posterius canalis carotici interni (Figs. 11A and 11B) is formed by the pterygoid. In Eretmochelys imbricata, Chelonia mydas, and Natator depressus, the canalis caroticus internus (Figs. 11A–11D) is formed by the pterygoid and basisphenoid, but only by the pterygoid in Lepidochelys olivacea and Caretta caretta. In E. imbricata and Chelonia mydas, a narrow and short canal leaves the canalis caroticus internus dorsally at about half of its length to join the cavum labyrinthicum. In E. imbricata, Chelonia mydas, and N. depressus, the canalis pro ramo nervi vidiani connects the canalis cavernosus to the canalis caroticus internus (Figs. 11A, 11B and 11D), but in L. olivacea and Caretta caretta no canalis pro ramo nervi vidiani is visible. Unlike in all other turtles, the internal carotid artery exits the basicranium to enter the sulcus cavernosus prior to its split into subordinate arteries in all observed cheloniids (Figs. 11A and 11D). This had previously been inferred by Zangerl (1953) for Chelonia mydas and observed in dissected specimens of the same species by Albrecht (1976), and then observed for cheloniids more widely by Evers & Benson (2019). The position of the canalis caroticus basisphenoidalis provides osteological evidence for this unusual arrangement. This canal usually diverges off the canalis caroticus internus within the basicranium and at the basisphenoid-pterygoid suture. In cheloniids, however, the canalis caroticus basisphenoidalis diverges from the sulcus cavernosus into the basisphenoid (Figs. 11A and 11D). Therefore, a short part of the internal carotid artery must be inferred to extend dorsally uncovered within the sulcus cavernosus across all cheloniids. Consequentially, the palatine artery, the presence of which has been confirmed for at least Chelonia mydas by Albrecht (1976), is never encased in a canal in cheloniids (Fig. 11D), confirming observations by Evers & Benson (2019), who reported this condition to be present in all extant cheloniids and Dermochelys coriacea, but not in all extinct cheloniids. A characteristic commonly used in turtle systematics (and first used explicitly by Shaffer et al., 1997) is the spacing of the foramina anterius canalis carotici basisphenoidalis, which in most turtles are widely spaced across the sella turcica, but come close together in cheloniids (Fig. 11A) and some extinct turtles, such as plesiochelyids (Gaffney, 1976). Recently, Evers & Benson (2019) defined a third passage for the cerebral arteries, based on the observation that right and left arteries converge within the basisphenoid and exit through a single median foramen anterius canalis carotici basisphenoidalis in L. olivacea, Caretta caretta, and N. depressus, but not in other extant cheloniids. Here, we confirm these observations. As the palatine artery is never encased in a bony canal, the identification of the canals traversing the pterygoid as being the canalis carotico-pharyngealis is not fully certain. However, respectively one and two canals in E. imbricata and Caretta caretta connect the ventral surface of the pterygoid to the portion of the sulcus cavernosus in which the carotid artery lies, making them plausible candidates for being the canalis carotico-pharyngealis. The canalis stapedio-temporalis is about twice as large as the canalis caroticus internus and about four to five times larger than the canalis caroticus basisphenoidalis in all observed cheloniids (Table 2). Although we cannot infer the diameter of the palatine artery in the absence of a formed canalis caroticus lateralis, observations made by Albrecht (1976) show that the palatine artery is larger than the cerebral artery.

Figure 11 The carotid circulation and vidian canal system of Eretmochelys imbricata (FMNH 22242).

Three-dimensional reconstructions of the basisphenoid, right pterygoid, in (A) dorsal, (B) ventral, and (C) left lateral view. Illustration in dorsal view (D) highlighting the placement of relevant arteries, nerves, and veins. Dark colors highlight sections fully covered by bone, light colors partially or fully uncovered sections. The red portion of the canalis cavernosus shows the inferred position of the posterior mandibular artery. Abbreviations: ac, arteria carotis cerebralis; aci, arteria carotis interna; ap, arteria palatina; am-a, anterior arteria mandibularis; am-p, posterior arteria mandibularis; bs, basisphenoid; ccb, canalis caroticus basisphenoidalis; cci, canalis caroticus internus; ccv, canalis cavernosus; cnf, canalis nervus facialis; cprnv, canalis pro ramo nervi vidiani; faccb, foramen anterius canalis carotici basisphenoidalis; fpcci, foramen posterius canalis carotici interni; gg, geniculate ganglion; pt, pterygoid; v-am, vestigial arteria mandibularis; vcl, vena capitis lateralis; VII, nervus facialis; VIIhy, nervus hyomandibularis; VIIvi, nervus vidiani.

Mandibular artery—Albrecht (1976) reported that the mandible of chelonioids is supplied by two separate branches, a vestigial one originating from the stapedial artery and a larger one supplied by the palatine artery. We find osteological evidence of the latter in the form of a sulcus within the canalis cavernosus (Figs. 11A–11C), combined with a foramen arteriomandibulare that is present in Chelonia mydas, but not in other cheloniids. In contrast to Albrecht (1976), however, we note that the foramen arteriomandibulare does not form a short canal in C. mydas, but rather represents a small fenestra to the canalis cavernosus. In addition, the lateral sulcus within the canalis cavernosus in our specimen is suggestive of a large mandibular artery (Figs. 11A–11C). This is further supported by the dried remains of the mandibular artery in our specimen of Eretmochelys imbricata. Although further studies on wet specimens will need to further clarify this issue, we reconstruct chelonioids as having similarly sized anterior and posterior mandibular arteries (Fig. 11D).

Canalis cavernosus—The canalis cavernosus (Figs. 11A–11D) extends from the foramen cavernosum to the level of the foramen stapedio-temporale. The canalis cavernosus is formed by the quadrate, prootic and pterygoid and the foramen cavernosum is formed by the prootic and pterygoid.

Facial nerve canal system—The facial nerve canal extends ventrolaterally from the fossa acustico-facialis to the canalis cavernosus through the prootic (Figs. 11A and 11D). The position of the geniculate ganglion is inferred to be located in the canalis cavernosus, where it branches into the vidian and hyomandibular nerves (Figs. 11A and 11D). Whereas a sulcus for the hyomandibular nerve is visible in the wall of the canalis cavernosus in L. olivacea and Chelonia mydas, the hyomandibular branch of the facial nerve has no osteological correlate in E. imbricata (Figs. 11A and 11D), Caretta caretta, and N. depressus, and is thus inferred to be fully contained within the canalis cavernosus. The canalis pro ramo nervi vidiani is present in E. imbricata, Chelonia mydas, and N. depressus, formed by the pterygoid, and connects the canalis cavernosus to the canalis caroticus internus (Figs. 11A, 11B and 11D). No canalis pro ramo nervi vidiani is visible in the specimens we used for L. olivacea and Caretta caretta and the path of the vidian nerve is therefore unclear in these two taxa. No canalis nervus vidianus is visible in any observed cheloniids. Siebenrock (1897) noted that the vidian nerve of Caretta caretta, Chelonia mydas, and E. imbricata is contained in the sulcus in which the internal carotid artery lies once it has joined the sulcus cavernosus, and Soliman (1964) clearly shows a vidian nerve in his illustrations of the cranial nerves in Eretmochelys imbricata. In the absence of separate canals for the vidian nerve, we infer that this nerve usually passes through the canalis caroticus internus and then through the sulcus cavernosus, but does not form an anterior vidian canal (Fig. 11D).

Dermochelyidae (Fig. 12)

Canalis caroticus internus—The internal carotid artery enters the skull from posterior through the fenestra postotica. The actual foramen posterius canalis carotici interni is positioned deeply within the cavum acustico-jugulare and formed by the prootic and pterygoid. Unlike in most cryptodires, in which the foramen posterius canalis carotici interni and canalis caroticus internus are ventrally deeply embedded within the pterygoid bone, the internal carotid artery of Dermochelys coriacea (Figs. 12A–12C) seems to have a more dorsally positioned, superficial course with regard to the pterygoid. In particular, the foramen posterius canalis carotici interni and canalis caroticus internus are formed by raised ridges on the dorsal surface of the pterygoid, the lateral of which is the comparatively shallow extension of the crista pterygoidei. The crista pterygoidei and the medial ridge are then roofed by the prootic to form the canalis caroticus internus. The canalis caroticus internus then continues between the pterygoid and prootic for a short distance, before the canal becomes encased by the basisphenoid and pterygoid more anteriorly. These observations differ from Albrecht (1976), who stated that the internal carotid artery is only surrounded by the pterygoid. As in cheloniids, the internal carotid artery joins the sulcus cavernosus and the split into the palatine and cerebral branches is not covered by bone (Nick, 1912; Albrecht, 1976; Evers & Benson, 2019; Figs. 12A and 12D). No canalis caroticus lateralis, canalis caroticus basisphenoidalis, and canalis carotico-pharyngealis are present in our specimens. The absence of an ossified canalis caroticus basisphenoidalis in D. coriacea is unique among extant turtles, although it has been observed in at least one fossil turtle, Sandownia harrisi (Evers & Joyce, 2020). In D. coriacea, osteological correlates for a short cerebral artery sulcus exist nonetheless. Directly anteromedial to the basicranial exit of the canalis caroticus internus, the lateral surface of the basisphenoid shows a weak but broad sulcus ventrally to the vestigially developed clinoid processes. This sulcus is interpreted as an incompletely ossified, anteriorly open canalis caroticus basisphenoidalis, and its broad size indicates that the cerebral artery is relatively large in diameter. Although the path for the cerebral artery was thus not fully enclosed by bone in the specimens we examined, Albrecht (1976) noticed on the specimen he examined that a canalis caroticus basisphenoidalis was present, supporting statements by Nick (1912). D. coriacea is known for its incomplete ossification pattern. For instance, much of the lateral side of the braincase, which, in other turtles, is ossified by a ventrally tall descending process of the parietal, remains cartilaginous in D. coriacea (Nick, 1912), and much of the hyoid skeleton remains entirely cartilaginous (Schumacher, 1973). Therefore, it is possible that completely ossified canalis caroticus basisphenoidalis develop in particularly old and well ossified individuals, so that our observations are not necessarily in contrast to those of Albrecht (1976) or Nick (1912). The canalis stapedio-temporalis is about three times larger than the canalis caroticus internus in both specimens (Table 2).

Figure 12 The carotid circulation and vidian canal system of Dermochelys coriacea (FMNH 171756).

Three-dimensional reconstructions of the basisphenoid, right pterygoid, in (A) dorsal, (B) ventral, and (C) left lateral view. Illustration in dorsal view (D) highlighting the placement of relevant arteries, nerves, and veins. Dark colors highlight sections fully covered by bone, light colors partially or fully uncovered sections. Abbreviations: ac, arteria carotis cerebralis; aci, arteria carotis interna; am, arteria mandibularis; ap, arteria palatina; bs, basisphenoid; cci, canalis caroticus internus; ccv, canalis cavernosus; cnf, canalis nervus facialis; pt, pterygoid; vcl, vena capitis lateralis; VII, nervus facialis; VIIhy, nervus hyomandibularis; VIIvi, nervus vidiani. Note that we do not highlight the course of the mandibular artery, because the mandibular blood supply of Dermochelys coriacea is currently unknown.

Mandibular artery—We are unaware of studies that clarify the source of the mandibular artery in D. coriacea, but also find no clear osteological correlates that would suggest origin from the stapedial artery vs one of the arteries of the internal carotid.

Canalis cavernosus—The canalis cavernosus of Dermochelys coriacea (Figs. 12A–12D) is located lateral to the canalis caroticus internus and connects the subtemporal region of the skull with the cavum acustico-jugulare. The canalis cavernosus is formed by the pterygoid, quadrate, and prootic and is extremely short. The foramen cavernosum is positioned distinctly posterior to the position of the anteriorly unossified trigeminal foramen. A distinct sulcus cavernosus, as developed in most turtles anterior to the foramen cavernosum, is not developed in D. coriacea, but the course of the vena capitis lateralis is still inferred to extend posteriorly along the dorsal surface of the pterygoid based on the position of the foramen cavernosum.

Facial nerve canal system—The facial nerve canal extends ventrolaterally from the fossa acustico-facialis through the prootic (Figs. 12A and 12D). The canalis nervus facialis exits the prootic in a position anterodorsal to the divergence point of the canalis caroticus internus and the canalis cavernosus. The geniculate ganglion is inferred to contact the canalis caroticus internus (Fig. 12D). The canalis pro ramo nervi vidiani is therefore absent and the vidian nerve directly enters the canalis caroticus internus (Fig. 12D). The hyomandibular nerve joins the canalis cavernosus via a very short canalis nervus hyomandibularis proximalis and then leaves no osteological correlate within the posterior section of the canalis cavernosus (Fig. 12A). No canalis nervus vidianus is visible, but Nick (1912) observed the vidian nerve paralleling the course of the internal carotid artery and, once the internal carotid artery has joined the sulcus cavernosus and split into the cerebral and palatine branches, paralleling the course of the palatine artery (Fig. 12D). This provides evidence for our interpretations of the facial nerve system in Dermochelys coriacea.

Platysternidae (Fig. 13)

Canalis caroticus internus—The internal carotid artery enters the skull anterior to the level of the columella auris and through the fenestra postotica. A short, dorsally open sulcus in the pterygoid leads within the cavum acustico-jugulare to the foramen posterius canalis carotici interni (Fig. 13A), which is formed by the pterygoid. The canalis caroticus internus (Figs. 13A and 13B) is formed by the prootic, pterygoid, and basisphenoid. The canalis pro ramo nervi vidiani enters the canalis caroticus internus about mid-length (Fig. 13B). The canalis caroticus lateralis is absent in our specimen, which confirms Albrecht (1976) observations that a foramen anterius canalis carotici lateralis is absent. In contrast, Jamniczky & Russell (2007) identified a canalis caroticus lateralis in their specimen. Following the description of the canal that Jamniczky & Russell (2007) identify as the canalis caroticus lateralis, we conclude that it is the canalis nervus vidianus instead (see below). A canalis carotico-pharyngealis is absent. The canalis caroticus basisphenoidalis, herein defined once the internal carotid artery enters the basisphenoid, is mainly formed by the basisphenoid, with ventral contributions of the pterygoid (Figs. 13A and 13B). The narrowly-spaced foramina anterius canalis carotici basisphenoidalis are formed by the basisphenoid (Figs. 13A and 13B). The canalis stapedio-temporalis is three times larger than the canalis caroticus internus and canalis caroticus basisphenoidalis (Table 2). The relative proportions between the canalis stapedio-temporalis and canalis caroticus basisphenoidalis corroborate the results of Jamniczky & Russell (2007).

Figure 13 The carotid circulation and vidian canal system of Platysternon megacephalum (SMF 69684).

Three-dimensional reconstructions of the basisphenoid, right pterygoid, in (A) dorsal, (B) ventral, and (C) left lateral view. Illustration in dorsal view (D) highlighting the placement of relevant arteries, nerves, and veins. Dark colors highlight sections fully covered by bone, light colors partially or fully uncovered sections. The red portion of the canalis cavernosus shows the inferred position of the mandibular artery. Abbreviations: ac, arteria carotis cerebralis; aci, arteria carotis interna; am, arteria mandibularis; bs, basisphenoid; ccb, canalis caroticus basisphenoidalis; cci, canalis caroticus internus; ccv, canalis cavernosus; cnf, canalis nervus facialis; cnv, canalis nervus vidianus; cprnv, canalis pro ramo nervi vidiani; faccb, foramen anterius canalis carotici basisphenoidalis; facnv, foramen anterius canalis nervi vidiani; fpcci, foramen posterius canalis carotici interni; gg, geniculate ganglion; pt, pterygoid; vcl, vena capitis lateralis; VII, nervus facialis; VIIhy, nervus hyomandibularis; VIIvi, nervus vidiani.

Mandibular artery—The placement of the mandibular artery within the canalis cavernosus is suggested by a faint groove along its lateral side (Figs. 13A–13D), but we are unaware of dissections that unambiguously document this path. Evers & Benson (2019) reported the presence of a foramen arteriomandibulare. We are able to confirm the presence of symmetric openings between the canalis cavernosus and the temporal fossa in our specimen, the same as employed by Evers & Benson (2019), but note that the anterior wall of the canalis cavernosus is extremely thin and that these openings are elongate slits. The study of other specimens should clarify if these indeed represent the exit foramina of the mandibular artery or the incomplete ossification of the anterior wall of the canalis cavernosus in this particular specimen.

Canalis cavernosus—The canalis cavernosus (Figs. 13A–13D) extends from the foramen cavernosum to the level of the foramen stapedio-temporale and is bordered by the quadrate, prootic, and pterygoid. The lateral wall of the canalis cavernosus is incompletely ossified in our specimen, exposing the canal along the lateral pterygoid-prootic-quadrate surfaces toward the subtemporal fossa. The foramen cavernosum is formed by the prootic and pterygoid.

Facial nerve canal system—The canalis nervus facialis extends laterally from the fossa acustico-facialis to the canalis cavernosus through the prootic (Fig. 13A). The position of the geniculate ganglion is inferred to be within the canalis cavernosus, where it splits into the hyomandibular and vidian nerves (Figs. 13A and 13D). The hyomandibular nerve has no osteological correlate within the canalis cavernosus (Figs. 13A and 13D). The vidian nerve enters the canalis pro ramo nervi vidiani, extends ventrolaterally along the prootic-pterygoid suture, and joins the canalis caroticus internus at about its mid-length (Figs. 13B and 13D). Anteriorly, the vidian nerve exits the canalis caroticus internus at the level of the foramen nervi trigemini just posterior to its transformation into the cerebral canal. The vidian nerve extends anteriorly through the pterygoid and leaves the skull on the lateral surface of the pterygoid, anteroventrally to the foramen nervi trigemini (Figs. 13A, 13B and 13D). More anteriorly, ventral to the anterior end of the epipterygoid, the vidian nerve likely re-enters the skull on the lateral surface of the pterygoid, and extends anteriorly through the pterygoid and palatine (Figs. 13A–13D). At its anterior end, the canalis nervus vidianus merges with canals coming from ventral and then splits into two branches (Figs. 13A and 13B). The medial branch joins the dorsal surface of the palatine and the lateral one joins the foramen palatinum posterius. Jamniczky & Russell (2007) identified the canalis nervus vidianus as being the canal for the palatine artery, but its location within the pterygoid, its exit towards the foramen palatinum posterius, the reported likely absence of the palatine artery in this taxon (Albrecht, 1976), the fact it does not join the sulcus cavernosus, and its correspondence with the vidian nerve canal of other testudinoids support our identification instead. The foramina anterius canalis nervi vidiani are formed by the pterygoid and palatine.

Emydidae (Fig. 14)

Canalis caroticus internus—The internal carotid artery enters the skull through the fenestra postotica in Emydoidea blandingii, Emys orbicularis, Deirochelys reticularia, Glyptemys insculpta, Graptemys geographica, Pseudemys floridana, and Terrapene ornata, but in a relatively more ventral position in Clemmys guttata, Glyptemys muhlenbergii, and Terrapene coahuila. The foramen posterius canalis carotici interni (Fig. 14A) is formed by the prootic, pterygoid, and basisphenoid in C. guttata, D. reticularia, Emydoidea blandingii, Gl. insculpta, and T. ornata, by the prootic and pterygoid in Emys orbicularis, Gr. geographica, Gl. muhlenbergii, and T. coahuila, and by the pterygoid and basisphenoid in P. floridana. This differs from Siebenrock (1897) and McDowell (1961) who only identified the pterygoid and prootic as the bones forming the foramen posterius canalis carotici interni in these emydids. The canalis caroticus internus (Fig. 14B) is formed by the prootic, pterygoid, and basisphenoid in all observed emydids but P. floridana, in which the carotid artery extends along the pterygoid-basisphenoid suture for its entire length. According to Jamniczky & Russell (2007), the canalis caroticus internus of Chrysemys picta and Emys orbicularis is formed by the basisphenoid and pterygoid bones only. Siebenrock (1897) also noted for Trachemys ornata that the foramen posterius canalis carotici interni is formed by the pterygoid and the canalis caroticus internus by the pterygoid and basisphenoid. The canalis pro ramo nervi vidiani (Figs. 14A and 14B) joins the internal carotid canal in all emydids but Gl. muhlenbergii, in which the vidian nerve takes a short-cut by directly entering the canalis caroticus lateralis. In all observed emydids, the split of the canalis caroticus internus into the canalis caroticus lateralis and canalis caroticus basisphenoidalis occurs along the pterygoid-basisphenoid suture (Figs. 14A and 14B). The palatine artery extends anteriorly through the canalis caroticus lateralis along the pterygoid-basisphenoid suture and joins the sulcus cavernosus slightly posteriorly to the anterior end of the rostrum basisphenoidale (Figs. 14A and 14B). In Gr. geographica, the anterior portion of the palatine artery shifts laterally and extends through the pterygoid only. The palatine artery is exposed ventrally for a short distance in C. guttata, D. reticularia, Gl. muhlenbergii, T. coahuila, and T. ornata through an unnamed fenestra. Albrecht (1967) noted the presence of a canalis carotico-pharyngealis in Trachemys scripta. Here, we confirm the presence of one canalis carotico-pharyngealis in D. reticularia, Emydoidea blandingii, Emys orbicularis, Gl. insculpta, and P. floridana, and two canalis carotico-pharyngealis in Gr. geographica. In C. guttata, Gl. muhlenbergii, T. coahuila, and T. ornata, no canalis carotico-pharyngealis is visible but this might be due to the ventral exposure of the palatine artery. The cerebral artery extends anteromedially through the basisphenoid and penetrates the sella turcica along widely-spaced foramina anterius canalis carotici basisphenoidalis (Fig. 14A). The canalis stapedio-temporalis is the largest canal and the canalis caroticus lateralis the smallest in all observed emydids, with the canalis stapedio-temporalis being about ten times larger (Table 2). However, some variation occurs regarding the size of the canalis caroticus basisphenoidalis in comparison with the canalis caroticus lateralis. For instance, the canalis caroticus basisphenoidalis of D. reticularia is ten times larger than the canalis caroticus lateralis, whereas the canalis caroticus basisphenoidalis and canalis caroticus lateralis of Gr. geographica have a similar size. The eventual split occurring between the canalis nervus vidianus and canalis caroticus lateralis is likely to change the size of the canalis caroticus lateralis anterior to this split, as in T. ornata or D. reticularia, in which the canalis caroticus lateralis becomes much smaller anterior to the separation with the canalis nervus vidianus. Our results differ from Jamniczky & Russell (2007) who found equally sized canalis caroticus lateralis and canalis caroticus basisphenoidalis in Emys orbicularis.

Figure 14 The carotid circulation and vidian canal system of Glyptemys insculpta (FMNH 22240).

Three-dimensional reconstructions of the basisphenoid, right pterygoid, in (A) dorsal, (B) ventral, and (C) left lateral view. Illustration in dorsal view (D) highlighting the placement of relevant arteries, nerves, and veins. Dark colors highlight sections fully covered by bone, light colors partially or fully uncovered sections. The red portion of the canalis cavernosus shows the inferred position of the mandibular artery. Abbreviations: ac, arteria carotis cerebralis; aci, arteria carotis interna; ap, arteria palatina; am, arteria mandibularis; bs, basisphenoid; ccb, canalis caroticus basisphenoidalis; cci, canalis caroticus internus; ccl, canalis caroticus lateralis; ccv, canalis cavernosus; cnf, canalis nervus facialis; cnv, canalis nervus vidianus; cprnv, canalis pro ramo nervi vidiani; faccb, foramen anterius canalis carotici basisphenoidalis; faccl, foramen anterius canalis caroticus lateralis; facnv, foramen anterius canalis nervi vidiani; fpcci, foramen posterius canalis carotici interni; gg, geniculate ganglion; pt, pterygoid; vcl, vena capitis lateralis; VII, nervus facialis; VIIhy, nervus hyomandibularis; VIIvi, nervus vidiani.

Mandibular artery—The presence of the mandibular artery can be inferred from a sulcus located along the lateral wall of the canalis cavernosus (Albrecht, 1967; Figs. 14A–14D).

Canalis cavernosus—The morphology of the canalis cavernosus (Figs. 14A–14D) is very similar in all observed emydids. The canalis cavernosus extends from the foramen cavernosum to the level of the foramen stapedio-temporale and is formed by the quadrate, prootic and pterygoid. The foramen cavernosum is formed by the prootic and pterygoid.

Facial nerve canal system—The facial nerve extends ventrolaterally from the fossa acustico-facialis to the canalis cavernosus through the prootic (Figs. 14A and 14D). The position of the geniculate ganglion is inferred to be within the canalis cavernosus in all emydids, giving off the hyomandibular and vidian nerves (Figs. 14A and 14D). The hyomandibular nerve has no osteological correlate within the canalis cavernosus in Gl. insculpta (Figs. 14A and 14D), Gl. muhlenbergii, and P. floridana. However, in C. guttata, D. reticularia, Emydoidea blandingii, Emys orbicularis, Gr. geographica, T. coahuila, and T. ornata, the course of the hyomandibular nerve can be seen as a sulcus in the wall of the canalis cavernosus. The vidian nerve of all emydids but Gl. muhlenbergii extends through the canalis pro ramo nervi vidiani to join the canalis caroticus internus (Figs. 14A, 14B and 14D), and is bordered by the prootic and pterygoid, with minor contributions of the basisphenoid in C. guttata, Emys orbicularis, Gl, insculpta, and P. floridana. This slightly differs from Shiino (1913), who did not mention the basisphenoid as forming parts of the canalis pro ramo nervi vidiani in C. guttata. In Gl. muhlenbergii, the vidian nerve does not join the canalis caroticus internus and extends through the prootic and pterygoid to merge anteriorly with the canalis caroticus lateralis. The vidian nerve is then inferred to follow the course of the canalis caroticus lateralis into the sulcus cavernosus. Anterior to the foramen anterius canalis carotici lateralis, at the level of the anterior margin of the descending process of the parietal, an extremely short canal that might contain a portion of the vidian nerve crosses the pterygoid medio-laterally. In C. guttata, Emydoidea blandingii, and Emys orbicularis, the vidian nerve is inferred to pass through the canalis caroticus internus and canalis caroticus lateralis to join the sulcus cavernosus, and anteriorly pierces the pterygoid to enter the canalis nervus vidianus that is formed by the pterygoid and epipterygoid in C. guttata, by the pterygoid, epipterygoid, and parietal in Emydoidea blandingii, and Emys orbicularis. The foramen anterius canalis nervi vidiani is formed by the pterygoid and epipterygoid in C. guttata, by the pterygoid and parietal in Emydoidea blandingii, and by the epipterygoid, palatine, and parietal in Emys orbicularis. D. reticularia exhibits a specific pattern on each side. On the left side, the vidian nerve splits from the canalis caroticus internus posterior to its split into the canalis caroticus lateralis and canalis caroticus basisphenoidalis, and the canalis nervus vidianus is formed by contributions of the prootic, pterygoid, basisphenoid, epipterygoid and parietal. On the right side, the vidian nerve is inferred to follow the course of the canalis caroticus internus and canalis caroticus lateralis, then splits from the canalis caroticus lateralis to extend through its own canal which is formed by the pterygoid, epipterygoid, and parietal. The foramina anterius canalis nervi vidiani are formed by the epipterygoid and parietal, and located ventrolaterally to the anterior margin of the processus inferior parietalis. In Gl. insculpta (Figs. 14A–14D), anterior to the canalis pro ramo nervi vidiani, the vidian nerve remains distinguishable from the canalis caroticus internus for a short distance as a hump located on the lateral side of the latter. The vidian nerve then splits from the canalis caroticus internus, extends anteriorly along the pterygoid-basisphenoid suture, and joins the posterior portion of the canalis caroticus lateralis. Posterior to the foramen anterius canalis carotici lateralis, the vidian nerve diverges from the canalis caroticus lateralis to extend through the pterygoid and exits the skull ventrolaterally to the anterior margin of the processus inferior parietalis. In Gr. geographica, P. floridana, T. coahuila and T. ornata, the vidian nerve is inferred to follow the course of the canalis caroticus internus and canalis caroticus lateralis, then splits from the latter posterior to or at the level of the foramen anterius canalis carotici lateralis, and extends anteriorly through the canalis nervus vidianus. The canalis nervus vidianus and foramen anterius canalis nervi vidiani are formed by the pterygoid and palatine in Gr. geographica, and by the pterygoid and epipterygoid in T. coahuila and T. ornata. The canalis nervus vidianus of P. floridana is formed by the pterygoid and palatine, but the foramen anterius canalis nervi vidiani by the palatine only. Siebenrock (1897) noted that the canalis nervus vidianus of Trachemys ornata is formed by the pterygoid and palatine. Albrecht (1967) described a different pattern for the vidian nerve in Trachemys scripta, Chrysemys picta, and Pseudemys concinna. In these taxa, some branches of the vidian nerve enter a canal that Albrecht (1967) calls “posterior canalis nervi vidiani” and that extends anterolaterally from the canalis pro ramo nervi vidiani close to the canalis carotico-pharyngealis. The other branches of the vidian nerve follow the course of the canalis caroticus internus and canalis caroticus lateralis. Some of these latter branches then join the branches of the “posterior canalis nervi vidiani” via the canalis carotico-pharyngeale, and together, extend anteriorly through his “anterior canalis nervi vidiani” through the pterygoid and palatine. The “anterior canalis nervi vidiani” then opens into the foramen palatinum posterius.

Testudinidae (Fig. 15)

Canalis caroticus internus—In all examined testudinids, the internal carotid artery enters the skull through the fenestra postotica. A sulcus for the internal carotid artery is present on the dorsal side of the pterygoid in all observed specimens but Aldabrachelys gigantea. The foramen posterius canalis carotici interni (Fig. 15A) is formed within the cavum acustico-jugulare by the prootic and pterygoid in Agrionemys horsfieldii, Al. gigantea, Gopherus agassizii, Gopherus polyphemus, Indotestudo elongata, Indotestudo forstenii, Malacochersus tornieri, and Testudo marginata, by the pterygoid in Gopherus flavomarginatus, by the prootic, pterygoid, and basisphenoid in Kinixys erosa, and by the quadrate, prootic, pterygoid, and basisphenoid in Psammobates tentorius. The canalis caroticus internus (Fig. 15B) is formed by the prootic, pterygoid, and basisphenoid in all observed testudinids but P. tentorius, in which the quadrate posterolaterally contributes to it. Jamniczky & Russell (2007) did not mention the prootic as contributing to the canalis caroticus internus of G. polyphemus and Ag. horsfieldii. The canalis caroticus internus contacts the canalis pro ramo nervi vidiani in Ag. horsfieldii Al. gigantea, G. agassizii, G. flavomarginatus, G. polyphemus, I. forstenii, M. tornieri, P. tentorius, but not in I. elongata, K. erosa, and T. marginata. The canalis caroticus lateralis and canalis carotico-pharyngealis are absent in all observed testudinids, with exception of a highly asymmetric specimen of Manouria impressa, which possesses a canalis caroticus lateralis on the right side, but not on the left side. The canal on the right side appears to be a “regular” canalis caroticus lateralis in the sense that it is in the expected position of such a canal in the basisphenoid-pterygoid suture, and that it extends toward an exiting foramen within the sulcus cavernosus. The canal size is similar to that in geoemydids. As this specimen appears to display a congenital abnormality, we disregard it from further consideration, but note that it may be informative in better understanding how the cranial circulation patterns originate during development. Our identification of the only canal that splits from the canalis caroticus internus as being the canalis nervus vidianus (see Figs. 15A and 15B) rests upon three criteria that are encountered in all observed testudinids. First, this canal always extends anteriorly in a relative lateral position through the pterygoid. Second, the canal does not extend along the pterygoid-basisphenoid suture, unlike in all turtles with an unambiguous canalis caroticus lateralis. And third, the canal never connects to the sulcus cavernosus, whereas the canalis caroticus lateralis, when present in other turtles, connects to the sulcus cavernosus via the foramen anterius canalis carotici lateralis along the pterygoid-basisphenoid suture. These criteria also imply that the palatine artery is likely absent in the taxa we observed. However, its presence has been highlighted in several studies. Shindo (1914) noted the presence of a very small palatine artery in Testudo graeca, splitting from the internal carotid artery close to the foramen anterius canalis carotici basisphenoidalis, and extending anteriorly with the vidian nerve through the palatine. McDowell (1961) made a similar statement for the testudinids he dissected, observing a vestigial palatine artery in Gopherus berlandieri and T. graeca that extends through its own canal separate from the vidian nerve. Albrecht (1976) mentioned the palatine artery as being present but vestigial in G. flavomarginatus, G. polyphemus, M. tornieri, and Chelonoidis nigra, but these claims are not based on dissections and he did not provide detailed descriptions about where the palatine canal is located. Jamniczky & Russell (2007) finally stated that the canalis caroticus lateralis and foramen anterius canalis carotici lateralis are absent in G. polyphemus. Several possibilities exist to explain these differences. First, it is possible that the presence of the palatine artery is highly polymorphic, but the systematic absence in our sample makes that somewhat unlikely. Second, it is possible that Shindo (1914) and McDowell (1961) misidentified the vidian nerve as the palatine artery and that the palatine artery is systematically absent. Third, it is possible that the palatine artery of tortoises regularly follows the course of the vidian nerve into the palatine, as hinted by Shindo (1914). Our criteria of homology may therefore have misled us to conclude the palatine artery to be systematically absent. Future studies using wet specimens will be needed to clarify this issue. The cerebral artery extends anteromedially through the basisphenoid in all testudinids to penetrate the sella turcica (Fig. 15A), with exception of G. flavomarginatus, where the artery extends along the pterygoid-basisphenoid suture before entering the basisphenoid. The foramina anterius canalis carotici basisphenoidalis are widely spaced (Fig. 15A). The canalis stapedio-temporalis is overall the largest canal and the canalis caroticus basisphenoidalis the smallest in all observed taxa (Table 2), which contradicts Jamniczky & Russell (2007) who described the canalis stapedio-temporalis in G. polyphemus to be equally sized to the canalis caroticus internus and canalis caroticus basisphenoidalis. However, as in emydids, some variation occurs. The canalis stapedio-temporalis of G. agassizii is more than ten times larger than the canalis caroticus basisphenoidalis, whereas the canalis stapedio-temporalis of I. forstenii is only slightly larger than the canalis caroticus basisphenoidalis. The canalis caroticus basisphenoidalis of I. forstenii is also slightly larger than the canalis caroticus internus, whereas the canalis caroticus basisphenoidalis is always smaller than the canalis caroticus internus in other observed testudinids.

Figure 15 The carotid circulation and vidian canal system of Gopherus agassizii (FMNH 216746).

Three-dimensional reconstructions of the basisphenoid, right pterygoid, in (A) dorsal, (B) ventral, and (C) left lateral view. Illustration in dorsal view (D) highlighting the placement of relevant arteries, nerves, and veins. Dark colors highlight sections fully covered by bone, light colors partially or fully uncovered sections. The red portion of the canalis cavernosus shows the inferred position of the mandibular artery. Abbreviations: ac, arteria carotis cerebralis; aci, arteria carotis interna; am, arteria mandibularis; bs, basisphenoid; ccb, canalis caroticus basisphenoidalis; cci, canalis caroticus internus; ccv, canalis cavernosus; cnf, canalis nervus facialis; cnv, canalis nervus vidianus; faccb, foramen anterius canalis carotici basisphenoidalis; facnv, foramen anterius canalis nervi vidiani; fam, foramen arteriomandibulare; fpcci, foramen posterius canalis carotici interni; gg, geniculate ganglion; pt, pterygoid; vcl, vena capitis lateralis; VII, nervus facialis; VIIhy, nervus hyomandibularis; VIIvi, nervus vidiani.

Mandibular artery—The mandibular artery of testudinoids splits off the stapedial artery within the cavum acustico-jugulare, passes through the canalis cavernosus, and exits the skull through the trigeminal foramen (McDowell, 1961; Albrecht, 1967; Figs. 15A–15D). The passage within the canalis cavernosus is typically visible as a lateral sulcus. McDowell (1961) noted, however, that the mandibular artery exits at an earlier point through a separate foramen that he named the foramen arteriomandibulare. The variable presence of this foramen was utilized by Crumly (1982, 1994) to infer phylogenetic relationships among tortoises. Here, we confirm the presence of the foramen in G. polyphemus and additionally report it for G. agassizii (Figs. 15A and 15B), G. flavomarginatus, Al. gigantea, and K. erosa. This foramen is formed by the quadrate and pterygoid in Gopherus spp., by the quadrate in Al. gigantea, and by the quadrate, prootic, and pterygoid in K. erosa. The mandibular artery of M. tornieri (on the right side) and T. marginata is not contained in the canalis cavernosus for a short distance between the foramen stapedio-temporale and the foramen cavernosum and likely extends through its own canal. In Ag. horsfieldii, I. forstenii, I. elongata, and P. tentorius, no separate mandibular artery foramen is evident, indicating that the arteria mandibularis passes anteriorly through the canalis cavernosus and then exits the skull through the trigeminal foramen.

Canalis cavernosus—The canalis cavernosus (Figs. 15A–15D) extends from the foramen cavernosum to the level of the foramen stapedio-temporale. In M. tornieri, the lateral wall of the canalis cavernosus is partially unossified, exposing it towards the subtemporal fossa. The canalis cavernosus of testudinids is otherwise formed by the quadrate, prootic, and pterygoid. The foramen cavernosum is formed by the prootic and pterygoid.

Facial nerve canal system—The facial nerve extends laterally through the prootic from the fossa acustico-facialis to the canalis cavernosus in Ag. horsfieldii, G. agassizii (Figs. 15A and 15D), G. flavomarginatus, G. polyphemus, K. erosa, M. tornieri, and T. marginata, and the geniculate ganglion is inferred to be within the canalis cavernosus. The geniculate ganglion is inferred to be located medially to the canalis cavernosus and dorsally to the canalis caroticus internus in Al. gigantea, I. elongata, and I. forstenii. P. tentorius exhibits asymmetry, as the geniculate ganglion is located in between the canalis cavernosus and canalis caroticus internus on the right side, but contacts the canalis caroticus internus on the left side. The geniculate ganglion branches into the hyomandibular and vidian nerves. The hyomandibular nerve has no osteological correlate within the canalis cavernosus in G. flavomarginatus and I. elongata, but its course can be seen as a sulcus in the wall of the canalis cavernosus in Ag. horsfieldii, G. agassizii (Fig. 15A), I. forstenii, K. erosa, M. tornieri, P. tentorius, and T. marginata, and it extends through its own canalis nervus hyomandibularis distalis in Al. gigantea and G. polyphemus. In all observed testudinids but K. erosa, and T. marginata, the vidian nerve extends through the canalis pro ramo nervi vidiani to connect with the canalis caroticus internus, and leaves the latter more anteriorly to extend through the canalis nervus vidianus (Figs. 15A, 15B and 15D). The canalis pro ramo nervi vidiani is formed by the prootic and pterygoid in Ag. horsfieldii, Al.gigantea, G. agassizii, G. flavomarginatus, G. polyphemus, I. elongata, and I. forstenii, but only by the prootic in M. tornieri and P. tentorius. The vidian nerve of K. erosa and T. marginata extends ventromedially from the geniculate ganglion but does not join the canalis caroticus internus and instead directly enters the canalis nervus vidianus. Additional variation exists regarding the bones forming the canalis nervus vidianus and foramina anterius canalis nervi vidiani. The canalis nervus vidianus is formed by the prootic, pterygoid, and palatine in Ag. horsfieldii, G. polyphemus, and M. tornieri, by the pterygoid, epipterygoid, and palatine in Al. gigantea, by the pterygoid and palatine in G. agassizii, I. elongata, and P. tentorius, by the pterygoid, epipterygoid, palatine, and parietal in G. flavomarginatus, by the pterygoid in I. forstenii, by the prootic, pterygoid, palatine, and parietal in K. erosa, and by the prootic, pterygoid, epipterygoid, palatine, and parietal in T. marginata. In G. agassizii, the vidian nerve exits the skull anteroventrally to the foramen nervi trigemini for a short distance, before re-entering it by piercing the lateral surface of the palatine. The foramina anterius canalis nervi vidiani are formed by the pterygoid and palatine in Ag. horsfieldii, M. tornieri, and P. tentorius, by the epipterygoid and palatine in Al. gigantea, by the palatine in G. agassizii, G. polyphemus, and I. elongata, by the palatine and parietal in G. flavomarginatus and K. erosa, by the pterygoid in I. forstenii, and by the epipterygoid, palatine, and parietal in T. marginata. Gaupp (1888) noted that the canalis nervus vidianus of Testudo graeca is formed by the basisphenoid and pterygoid.

Geoemydidae (Fig. 16)

Canalis caroticus internus—In all observed geoemydids, the internal carotid artery enters the skull through the fenestra postotica. A sulcus for the internal carotid artery is formed on the dorsal side of the pterygoid. The foramen posterius canalis carotici interni (Fig. 16A) is located within the cavum acustico-jugulare and formed by the prootic and the pterygoid in all taxa in our sample but Geoclemys hamiltonii, where it is only formed by the pterygoid. This slightly differs from Siebenrock (1897) and McDowell (1961) who only identified the pterygoid as forming the foramen posterius canalis carotici interni in Cyclemys dentata, Geoclemys hamiltonii, and Morenia sp. McDowell (1961) also noted that the opisthotic contributes to the foramen posterius canalis carotici interni of Batagur sp., which is not the case in the specimen of Batagur baska we studied. The canalis caroticus internus (Fig. 16B) is formed by the prootic, pterygoid, and basisphenoid in all observed geoemydids but Geoclemys hamiltonii, in which the canalis caroticus internus is formed by the pterygoid and basisphenoid only. Jamniczky & Russell (2007) noted that the canalis caroticus internus of Cuora amboinensis and Rhinoclemmys pulcherrima is formed by the basisphenoid and pterygoid. The canalis pro ramo nervi vidiani enters the canalis caroticus internus in all geoemydids (as illustrated on Fig. 16A for B. baska) but Cyclemys dentata, Geoemyda spengleri, and Rhinoclemmys melanosterna. The canalis caroticus internus splits into the canalis caroticus lateralis and canalis caroticus basisphenoidalis in all sampled taxa (as illustrated on Figs. 16A and 16B for B. baska) but Malayemys subtrijuga, Pangshura tecta, and R. melanosterna, where the lateral carotid canal appears to be absent. Jamniczky & Russell (2007) noted the absence of the canalis caroticus lateralis and foramen anterius canalis carotici lateralis in Cu. amboinensis and R. pulcherrima. The canalis caroticus lateralis extends anteriorly along the pterygoid-basisphenoid suture in B. baska (Figs. 16A and 16B), Cu. amboinensis, Cy. dentata, Geoclemys hamiltonii, Geoemyda spengleri, Mauremys leprosa, and Mo. ocellata, and the foramen anterius canalis carotici lateralis is formed by the pterygoid and basisphenoid in these taxa. In S. crassicollis, the canalis caroticus lateralis and foramen anterius canalis carotici lateralis are formed by the pterygoid only. The canalis carotico-pharyngealis is absent in all observed specimens but Cu. amboinensis, in which one canalis carotico-pharyngealis is present. The canalis caroticus basisphenoidalis and foramen anterius canalis carotici basisphenoidalis are formed by the basisphenoid in all geoemydids (see Figs. 16A and 16B for B. baska) but S. crassicollis, in which the pterygoid posteriorly contributes to the canalis caroticus basisphenoidalis. The foramina anterius canalis carotici basisphenoidalis are widely separated (Fig. 16A). In all observed geoemydids, the canalis stapedio-temporalis is the largest canal and the canalis caroticus lateralis, when present, the smallest (Table 2). As in emydids and testudinids, some variation regarding canal size occurs. The canalis stapedio-temporalis is generally nearly ten times larger than the canalis caroticus lateralis, if present, but this difference is more important in Cy. dentata and Geoemyda spengleri, in which the canalis caroticus lateralis is greatly reduced, and in Malayemys subtrijuga, P. tecta, and R. melanosterna, in which the canalis caroticus lateralis is absent. Moreover, the canalis stapedio-temporalis is about five times larger than the canalis caroticus basisphenoidalis in B. baska and Cu. amboinensis, but only twice as large as the canalis caroticus basisphenoidalis in Malayemys subtrijuga and R. melanosterna, as noted by Jamniczky & Russell (2007) in Rhinoclemmys pulcherrima.

Figure 16 The carotid circulation and vidian canal system of Batagur baska (NHMUK 67.9.28.7).

Three-dimensional reconstructions of the basisphenoid, right pterygoid, in (A) dorsal, (B) ventral, and (C) left lateral view. Illustration in dorsal view (D) highlighting the placement of relevant arteries, nerves, and veins. Dark colors highlight sections fully covered by bone, light colors partially or fully uncovered sections. The red portion of the canalis cavernosus shows the inferred position of the mandibular artery. Abbreviations: ac, arteria carotis cerebralis; aci, arteria carotis interna; ap, arteria palatina; am, arteria mandibularis; bs, basisphenoid; ccb, canalis caroticus basisphenoidalis; cci, canalis caroticus internus; ccl, canalis caroticus lateralis; ccv, canalis cavernosus; cnf, canalis nervus facialis; cnv, canalis nervi vidiani; cprnv, canalis pro ramo nervi vidiani; faccb, foramen anterius canalis caroticus cerebralis; faccl, foramen anterius canalis caroticus lateralis; facnv, foramen anterius canalis nervi vidiani; fpcci, foramen posterius canalis carotici interni; gg, geniculate ganglion; pt, pterygoid; vcl, vena capitis lateralis; VII, nervus facialis; VIIhy, nervus hyomandibularis; VIIvi, nervus vidiani.

Mandibular artery—Siebenrock (1897) described a similar pattern for the mandibular artery of Cy. dentata as in testudinids, but this is not the case in our specimen. However, a distinct protrusion is visible on the lateral wall of the canalis cavernosus of this specimen that likely corresponds to the mandibular artery (Figs. 16A–16D).

Canalis cavernosus—The canalis cavernosus (Figs. 16A–16D) extends from the foramen cavernosum to the level of the foramen stapedio-temporale and is formed by the quadrate and prootic. The foramen cavernosum is formed by the prootic and pterygoid.

Facial nerve canal system—The canalis nervus facialis extends laterally through the prootic from the fossa acustico-facialis to the canalis cavernosus (Fig. 16A). The position of the geniculate ganglion is inferred to be within the canalis cavernosus in all observed geoemydids (see Figs. 16A and 16D for B. baska) but Mauremys leprosa, in which the geniculate ganglion is not in contact with the canalis cavernosus, and R. melanosterna, in which the geniculate ganglion contacts the canalis caroticus internus. The geniculate ganglion splits into the hyomandibular and vidian nerves (Figs. 16A and 16D). The course of the hyomandibular nerve is visible as a sulcus in all observed geoemydids (Figs. 16A and 16D) but Mo. ocellata, in which the hyomandibular nerve has no osteological correlate within the canalis cavernosus and is inferred to follow the course of it. In all geoemydids but Cy. dentata, Geoemyda spengleri, and R. melanosterna, the vidian nerve extends through the canalis pro ramo nervi vidiani to join the canalis caroticus internus (Figs. 16A and 16D). Three patterns for the vidian nerve can be identified for the taxa exhibiting a canalis pro ramo nervi vidiani. In Malayemys subtrijuga and P. tecta, the vidian nerve leaves the canalis caroticus internus anterior to the canalis pro ramo nervi vidiani to enter the canalis nervus vidianus. In Mauremys leprosa and Mo. ocellata, the vidian nerve leaves the canalis caroticus internus, and anteriorly enters the canalis caroticus lateralis to extend into the sulcus cavernosus, and then leaves the latter laterally, anterior to the foramen anterius canalis carotici lateralis. In B. baska (Figs. 16A and 16D), Cu. amboinensis, Geoclemys hamiltonii, and S. crassicollis, the vidian nerve is inferred to extend through the canalis caroticus internus and canalis caroticus lateralis into the sulcus cavernosus, and then enters the canalis nervus vidianus anterior to the foramen anterius canalis carotici lateralis. The vidian nerve of Geoclemys hamiltonii is laterally exposed for a short distance. Cy. dentata and Geoemyda spengleri diverge from these patterns by having a vidian nerve that respectively merges with one and two canals originating from the posterior portion of the canalis caroticus internus. The vidian nerve of Geoemyda spengleri also shortly extends through the canalis caroticus lateralis. As the facial nerve system of R. melanosterna connects to the canalis caroticus internus, the canalis pro ramo nervi vidiani is absent and the vidian nerve directly joins the canalis caroticus internus, follows the course for the internal carotid artery for a short distance, and enters the canalis nervus vidianus. As in testudinids, a substantial amount of variation exists about the bones forming the canalis nervus vidianus and foramina anterius canalis nervi vidiani. The canalis nervus vidianus is formed by the pterygoid and palatine in B. baska and P. tecta, by the pterygoid, palatine, and parietal in Cu. amboinensis, Geoclemys hamiltonii, Malayemys subtrijuga, Mo. ocellata, and S. crassicollis, by the prootic, pterygoid, and parietal in Cy. dentata, by the prootic and pterygoid in Geoemyda spengleri, by the pterygoid, epipterygoid, and parietal in Mauremys leprosa, and by the pterygoid, epipterygoid, palatine, and parietal in R. melanosterna. The foramina anterius canalis nervi vidiani are formed by the palatine in B. baska, Mo. ocellata, P. tecta, and R. melanosterna, by the pterygoid and parietal in Cu. amboinensis, by the parietal in Cy. dentata and Mauremys leprosa, by the palatine and parietal in Geoclemys hamiltonii, Malayemys subtrijuga, and S. crassicollis, and by the pterygoid in Geoemyda spengleri.

Discussion

Summary of carotid canal system

Internal carotid artery—Almost all extant turtles have an internal carotid artery that is fully enclosed in a canalis caroticus internus that extends anteriorly into the basicranium. The sole exception is podocnemidids, where the internal carotid artery passes through a large opening, the cavum pterygoidei, which is unique to that clade. The highly derived placement of podocnemidids within Testudines strongly suggests that this is a secondary modification. We note here, that it is a matter of semantic preference if the foramen posterius canalis carotici interni is thought to be present in podocnemidids, but modified (sensu Gaffney, Tong & Meylan, 2006), or absent due to the presence of a cavum pterygoidei (as defined herein). Our study therefore does not provide novel insights that contradict previous ones. The pterygoid bone is always involved in the formation of the foramen posterius canalis carotici interni in cryptodires but never contributes to this foramen in extant pleurodires (see Table 3 for a summary of the main differences between internal carotid artery systems). This generalization, of course, only applies in the strict sense of the foramen posterius canalis carotici interni employed herein, as the enlarged cavum pterygoidei of podocnemidis has a clear contribution from the pterygoid and has been interpreted as the foramen posterius canalis carotici interni by others (Siebenrock, 1897). Incidentally, a contribution from the pterygoid is apparent in the stem pleurodire Notoemys laticentralis (Lapparent de Broin, De la Fuente & Fernandez, 2007). This may suggest that a pterygoid contribution is plesiomorphic for crown Testudines.

Table 3 Comparative table summarizing the main differences between the internal carotid artery systems.

Taxon	Specimen number	Clade	FPCCI	CCI	CCL	CCB	Origin of the mandibular artery	
Chelodina oblonga	NHMUK 64.12.22	Chelidae	qu + pro	qu + pro + bs	NA	bs	stap. art.	
Chelus fimbriatus	NHMUK 81.9.27.4	Chelidae	pro + bs	pro + bs	NA	bs	stap. art.	
Emydura subglobosa	PIMUZ lab 2009.37	Chelidae	qu + pro + bs	qu + pro + pt + bs	NA	bs	stap. art.	
Hydromedusa tectifera	SMF 70500	Chelidae	pro	pro + bs	NA	bs	stap. art.	
Phrynops geoffroanus	SMF 45470	Chelidae	qu + pro	qu + pro + bs	NA	bs	stap. art.	
Phrynops hilarii	NHMUK 91.3.16.1	Chelidae	pro	pro + bs	NA	bs	stap. art.	
Podocnemis unifilis	NHMUK 60.4.16.9	Podocnemididae	NA	NA	NA	bs	stap. art.	
Podocnemis unifilis	FMNH 45657	Podocnemididae	NA	NA	NA	bs	stap. art.	
Pelomedusa subrufa	NMB 16229	Pelomedusidae	pro + bs	pro + bs	NA	bs	stap. art.	
Pelusios subniger	NMB 16230	Pelomedusidae	pro + bs	pro + bs	NA	bs	stap. art.	
Carettochelys insculpta	NHMUK 1903.7.10.1	Carettochelyidae	pt	pt + bs	NA	bs	stap. art.	
Carettochelys insculpta	SMF 56626	Carettochelyidae	pt	pt + bs	NA	bs	stap. art.	
Amyda cartilaginea	FMNH 244117	Trionychidae	pt	pro + pt + bs	pt + bs	bs	int. car. art	
Apalone mutica	PCHP 2746	Trionychidae	pt	pro + pt + bs	pt + bs	bs	int. car. art	
Apalone spinifera emoryi	FMNH 22178	Trionychidae	pt	pro + pt + bs	pt + bs	bs	int. car. art	
Chitra indica	NHMUK 1926.12.16.1	Trionychidae	pt	pro + pt + bs	pt + bs	bs	int. car. art	
Cyclanorbis senegalensis	NHMUK 65.5.9.21	Trionychidae	pt	pro + pt + bs	pt	bs	int. car. art	
Cycloderma frenatum	NHMUK 84.2.4.1	Trionychidae	pt	pt + bs	pt	bs	int. car. art	
Lissemys punctata	SMF 74141	Trionychidae	pt	pt + bs	pt	bs	int. car. art	
Pelodiscus sinensis	IW576-2b	Trionychidae	pt	pro + pt + bs	pt + bs	bs	int. car. art	
Kinosternon baurii	FMNH 211705	Kinosternoidea	pro + pt	pro + pt + bs	pt + bs	bs	pal. art.	
Kinosternon scorpioides	SMF 71893	Kinosternoidea	pro + pt	pro + pt + bs	pt	bs	pal. art.	
Kinosternon subrubrum hippocrepis	FMNH 211711	Kinosternoidea	pro + pt	pro + pt + bs	pro + pt + bs	bs	pal. art.	
Staurotypus salvinii	NHMUK 1879.1.7.5	Kinosternoidea	pro + pt	pro + pt + bs	pro + pt + bs	bs	pal. art.	
Sternotherus minor	FMNH 211696	Kinosternoidea	pro + pt	pro + pt + bs	pro + pt + bs	bs	pal. art.	
Dermatemys mawii	SMF 59463	Kinosternoidea	pro + pt	pro + pt + bs	pro + pt + bs	bs	stap./pal. art.	
Chelydra serpentina	SMF 32846	Chelydridae	pro + pt	pro + pt + bs	pt + bs	bs	stap. art.	
Macrochelys temminckii	FMNH 22111	Chelydridae	pt	pro + pt + bs	pt + bs	bs	stap. art.	
Caretta caretta	NHMUK 1938.1.9.1	Cheloniidae	pt	pt	NA	bs	stap./pal. art.	
Caretta caretta	NHMUK 1940.3.15.1	Cheloniidae	pt	pt	NA	bs	stap./pal. art.	
Chelonia mydas	NHMUK 1969.776	Cheloniidae	pt	pt + bs	NA	bs	stap./pal. art.	
Eretmochelys imbricata	FMNH 22242	Cheloniidae	pt	pt + bs	NA	bs	stap./pal. art.	
Lepidochelys olivacea	SMNS 11070	Cheloniidae	pt	pt	NA	bs	stap./pal. art.	
Natator depressus	R112123	Cheloniidae	pt	pt + bs	NA	bs	stap./pal. art.	
Dermochelys coriacea	FMNH 171756	Dermochelyidae	pro + pt	pro + pt + bs	NA	NA	unknown	
Dermochelys coriacea	UMZC R3031	Dermochelyidae	pro + pt	pro + pt + bs	NA	NA	unknown	
Platysternon megacephalum	SMF 69684	Emysternia	pt	pro + pt + bs	NA	pt + bs	stap. art.	
Clemmys guttata	FMNH 22114	Emysternia	pro + pt + bs	pro + pt + bs	pt + bs	bs	stap. art.	
Deirochelys reticularia	FMNH 98754	Emysternia	pro + pt + bs	pro + pt + bs	pt + bs	bs	stap. art.	
Emydoidea blandingii	FMNH 22144	Emysternia	pro + pt + bs	pro + pt + bs	pt + bs	bs	stap. art.	
Emys orbicularis	SMF 1987	Emysternia	pro + pt	pro + pt + bs	pt + bs	bs	stap. art.	
Glyptemys insculpta	FMNH 22240	Emysternia	pro + pt + bs	pro + pt + bs	pt + bs	bs	stap. art.	
Glyptemys muhlenbergii	UF 85274	Emysternia	pro + pt	pro + pt + bs	pt + bs	bs	stap. art.	
Graptemys geographica	NHMUK 55.12.6.11	Emysternia	pro + pt	pro + pt + bs	pt + bs	bs	stap. art.	
Pseudemys floridana	FMNH 8222	Emysternia	pt + bs	pt + bs	pt + bs	bs	stap. art.	
Terrapene coahuila	FMNH 47372	Emysternia	pro + pt	pro + pt + bs	pt + bs	bs	stap. art.	
Terrapene ornata	FMNH 23014	Emysternia	pro + pt + bs	pro + pt + bs	pt + bs	bs	stap. art.	
Agrionemys horsfieldii	PCHP 2929	Testudinidae	pro + pt	pro + pt + bs	NA	bs	stap. art.	
Aldabrachelys gigantea	NHMUK 77.11.12.2	Testudinidae	pro + pt	pro + pt + bs	NA	bs	stap. art.	
Gopherus agassizii	FMNH 216746	Testudinidae	pro + pt	pro + pt + bs	NA	bs	stap. art.	
Gopherus flavomarginatus	FMNH 98916	Testudinidae	pt	pro + pt + bs	NA	pt + bs	stap. art.	
Gopherus polyphemus	FMNH 211815	Testudinidae	pro + pt	pro + pt + bs	NA	bs	stap. art.	
Indotestudo elongata	SMF 71585	Testudinidae	pro + pt	pro + pt + bs	NA	bs	stap. art.	
Indotestudo forstenii	SMF 73257	Testudinidae	pro + pt	pro + pt + bs	NA	bs	stap. art.	
Kinixys erosa	SMF 40166	Testudinidae	pro + pt + bs	pro + pt + bs	NA	bs	stap. art.	
Malacochersus tornieri	SMF 58702	Testudinidae	pro + pt	pro + pt + bs	NA	bs	stap. art.	
Psammobates tentorius verroxii	SMF 57142	Testudinidae	qu + pro + pt + bs	qu + pro + pt + bs	NA	bs	stap. art.	
Testudo marginata	FMNH 51672	Testudinidae	pro + pt	pro + pt + bs	NA	bs	stap. art.	
Batagur baska	NHMUK 67.9.28.7	Geoemydidae	pro + pt	pro + pt + bs	pt + bs	bs	stap. art.	
Cuora amboinensis	NHMUK 69.42.145	Geoemydidae	pro + pt	pro + pt + bs	pt + bs	bs	stap. art.	
Cyclemys dentata	NHMUK 97.11.22.3	Geoemydidae	pro + pt	pro + pt + bs	pt + bs	bs	stap. art.	
Geoclemys hamiltonii	NHMUK 87.9.30.1	Geoemydidae	pt	pt + bs	pt + bs	bs	stap. art.	
Geoemyda spengleri	FMNH 260381	Geoemydidae	pro + pt	pro + pt + bs	pt + bs	bs	stap. art.	
Malayemys subtrijuga	NHMUK 1920.1.20.2545	Geoemydidae	pro + pt	pro + pt + bs	NA	bs	stap. art.	
Mauremys leprosa	NHMUK unnumbered	Geoemydidae	pro + pt	pro + pt + bs	pt + bs	bs	stap. art.	
Morenia ocellata	NHMUK 87.3.11.7	Geoemydidae	pro + pt	pro + pt + bs	pt + bs	bs	stap. art.	
Pangshura tecta	NHMUK 1889.2.6.1	Geoemydidae	pro + pt	pro + pt + bs	NA	bs	stap. art.	
Rhinoclemmys melanosterna	FMNH 44446	Geoemydidae	pro + pt	pro + pt + bs	NA	bs	stap. art.	
Siebenrockiella crassicollis	NHMUK 1864.9.2.47	Geoemydidae	pro + pt	pro + pt + bs	pt	pt + bs	stap. art.	
Note:

Abbreviations: bs, basisphenoid; CCB, canalis caroticus basisphenoidalis; CCI, canalis caroticus internus; CCL, canalis caroticus lateralis; FPCCI, foramen posterius canalis carotici interni; int. car. art., internal carotid artery; pal. art., palatine artery; pro, prootic; pt, pterygoid; stap. art., stapedial artery; qu, quadrate. Note that NA (non-applicable) is used for the canalis caroticus lateralis either when the palatine artery is absent (pleurodires, Carettochelys insculpta, Platysternon megacephalum, testudinids, and some geoemydids), or because the palatine artery is not embedded in bone (chelonioids), and for the canalis caroticus basisphenoidalis as it is not fully ossified in Dermochelys coriacea. Columns of canals and foramina indicate bones involved in their formation.

Cerebral artery—Little variation exists for the cerebral artery in extant turtles outside of Chelonioidea. All examined taxa have a canalis caroticus basisphenoidalis that extends through the basisphenoid and exits in the sella turcica. The only exception to this is found in Dermochelys coriacea, in which the cerebral artery might not be fully enclosed by bone, a feature that has otherwise only been reported in the fossil turtle Sandownia harrisi (Evers & Joyce, 2020). The exiting foramina for the cerebral artery, the foramina anterius canalis carotici basisphenoidalis, are paired across the skull midline in nearly all species of extant turtles. Only in three species of cheloniids (Lepidochelys olivacea, Caretta caretta, Natator depressus), the left and right canalis caroticus basisphenoidalis merge within the basisphenoid and exit the sella turcica via a single, median foramen (see also Hooks, 1998; Evers, Barrett & Benson, 2019 for fossil occurrences of this feature). Otherwise, variation regarding the cerebral circulation is limited to the spacing of the foramina anterius canalis carotici basisphenoidalis (Hirayama, 1998). In our sample, the foramina anterius canalis carotici basisphenoidalis are widely-spaced in all turtles but Macrochelys temminckii, platysternids, and the remaining chelonioids, in which these openings are rather narrowly-spaced.

The repeated loss of the lateral carotid canal—We observe the complete loss of the lateral carotid canal in all pleurodires (contra Albrecht, 1967, 1976; Gaffney, Tong & Meylan, 2006; Hermanson et al., 2020), carettochelyids (see also Joyce, Volpato & Rollot, 2018), platysternids, testudinids, and some geoemydids, in all cases due to the absence of a palatine artery. A canalis caroticus lateralis is furthermore absent in chelonioids, but not a result of the absence of the palatine artery, but due to its anterior displacement (see also Zangerl, 1953; Albrecht, 1976; Gaffney, 1979; Evers & Benson, 2019). As these conclusions are perhaps the most novel conclusion of this study, we briefly repeat out rationales, while outlining difficulties for paleontologists assessing fossil material.

The special embedding of the internal carotid artery within the cavum pterygoidei in podocnemidids makes it difficult to evaluate the possible presence of a lateral carotid canal in this group, as no actual canal of the size of a blood vessel is developed between the cavum pterygoidei and sulcus cavernosus. Previous authors, such as Gaffney, Tong & Meylan (2006) or Hermanson et al. (2020), interpreted the fenestra between the cavum pterygoidei and sulcus cavernosus to be a trough for the palatine artery. By contrast, we hypothesize the universal absence of a palatine artery in podocnemidids based on the dissection study of Albrecht (1976), who only found a vestigial blood vessel to pass this fenestra in one out of six specimens. The podocnemidid “foramen anterius canalis carotici lateralis” of previous authors is therefore an unnamed fenestra of unclear function. Our re-interpretation, incidentally, is more parsimonious than the interpretation of Hermanson et al. (2020), who inferred a loss of the palatine artery at the base of Pelomedusoides, its reappearance in Podocnemidoidae, and yet another loss for stereogenyine podocnemidids. Our model suggests the persistent absence of a palatine artery in all pelomedusoids, indeed, all pleurodires.

The interpretation of chelids based on osteological material is difficult as well. Albrecht (1976) concluded that a lateral carotid canal is present in all chelids but Chelus fimbriatus, but noted at the same time that this canal does not hold a palatine artery, but rather the vidian nerve. If terminology is to reflect homology, chelids possess a vidian canal, not a lateral carotid canal. Although the said canal subtly reveals its identity as pertaining to the vidian nerve by connecting the geniculate ganglion to the sulcus cavernosus, it suspiciously resembles the lateral carotid canal of other turtles by approximating the pterygoid-basisphenoid suture. It is therefore understandable that this unusually located vidian canal has been misidentified as a lateral carotid canal in the literature (e.g., Hermanson et al., 2020; figured for Hydromedusa tectifera). As a result, the absence of the lateral carotid canal, which is ultimately based on the loss of the palatine artery, is a likely synapomorphy of Pleurodira.

Within Trionychia, we infer the independent loss of the palatine artery for carettochelyids (see also Joyce, Volpato & Rollot, 2018). Interestingly, the artery that extends through the short canalis caroticus lateralis in trionychids does not serve the function of a “regular” palatine artery, but instead supplies the mandible (Albrecht, 1967). Thus, the lateral carotid canal of trionychids, although present according to our observations, fulfills a different role and complicates assessment on its homology. We provide our reasoning for concluding that the mandibular artery of trionychids is a repurposed palatine artery below (see discussion of “Mandibular Artery”).

In addition to the definite reductions in pleurodires and carettochelyids, the canalis caroticus lateralis is lost several times within testudinoids, which generally show the largest amount of variation. As the canalis caroticus lateralis is absent in Platysternon megacephalum, but present in all emydids, there seems to have been one independent loss within Emysternia. Within geoemydids, the canalis caroticus lateralis is generally present, with exception of Malayemys subtrijuga, Pangshura tecta, and Rhinoclemmys melanosterna, which represent several independent losses of the palatine artery based on current geoemydid in-group relationships (Garbin, Ascarrunz & Joyce, 2018). The canalis caroticus lateralis is absent in all testudinids we examined, with the exception of a highly unusual specimen of Manouria impressa (SMF 69777; Fig. 17), which has an asymmetrical arterial pattern with a “regular” canalis caroticus lateralis on the right side, but none on the left side. The occurrence of the palatine artery had been mentioned by Shindo (1914), McDowell (1961), and Albrecht (1976) for some testudinids. Thus, although the vast majority of testudinid specimens examined herein show no evidence for the presence of the palatine artery, it is possible that this feature is highly polymorphic within testudinids. To address this further, studies focusing on intraspecific variation regarding the palatine artery are necessary, which is beyond the scope of this contribution. We note, however, that the canalis caroticus lateralis is extremely small in diameter whenever it is present (Table 2). In summary, a highly reduced to absent palatine artery appears to be a common feature of all extant testudinoids.

Figure 17 Asymmetry in osteological correlates for the palatine artery in Manouria impressa (SMF 69777).

(A), dorsal view of horizontally cut basicranium for orientation. (B), as A, but zoomed in on details of anterior exiting foramina for the carotid arterial system. (C), cranium in left lateral view, showing position of axial slices shown in D–E. (D), axial CT slice at position of foramina anterius canalis carotici basisphenoidalis. (E), axial CT slice at position of carotid split. Note that canals and foramina for the palatine artery are present on the right side, although the palatine artery is generally absent in testudinids. Abbreviations: bo, blind opening (opens into bone, but does not connect to blood or nervous system); ccb, canalis caroticus basisphenoidalis; faccb, foramen anterius canalis carotici basisphenoidalis; faccl, foramen anterius canalis carotici lateralis; r-ccl, right canalis caroticus lateralis; tcb, trabeculae of cancellous bone (small internal openings not connect to blood or nervous system).

Although the palatine artery is present in extant chelonioids (Albrecht, 1976), the internal carotid artery of these turtles enters the sulcus cavernosus prior to its split into the palatine and cerebral branches, so that the palatine artery is never encased in a bony canal (Zangerl, 1953; Albrecht, 1976; Gaffney, 1979; Evers & Benson, 2019). However, the condition likely evolved independently in dermochelyids and extant cheloniids, as stem-group cheloniids show a regular bifurcation pattern with a regularly enclosed palatine artery (Evers, Barrett & Benson, 2019).

The mandibular artery—The mandible of turtles is typically supplied by a large artery that variously originates from different parts of the carotid arterial system (Albrecht, 1967, 1976). These different origins make it difficult to homologize arteries, but the term “mandibular artery” has historically been used for all variants of mandible-supplying arteries (Gaffney, 1979) and we generally adhere to this practice. However, as the mandible is sometimes supplied by two arteries from two sources, we here follow the convention of Albrecht (1967, 1976) by distinguishing between an anterior and posterior mandibular artery. The mandibular artery is mostly supplied by the stapedial artery in pleurodires, carettochelyids, chelydrids, and testudinoids, by the palatine artery in kinosternids, but by both sources in cheloniids (McDowell, 1961; Albrecht, 1967, 1976). In trionychids, the mandibular artery takes the same proximal course as the palatine artery of other turtles (i.e., via the lateral carotid canal), but then only supplies the mandible and does not send a branch anteriorly to supply the facial region (Albrecht, 1967). This raises the question of whether the “mandibular artery” of trionychids is a modified palatine artery (in which case the palatine artery is present in trionychids), or neomorphic (in which case the palatine artery of trionychids was lost). This distinction is important for the reconstruction of the plesiomorphic condition for both crown Cryptodira and crown Testudines. Currently, unpublished CT scans of stem trionychians available to us (e.g., Basilemys sp.; Adocus sp.) indicate that a “regular” palatine artery was present in stem-trionychians. This suggests that the mandibular artery of trionychids indeed is a palatine artery in which the function of the artery (as in the skull region it supplies with blood) has changed. A similar change has occurred at least a second time, within kinosternids, but in that clade the palatine artery retains its “regular” function and only adds the role of supplying the mandible (Albrecht, 1967).

In turtles in which the mandibular artery branches off the stapedial artery, this branching point can occur either before the stapedial artery enters the canalis stapedio-temporalis (as reported for cryptodires in which the mandible is supplied by a branch of the stapedial artery: Albrecht, 1967; Gaffney, 1979), or after it has left the foramen stapedio-temporale (as reported for pleurodires: Albrecht, 1967, 1976). Among pleurodires, Albrecht (1976) observed that the mandibular artery branches off the stapedial artery and identified two different patterns. In pelomedusids and chelids, with the exception of Chelodina longicollis, a large mandibular artery splits from the stapedial artery after the later has exited the skull through the foramen stapedio-temporale. In Chelodina longicollis and podocnemidids, the stapedial artery has two branches, the external and internal mandibular arteries. The external mandibular artery splits from the stapedial artery posterior to the cranium and extends anteroventrally to supply the tissue directly medial to the mandible. After exiting the skull through the foramen stapedio-temporale, the stapedial artery arches anteriorly over the fenestra subtemporalis and the internal mandibular artery splits from the latter lateral to the foramen nervi trigemini, and courses ventrally to the fossa Meckelii in the mandible. As either course bypasses bone, we are unable to find osteological correlates that would document this pattern in fossil taxa.

When the mandibular artery branches off the stapedial artery within the cavum acustico-jugulare (testudinoids, chelydrids, carettochelyids, the posterior (vestigial) mandibular artery of chelonioids, and, potentially, dermatemydids), it takes an anteriorly directed course through the canalis cavernosus. From there, it usually exits the cranium, either through the trigeminal foramen or a separate foramen arteriomandibulare. The latter exit has been observed directly from dissections for the posterior (vestigial) mandibular artery of the chelonioid Chelonia mydas (Albrecht, 1976) and the mandibular artery of the testudinids Gopherus berlandieri, Stigmochelys pardalis, and Chelonoidis denticulatus (McDowell, 1961).

The presence of the foramen arteriomandibulare should serve as an excellent osteological correlate for the location of the mandibular artery in the canalis cavernosus. We confirm the previously established presence of this foramen in testudinids (McDowell, 1961), but note that Bramble (1971) and Crumly (1982, 1994) found additional variation, which may have relevance for systematics. Among testudinoids, Evers & Benson (2019) reported a foramen arteriomandibulare for Platysternon megacephalum, but we here note by reference to the same specimen, that it is unclear if the slit-like foramina indeed represent a true foramen arteriomandibulare or the incomplete ossification to the anterior wall of the canalis cavernosus. In contrast, we here confirm the exit of the mandibular artery through a separate foramen in Carettochelys insculpta through the presence of a foramen arteriomandibulare in osteological material, as previously reported by Evers & Benson (2019), in combination with personal observations of a stained specimen. This clearly contradicts the speculations of Joyce, Volpato & Rollot (2018) that the mandibular artery of carettochelyids may be supplied by the certebral artery.

Evers & Benson (2019) also reported the presence of a foramen arteriomandibulare in the pelomedusid Pelomedusa subrufa. If correct, this observation would be particularly relevant, as it would indicate a completely different course of the mandibular artery (i.e., through the canalis cavernosus) than usually reported for pleurodires (Albrecht, 1976). Given that dissections by Albrecht (1976) of Pelomedusa subrufa show that the canalis cavernosus houses no artery (contradicting observations by Evers & Benson, 2019), we re-examined the specimen used by Evers & Benson (2019: SMF 70504). The “mandibular artery foramen” of that specimen is highly irregular around its margins, and parallels the anterior portion of the canalis cavernosus, much as in the specimen of Platysternon megacephalum mentioned above. We, therefore, here re-interpret the “foramina” as the incompletely ossified anterior wall of the canalis cavernosus.

Although the presence of a foramen arteriomandibulare is the best direct osteological correlate for the course of the mandibular artery, our segmentations show that the course of this artery through the canalis cavernosus is often indicated by a subtle subdivision of the latter, which can serve as a second, more subtle osteological correlate for the placement of the mandibular artery within the canalis cavernosus. Such a subdivision is present in Carettochelys insculpta (Fig. 6), Dermatemys mawii (Fig. 9), chelydrids (Fig. 10), cheloniids (Fig. 11), Platysternon megacephalum (Fig. 13), emydids (Fig. 14), testudinids (Fig. 15), and geoemydids (Fig. 16). In these clades the dorsolateral part of the canalis cavernosus is separated from the medioventral part by a longitudinal, dorsal constriction between these subsections of the canal, which is visible on models of the canalis cavernosus as a longitudinal sulcus. In coronal slices of CT scans of these taxa, the canalis cavernosus shows a reniform cross-section. This morphology is developed to different degrees. For instance, it is particularly strong in testudinids (Fig. 15) and geoemydids (Fig. 16). However, there is also within-clade variation to the feature. For instance, the subdivision of the canalis cavernosus is relatively subtle in the geoemydid Geoclemys hamiltonii, but well-developed to the point that almost two canals are developed in sections of the skull in the geoemydids Cyclemys dentata and Pangshura tecta. We suggest the presence of this subdivision within the canalis cavernosus to be an osteological correlate for the condition that a mandibular artery extends through the canalis cavernosus. The slight spatial separation within the canalis cavernosus thereby represents the adjacent pathways for the vena capitis lateralis and mandibular artery. Two reasons for hypothesizing the subdivision as an indicator for the mandibular artery can be identified. The presence of the subdivision of the canalis cavernosus coincides with the passage of the mandibular artery through that structure in all taxa for which direct dissection observations have been made. Additionally, the foramen arteriomandibulare, that is, a clear indication that the mandibular artery passes through the canalis cavernosus, is only ever present in taxa which also have the subdivision. Thus, the presence of this subdivision provides important information on some clades for which currently no dissection study is available, such as Dermatemys mawii. Thus, we speculate that the subdivision of the canalis cavernosus can be useful in the future to investigate the course of the mandibular artery in fossil clades.

In summary, several patterns for the mandibular artery can be identified from the information available and summarized above: (i) the mandibular artery branches off the stapedial artery without extending through the canalis cavernosus in pleurodires, and the chelid Chelodina longicollis and podocnemidids have a second mandibular artery branch additionally supplying the mandible. In the second pattern (ii), the mandibular artery branches off the stapedial artery within the cavum acustico-jugulare, follows the course of the canalis cavernosus, and exits the skull through the foramen nervi trigemini or a separate foramen arteriomandibulare. This pattern is observed among cryptodires with the exceptions of kinosternids and trionychids. In kinosternids (iii), the mandibular artery does not branch off the stapedial artery but rather from the palatine artery, anterior to the foramen anterius canalis carotici lateralis. Finally (iv), the mandibular artery of trionychids is interpreted to be a modified palatine artery and extends through the canalis caroticus lateralis.

The facial nerve system of turtles

Gaffney (1979) proposed two main patterns for the course of the facial nerve in pleurodires and cryptodires, respectively. According to this model, pleurodires have the geniculate ganglion located at the junction between the canalis nervus facialis and the canalis caroticus internus, whereas the geniculate ganglion of cryptodires is developed at the point of contact between the canalis nervus facialis and the canalis cavernosus. This hypothesis was extrapolated from data collected by Siebenrock (1897) and Soliman (1964) on a few species of cryptodires and pleurodires. The summary provided by Gaffney (1979) lead to the construction of a phylogenetic character for the passage of the hyomandibular branch of the facial nerve (Gaffney, Meylan & Wyss, 1991). Since, few studies have further scrutinized the proposals of Gaffney (1979) (but see Evers & Benson, 2019). Here, we propose that two main patterns of the facial nerve system can be recognized among extant turtles (Fig. 18). These patterns differ from one another in the relative position of the geniculate ganglion, but, unlike in Gaffney (1979), they do not correspond closely to the phylogenetic distinction of pleurodires and cryptodires (see Table 4 for a summary of the main differences between facial nerve systems).

Figure 18 Schematic overview of patterns pertaining to the split of the facial nerve into its hyomandibular and vidian branches.

Abbreviations: caj, cavum acustico-jugulare; ccv, canalis cavernosus; cci, canalis caroticus internus; ccl, canalis caroticus lateralis; cnf, canalis nervus facialis; cnhp, canalis nervus hyomandibularis proximalis; cnv, canalis nervus vidianus; cprnv, canalis pro ramo nervi vidiani; faf, fossa acustico-facialis; fpcci, foramen posterius canalis carotici interni; gg, geniculate ganglion; VII, nervus facialis; VIIhy, nervus hyomandibularis; VIIvi, nervus vidiani.

Table 4 Comparative table summarizing the main differences between the facial nerve systems.

Taxon	Specimen number	Clade	Geniculate ganglion position	Hyomandibular nerve path	Vidian nerve path	Bones forming the canalis nervus vidianus	
Chelodina oblonga	NHMUK 64.12.22	Chelidae	cci	cnhp + sulcus	cci + cnv	pro + pt + bs	
Chelus fimbriatus	NHMUK 81.9.27.4	Chelidae	cci	cnhp + sulcus	cnv	pro + pt + par	
Emydura subglobosa	PIMUZ lab 2009.37	Chelidae	cci	cnhp	cci + cnv	pro + pt	
Hydromedusa tectifera	SMF 70500	Chelidae	cci	cnhp + sulcus	cci + cnv	pro + pt + bs	
Phrynops geoffroanus	SMF 45470	Chelidae	cci	cnhp + sulcus	cci + cnv	pro + pt	
Phrynops hilarii	NHMUK 91.3.16.1	Chelidae	cci	cnhp + sulcus	cci + cnv	pro + pt	
Podocnemis unifilis	FMNH 45657	Podocnemididae	pro	cnhp + sulcus + cnhd	scv + cnv	pt + pal + par	
Podocnemis unifilis	NHMUK 60.4.16.9	Podocnemididae	pro	cnhp + sulcus + cnhd	cnv	pt + pal + par	
Pelomedusa subrufa	NMB 16229	Pelomedusidae	cci	cnhp + sulcus	cci + cnv	pt	
Pelusios subniger	NMB 16230	Pelomedusidae	cci	cnhp + sulcus	cnv	pro + pt + bs	
Carettochelys insculpta	NHMUK 1903.7.10.1	Carettochelyidae	pro	cnhp + sulcus	cci + cnv	pt + pal	
Carettochelys insculpta	SMF 56626	Carettochelyidae	pro	cnhp + sulcus	cci + cnv	pt + pal	
Amyda cartilaginea	FMNH 244117	Trionychidae	ccv	conf	cci + ccl + scv + cnv	pt + pal	
Apalone mutica	PCHP 2746	Trionychidae	ccv	sulcus	cci + ccl + scv + cnv	pt + pal	
Apalone spinifera emoryi	FMNH 22178	Trionychidae	ccv	sulcus	cci + ccl + scv + cnv	pal	
Chitra indica	NHMUK 1926.12.16.1	Trionychidae	pro	cnhp	cci + ccl + scv + cnv	pal	
Cyclanorbis senegalensis	NHMUK 65.5.9.21	Trionychidae	pro	cnhp	cci + ccl + scv + cnv	pal	
Cycloderma frenatum	NHMUK 84.2.4.1	Trionychidae	pro	cnhp	cci + scv + cnv	pt + pal + par	
Lissemys punctata	SMF 74141	Trionychidae	pro	cnhp	cci + ccl + scv + cnv	pt + pal	
Pelodiscus sinensis	IW576-2b	Trionychidae	ccv	sulcus	cci + ccl + scv + cnv	pal	
Kinosternon baurii	FMNH 211705	Kinosternoidea	ccv	sulcus	cci + ccl + scv + cnv	pt + pal + par	
Kinosternon scorpioides	SMF 71893	Kinosternoidea	ccv	sulcus	cci + ccl + scv + cnv	pt + epi + pal	
Kinosternon subrubrum hippocrepis	FMNH 211711	Kinosternoidea	ccv	sulcus	cci + ccl + scv + cnv	pt + pal	
Staurotypus salvinii	NHMUK 1879.1.7.5	Kinosternoidea	ccv	conf	cci + cnv	pt + pal	
Sternotherus minor	FMNH 211696	Kinosternoidea	ccv	sulcus	cci + ccl + scv + cnv	pt + pal	
Dermatemys mawii	SMF 59463	Kinosternoidea	ccv	conf	cci + ccl + scv + cnv	pal	
Chelydra serpentina	SMF 32846	Chelydridae	ccv	sulcus	cci + cnv	pt	
Macrochelys temminckii	FMNH 22111	Chelydridae	ccv	conf	cci + cnv	pt	
Caretta caretta	NHMUK 1938.1.9.1	Cheloniidae	ccv	conf	cci + scv	NA	
Caretta caretta	NHMUK 1940.3.15.1	Cheloniidae	ccv	conf	cci + scv	NA	
Chelonia mydas	NHMUK 1969.776	Cheloniidae	ccv	sulcus	cci + scv	NA	
Eretmochelys imbricata	FMNH 22242	Cheloniidae	ccv	conf	cci + scv	NA	
Lepidochelys olivacea	SMNS 11070	Cheloniidae	ccv	sulcus	cci + scv	NA	
Natator depressus	R112123	Cheloniidae	ccv	conf	cci + scv	NA	
Dermochelys coriacea	FMNH 171756	Dermochelyidae	cci	cnhp + conf	cci + scv	NA	
Dermochelys coriacea	UMZC R3031	Dermochelyidae	cci	cnhp + conf	cci + scv	NA	
Platysternon megacephalum	SMF 69684	Emysternia	ccv	conf	cci + cnv	pt + pal	
Clemmys guttata	FMNH 22114	Emysternia	ccv	sulcus	cci + ccl + scv + cnv	pt + epi	
Deirochelys reticularia	FMNH 98754	Emysternia	ccv	sulcus	cci + ccl + cnv	pro + pt + bs + epi + par	
Emydoidea blandingii	FMNH 22144	Emysternia	ccv	sulcus	cci + ccl + scv + cnv	pt + epi + par	
Emys orbicularis	SMF 1987	Emysternia	ccv	sulcus	cci + ccl + scv + cnv	pt + epi + par	
Glyptemys insculpta	FMNH 22240	Emysternia	ccv	conf	cci + scc + ccl + cnv	pt	
Glyptemys muhlenbergii	UF 85274	Emysternia	ccv	conf	ccl + scv + cnv	pt	
Graptemys geographica	NHMUK 55.12.6.11	Emysternia	ccv	sulcus	cci + ccl + cnv	pt + pal	
Pseudemys floridana	FMNH 8222	Emysternia	ccv	conf	cci + ccl + cnv	pt + pal	
Terrapene coahuila	FMNH 47372	Emysternia	ccv	sulcus	cci + ccl + cnv	pt + epi	
Terrapene ornata	FMNH 23014	Emysternia	ccv	sulcus	cci + ccl + cnv	pt + epi	
Agrionemys horsfieldii	PCHP 2929	Testudinidae	ccv	sulcus	cci + cnv	pro + pt + pal	
Aldabrachelys gigantea	NHMUK 77.11.12.2	Testudinidae	pro	cnhd	cci + cnv	pt + epi + pal	
Gopherus agassizii	FMNH 216746	Testudinidae	ccv	sulcus	cci + cnv	pt + pal	
Gopherus flavomarginatus	FMNH 98916	Testudinidae	ccv	conf	cci + cnv	pt + epi + pal + par	
Gopherus polyphemus	FMNH 211815	Testudinidae	ccv	cnhd	cci + cnv	pro + pt + pal	
Indotestudo elongata	SMF 71585	Testudinidae	pro	cnhp + conf	cci + cnv	pt + pal	
Indotestudo forstenii	SMF 73257	Testudinidae	pro	sulcus	cci + cnv	pt	
Kinixys erosa	SMF 40166	Testudinidae	ccv	sulcus	cnv	pro + pt + pal + par	
Malacochersus tornieri	SMF 58702	Testudinidae	ccv	sulcus	cci + cnv	pro + pt + pal	
Psammobates tentorius verroxii	SMF 57142	Testudinidae	pro/cci	sulcus	cci + cnv	pt + pal	
Testudo marginata	FMNH 51672	Testudinidae	ccv	sulcus	cnv	pro + pt + epi + pal + par	
Batagur baska	NHMUK 67.9.28.7	Geoemydidae	ccv	sulcus	cci + ccl + scv + cnv	pt + pal	
Cuora amboinensis	NHMUK 69.42.145	Geoemydidae	ccv	sulcus	cci + ccl + scv + cnv	pt + pal + par	
Cyclemys dentata	NHMUK 97.11.22.3	Geoemydidae	ccv	sulcus	cnv	pro + pt + par	
Geoclemys hamiltonii	NHMUK 87.9.30.1	Geoemydidae	ccv	sulcus	cci + ccl + scv + cnv	pt + pal + par	
Geoemyda spengleri	FMNH 260381	Geoemydidae	ccv	sulcus	ccl + cnv	pro + pt	
Malayemys subtrijuga	NHMUK 1920.1.20.2545	Geoemydidae	ccv	sulcus	cci + cnv	pt + pal + par	
Mauremys leprosa	NHMUK unnumbered	Geoemydidae	pro	sulcus	cci + scc + ccl + scv + cnv	pt + epi + par	
Morenia ocellata	NHMUK 87.3.11.7	Geoemydidae	ccv	conf	cci + scc + ccl + scv + cnv	pt + pal + par	
Pangshura tecta	NHMUK 1889.2.6.1	Geoemydidae	ccv	sulcus	cci + cnv	pt + pal	
Rhinoclemmys melanosterna	FMNH 44446	Geoemydidae	cci	sulcus	cci + cnv	pt + epi + pal + par	
Siebenrockiella crassicollis	NHMUK 1864.9.2.47	Geoemydidae	ccv	sulcus	cci + ccl + scv + cnv	pt + pal + par	
Note:

Abbreviations: bs, basisphenoid; cci, canalis caroticus internus; ccl, canalis caroticus lateralis; ccv, canalis cavernosus; cnhd, canalis nervus hyomandibularis distalis; cnhp, canalis nervus hyomandibularis proximalis; cnv, canalis nervus vidianus; conf, hyomandibular nerve confluent with the canalis cavernosus; epi, epipterygoid; pal, palatine; par, parietal; pro, prootic; pt, pterygoid; scc, short-cut canal; scv, sulcus cavernosus; sulcus, hyomandibular nerve contained in a sulcus within the canalis cavernosus. Note that NA (non-applicable) is used when the canalis nervus vidianus is absent (chelonioids).

Our facial pattern I is characterized by a geniculate ganglion positioned within the canalis cavernosus, or at the interface between the canalis nervus facialis and the canalis cavernosus, and thus outside of the prootic (Fig. 18). Facial pattern I essentially corresponds to the “cryptodiran pattern” of Gaffney (1979). However, facial pattern I is actually only realized in a subset of cryptodires, namely in chelydroids (kinosternids, chelydrids, Dermatemys mawii), cheloniids, emysternians (emydids, Platysternon megacephalum), a subset of trionychids (Apalone spinifera, Apalone mutica, Amyda cartilaginea, and Pelodiscus sinensis among our sample), a subset of testudinids (Agrionemys horsfieldii, Gopherus agassizii, Gopherus flavomarginatus, Gopherus polyphemus, Kinixys erosa, Malacochersus tornieri, and Testudo marginata among our sample), and a subset of geoemydids (Batagur baska, Cuora amboinensis, Cyclemys dentata, Geoclemys hamiltonii, Geoemyda spengleri, Malayemys subtrijuga, Morenia ocellata, Pangshura tecta, and Siebenrockiella crassicollis among our sample). Our facial pattern I is further subdivided into three sub-patterns that are distinct in the way the proximal portion of the vidian nerve is transmitted anteriorly. Most turtles with facial pattern I have a canalis pro ramo nervi vidiani which transmits the vidian nerve from within the canalis cavernosus into the canalis caroticus internus, and we designate this condition as facial pattern IA. The facial pattern we name IB is similar in that a canalis pro ramo nervi vidiani is also present, but this canal extends to the canalis caroticus lateralis instead. Thus, the canalis pro ramo nervi vidiani intersects with the carotid canal system slightly more anteriorly than in facial pattern IA. Facial pattern IB is seen in the emydid Glyptemys muhlenbergii and the geoemydids Cyclemys dentata and Geoemyda spengleri. We observed a third pattern, named facial pattern IC, in the testudinids Kinixys erosa and Testudo marginata. These turtles lack a by-passage of the vidian nerve through the canalis caroticus internus or any other canal associated with the carotid arterial system altogether. Instead, the vidian nerve passes directly into the canalis nervus vidianus from the canalis cavernosus.

Our facial pattern II is characterized by a geniculate ganglion position outside the canalis cavernosus and in a more proximal position with regard to the facial nerve stem coming from the brain, and thus within the prootic. All turtles with facial pattern II have a canalis nervus hyomandibularis proximalis, which transmits the hyomandibular nerve from within the prootic to the canalis cavernosus. However, facial pattern II can be subdivided according to the course of the vidian nerve. In turtles with the facial pattern we name IIA, the vidian nerve is contained in a separate canal, the canalis pro ramo nervi vidiani, which extends from the geniculate ganglion to the canalis caroticus internus. The canalis pro ramo nervi vidiani in turtles of facial pattern IIA always extends through the prootic, and thus differs slightly from the canal with the same name in turtles of facial patterns IA and IB, in which the canal diverges from the canalis cavernosus and thus often from within the pterygoid. We retain the same name for all of these canals, as either transmit the vidian nerve toward the carotid arterial system. In facial pattern IIB, the vidian nerve directly enters the canalis caroticus internus and a canalis pro ramo nervi vidiani is absent. The direct transmission of the vidian nerve is enabled by an intersection of the canalis caroticus internus with the canalis nervus facialis. Facial patterns IIB and IIC share the direct intersection of the canalis nervus facialis and the canalis caroticus internus. However, in facial pattern IIC, a separate, anteriorly directed canalis nervus vidianus emerges from the area of the canal intersection. Thus, instead of passing through the carotid canal system, the vidian nerve directly enters its own canal, similar to facial pattern IC. Facial pattern IIA is found in podocnemidids, Carettochelys insculpta, some trionychids (Chitra indica, Cyclanorbis senegalensis, Cycloderma frenatum and Lissemys punctata among our sample), some testudinids (Aldabrachelys gigantea, Indotestudo elongata, and Indotestudo forstenii among our sample), and the geoemydid Mauremys leprosa. Facial pattern IIB, which essentially is the “pleurodiran pattern” of Gaffney (1979), is found in most chelids and Pelomedusa subrufa, but additionally also in the cryptodires Dermochelys coriacea and Rhinoclemmys melanosterna. Facial pattern IIC is found in the chelid Chelus fimbriatus and the pelomedusid Pelusios subniger. It is noteworthy that our facial pattern IIA is basically identical to the facial nerve pattern recognized by Evers & Benson (2019: see character 127.1) for thalassochelydians, which were found to be crownward stem-turtles in that study. Our finding of similar facial nerve patterns in some pleurodires (podocnemidids) and some of the phylogenetically earliest branching cryptodires (some trionychids and Carettochelys insculpta) possibly suggests, that facial pattern IIA could be plesiomorphic for crown-group turtles. However, only the re-examination of a broad set of fossil turtles can test this further.

Two additional sources of variation to the facial nerve system are apparent, but these are unrelated to the facial patterns outlined above. The first kind of variation relates to the posterior (=distal) path of the hyomandibular nerve. In all turtles, the posterior part of the hyomandibular nerve traverses or parallels the canalis cavernosus to reach the cavum acustico-jugulare. However, the posterior course of the nerve may be situated (a) in a separate canal, the canalis nervus hyomandibularis distalis (hyomandibular pattern I); (b) a sulcus in the wall of the canalis cavernosus (hyomandibular pattern II); or (c) entirely within the canalis cavernosus (hyomandibular pattern III) (Fig. 19). This variation is independent of the presence of a canalis nervus hyomandibularis proximalis in turtles with facial pattern II, as the canalis nervus hyomandibularis proximalis of facial pattern II only transmits the proximal part of the hyomandibular nerve toward the canalis cavernosus. This is exemplified by turtles with facial pattern I that have a separate canalis nervus hyomandibularis distalis for the distal part of the nerve (e.g., Aldabrachelys gigantea, Gopherus polyphemus), or turtles with facial patterns IIA (e.g., Gopherus agassizii) or IIB (e.g., Pelomedusa subrufa) that have a hyomandibular sulcus irrespective of their canalis nervus hyomandibularis proximalis.

Figure 19 Schematic overview of patterns pertaining to the posterior portion of the hyomandibular nerve.

The five patterns presented herein only apply to taxa in which the vidian nerve enters the carotid canal system (i.e., turtles with patterns IA, IIA, and IIB). Abbreviations: caj, cavum acustico-jugulare; ccv, canalis cavernosus; cnhd, canalis nervus hyomandibularis distalis; snh, sulcus nervus hyomandibularis; VIIhy, nervus hyomandibularis.

The second source of variation not addressed by our ganglion position patterns comes from the anteriormost (=distal) course of the vidian nerve, which passes through the canalis nervus vidianus in the majority of turtles. The canalis nervus vidianus is present in all turtle clades but cheloniids and dermochelyids, in which the vidian nerve is inferred to extend through the canalis caroticus internus to the sulcus cavernosus, and then, in Dermochelys coriacea at least, to follow the path of the palatine artery (Nick, 1912). However, the path of the vidian nerve is highly variable within and between turtle clades, and several patterns can be identified, depending on whether the vidian nerve extends through the canalis caroticus internus, canalis caroticus lateralis, and sulcus cavernosus, or a combination of those. Those turtles classified in facial patterns IC and IIC have the vidian nerve that passes directly from the geniculate ganglion into the canalis nervus vidianus. In all other turtles, the vidian nerve extends through parts of the canal system that also houses the carotid arterial system. In podocnemidids (facial pattern IIA), the vidian nerve enters the cavum pterygoidei and likely extends close to the internal carotid artery before entering the canalis nervus vidianus. However, as the interaction between the vidian nerve and the internal carotid artery occurs within the cavum pterygoidei and is not documented in the bony skeleton, we cannot determine with certainty what the pattern for the vidian nerve is. In chelonioids, the vidian nerve is interpreted to follow the course of the palatine artery within the carotid arterial system, as no distinct canalis nervus vidianus can be observed. In turtles showing facial pattern IB, the vidian nerve passes directly into the palatine artery canal, from which it either directly enters the canalis nervus vidianus (the geoemydid Geoemyda spengleri), or first exits into the sulcus cavernosus before entering the canalis nervus vidianus (the emydid Glyptemys muhlenbergii). In turtles that exhibit facial patterns IA, IIA, or IIB, the vidian nerve always extends at least partially through the canalis caroticus internus. In these turtles the vidian nerve extends from the canalis caroticus internus to the canalis nervus vidianus via one of five pathways (Fig. 20). In vidian pattern I, the vidian nerve enters a separate canalis nervus vidianus that directly branches off the internal carotid canal (Pelomedusa subrufa, chelids except for Chelus fimbriatus; Carettochelys insculpta, Platysternon megacephalum, all testudinids but those with pattern IC, the trionychid Cycloderma frenatum, the kinosternid Staurotypus salvinii, the chelydrid Chelydra serpentina, and the geoemydids Malayemys subtrijuga, Pangshura tecta, and Rhinoclemmys melanosterna). In vidian pattern II, the vidian nerve parallels the internal carotid artery up to its split, and then follows the palatine artery into its canal, from which a separate canalis nervus vidianus diverges (the emydids Graptemys geographica, Pseudemys floridana, Terrapene coahuila, Terrapene ornata). In vidian pattern III, the vidian nerve also parallels the palatine artery, but exits the respective canal through the foramen anterius canalis carotici lateralis into the sulcus cavernosus, from which a separate canalis nervus vidianus emerges (all trionychids but Cycloderma frenatum, all kinosternids but Staurotypus salvinii, Dermatemys mawii, the emydids Clemmys guttata, Emydoidea blandingii, and Emys orbicularis, and the geoemydids Batagur baska, Cuora amboinensis, Geoclemys hamiltonii, and Siebenrockiella crassicollis). In vidian patterns IV and V, the vidian nerve enters the canalis caroticus lateralis, but not via the arterial pathway, but instead via a “shortcut” canal that connects the canalis caroticus internus and canalis caroticus lateralis. The vidian nerve then enters the canalis nervus vidianus that either branches directly off the canalis caroticus lateralis (vidian pattern IV; similar to vidian pattern II; the emydid Glyptemys insculpta), or emerges from the sulcus cavernosus after the vidian nerve has left the arterial canal via the foramen anterius canalis carotici lateralis (vidian pattern V; similar to vidian pattern III; the geoemydids Mauremys leprosa and Morenia ocellata). The differences between these five vidian patterns are relatively subtle and seem largely driven by the relative anteroposterior origin of the canalis nervus vidianus. This is possibly best exemplified by our examined specimen of the emydid Deirochelys reticularia (FMNH 98754), which exhibits an asymmetric pattern. On the left side of this specimen, the canalis nervus vidianus begins from the canalis caroticus internus (i.e., vidian pattern I), whereas on the right side, the canal splits from the canalis caroticus lateralis further anteriorly (i.e., vidian pattern II). Similarly, in a specimen of Cyclemys dentata (NHMUK 97.11.22.3), the exit of the canalis nervus vidianus is asymmetric, as it extends into the sulcus cavernosus on the left side, but connects to the canalis caroticus lateralis on the right side. Our patterns of the relative position of the canalis nervus vidianus are not easily matched with the phylogenetic relationships of turtles, so that the observed variation between taxa does not seem to be systematic.

Figure 20 Schematic overview of patterns pertaining to the anterior portion of the vidian nerve.

Abbreviations: cci, canalis caroticus internus; ccl, canalis caroticus lateralis; cnv, canalis nervus vidianus; fpcci, foramen posterius canalis carotici interni; scc, “short cut canal”; scv, sulcus cavernosus; VIIvi, vidian nerve.

Inferring canal homology from osteological material

Our study broadly confirms that the canals that penetrate the basicranium of turtles can be identified correctly based on osteological material alone. The greatest source of past error pertains to the correct identity of the lateral canal vs the vidian canal. As outlined throughout this contribution, these two canals can be distinguished clearly by a series of topological criteria. The lateral carotid canal typically splits from the internal carotid near the basisphenoid-pterygoid suture, extends parallel to the basisphenoid-pterygoid suture and sulcus cavernosus, with which it merges. The vidian canal, by contrast, originates from the facial nerve canal at the likely position of the geniculate ganglion, punctures the pterygoid and/or palatine, crosses the path of the sulcus cavernosus, and is directed towards the foramen palatinum posterius. These criteria are sufficient to correctly identify the single canal in all extant turtles, with two notable exceptions: podocnemidis and chelids.

The dissection studies of Albrecht (1967, 1976) highlight that the fenestra located between the cavum pterygoidei and the sulcus cavernosus of podocnemidids does not hold the palatine artery, as this blood vessel is absent. This situation is troubling, as a passage for the palatine artery is available at the right location, but not used. However, the morphology of this potential passage is very unlike for a regular blood or nerve canal, but rather approximates many other poorly defined opening, such as the foramen orbito-nasale, hiatus acusticus, or the fenestra caroticus by lacking clear evidence for the former passage of a structure. Nevertheless, we suggest scoring the lateral canal (or any canal) for fossil podocnemidids as unknown wherever a passage is apparent that lacks the characteristics of a true blood foramen.

In some chelids, a canal exists that connects the internal carotid canal with the sulcus cavernosus, but the dissection studies of Albrecht (1967, 1976) suggest that this is the vidian canal, not the lateral canal. Although this canal often connects with the geniculate ganglion and has the same diameter as a more anterior canal with unambiguous vidian affiliations (see Fig. 3 for Chelus fimbriatus), its passage close to the basisphenoid-pterygoid suture is more characteristic of a lateral canal. As for fossil podocnemidids, we recommend scoring fossil chelids as unknown for the presence of a lateral canal. For the unlikely event that a similar arrangement is found for a fossil group of turtles unrelated to chelids, we suggest scoring the lateral and vidian canals as unknown, whenever true evidence for a vidian canal only is lacking (e.g., a path deep within the pterygoid or continuous connection of the geniculate ganglion with the foramen palatinum posterius).

Canal sizes

The carotid arterial system is the sole source of blood to the cranium and its subordinate arteries supply all major cranial tissues. The majority of cranial tissues are supplied by branches of the internal carotid artery, but some parts are also supplied by the branches of the external carotid artery (Kardong, 1998). Generally, in reptiles, the anterior head region, including the orbit, is supplied by two branches of the common carotid artery: the stapedial artery and at least one major subordinate branch of the medially directed carotid branch that otherwise supplies the brain via the cerebral artery. The subordinate, facially-directed branch of the brain-supplying internal carotid branch is either the orbital artery, which branches off the cerebral artery from within the sella turcica, or the sphenopalatine artery, which branches off the cerebral/internal carotid before the latter enters the basisphenoid (Albrecht, 1967; Porter & Witmer, 2015; Porter, Sedlmayr & Witmer, 2016). The orbital and sphenopalatine arteries can co-exist (e.g., crocodiles; Sedlmayr, 2002; Porter & Witmer, 2015; most turtles: Albrecht, 1967), but one of these arteries usually dominates in terms of arterial size, and thus blood volume transmitted, so that only one “medial” artery is of volumetric importance for the blood supply of the anterior head region. The sphenopalatine artery of turtles is generally called the palatine artery, when present (Albrecht, 1967; Gaffney, 1979). The orbital artery is usually also present in turtles, albeit as a small artery (Albrecht, 1967). One exception are trionychids, in which the orbital artery (i.e., a branch of the cerebral artery that originates in the sella turcica) is exceptionally large and bifurcates anteriorly within the orbital cavity to supply the orbit (Albrecht, 1967). Albrecht (1967) used the neologism “pseudopalatine artery” for the orbital artery of trionychids, although the origination within the sella turcica and from the cerebral artery, as well as its direction toward the orbit are consistent with its identification as the orbital artery, an artery that Albrecht (1967) had also identified in other turtle groups. Thus, the major blood supply for the anterior head region, including the orbit, is achieved by three different arteries in turtles: either by both the palatine artery and the stapedial artery (when both arteries are well developed, e.g., cheloniids: Albrecht, 1976), or predominantly by the stapedial artery (when the palatine artery is absent or small, e.g., Chrysemys: Albrecht, 1967), or predominantly by the palatine artery (when the stapedial artery is reduced or absent, e.g., Sternotherus: Albrecht, 1967), or predominantly by the orbital artery (when the palatine artery is absent and stapedial artery is reduced in size, e.g., trionychids: Albrecht, 1967). Although the origin of the arteries suppling the anterior skull region thus varies across turtles, the terminal patterning of these arteries, such as the development of supra- and infraorbital arteries, are largely consistent (Albrecht, 1967). Variation as to which artery supplies specific organs is also documented for the mandible in turtles: whereas in non-turtle reptiles and many turtles, the mandibular artery originates as a branch of the stapedial artery (Albrecht, 1967, 1976; Gaffney, 1979; Porter & Witmer, 2015), the mandibular artery in some turtles may also branch from the palatine artery when the stapedial artery is reduced (e.g., Sternotherus odoratus: Albrecht, 1967) or the internal carotid artery when the stapedial and palatine arteries are reduced (e.g., trionychids: Albrecht, 1967).

In turtles, both the palatine and stapedial arteries are variably reduced or entirely lost among different clades (e.g., Albrecht, 1967, 1976; Gaffney, 1979; this study). As the above examples show, these arterial reductions affect the patterns of blood supply, as the arteries that supply the anterior head or mandible regions are always present but have different branching points along more proximal parts of the carotid arterial system, depending on which arteries are reduced and which are not. Based on his dissections of a few representatives of most turtle clades, Albrecht (1976) hypothesized that reductions in the size of the stapedial artery are counterbalanced by increases in size of the palatine or orbital arteries to ensure the arterial blood supply for the anterior region of the head. Although relatively many studies related to the cranial arteries or canals are available, and although qualitative observations for various turtle clades seem to provide support for Albrecht (1976) hypothesis (McDowell, 1961), only few publications explicitly compare or quantify canal sizes for a large number of species (Jamniczky & Russell, 2007). Here, we digitally measured the cross-sectional area of the canalis caroticus internus, the canalis caroticus lateralis, the canalis caroticus basisphenoidalis, and the canalis stapedio-temporalis in our CT-scans (see “Methods” for further details). These measurements, summarized in Table 2, provide approximate estimates of blood flow, as arterial canal diameter is proportional to arterial size in turtles (Albrecht, 1976; Jamniczky & Russell, 2004).

Our pGLS analysis finds strong evidence of a correlation between internal carotid artery size and stapedial artery size (Table 5). Variation in cross-sectional area of the stapedial artery canal explains a large portion of variance in the cross-sectional area of the internal carotid artery canal (R2 = 0.73). The regression line has an intercept of 1.097 and a slope of 0.776 (Fig. 21A; Table 5), indicating a moderately negative allometric scaling relationship between medial and lateral arterial sizes. The relationship between cross-sectional areas of stapedial and internal carotid arteries underwent evolutionary change on the tree, as the phylogenetic signal of this relationship as estimated during the fitting of the regression model is high (lambda = 0.96).

Table 5 Results of pGLS regressions of internal carotid artery size on stapedial artery size.

Model	Lambda	Variable	Coefficient	P-Value	R2	
(log10(CCI) ~ log10(CST))alltaxa	0.957	Intercept	1.098	<0.001	0.73	
		Slope	0.776	<0.001		
(log10(CCI) ~ log10(CST))ctk	0.738	Intercept	1.840	<0.001	0.89	
		Slope	0.735	<0.001		
(log10(CCI) ~ log10(CST))remaining	0.400	Intercept	0.892	0.02	0.75	
		Slope	0.764	<0.001		
Note:

CCI stands for cross-sectional area of the internal carotid artery canal. CST stands for cross-sectional area of stapedial artery canal. CTK-abbreviated model describes model only including chelonioids, trionychians, and kinosternoids. Remaining-abbreviated model includes all taxa not included in the CTK-model. Phylogenetic signal (lambda) was estimated during model fitting. R2 is the generalized coefficient of determination described by Nagelkerke (1991).

Figure 21 Relationship of internal carotid artery size and stapedial artery size in turtles.

(A) pGLS regression of log10-cross-sectional diameter of internal carotid artery canal on log10-cross-sectional diameter of stapedial artery canal. Solid grey line is the regression line describing a model with a single slope and intercept for all taxa. Dashed grey lines are regression lines for subsets of the data (see text and Table 3 for details). (B) Plot showing which data points were used as subsets for the multiple regression model test. Green points represent the subset containing kinosternoids, chelonioids, and trionychians, whereas blue points represent the subset containing the remaining turtle clades. (C) Residual plot of pGLS regression of the full dataset, ordered by clades. Symbols in A and C denote clade attributions.

Residual internal carotid artery size is consistently larger than zero in three clades, which are trionychians, kinosternoids, and chelonioids (Fig. 21B and 21C). These clades have larger internal carotid cross-sectional areas than expected by the regression, although this effect is much smaller in chelonioids than in trionychids and kinosternoids, which are visually separated from the remaining data (Fig. 21A), but seem to follow a trend parallel to the remaining data. To test if our data can be better explained by a model with different intercepts or slopes or both for the visually separated groups, we performed generalized least-squares phylogenetic analysis of covariance (pANCOVA) using the gls.ancova function of the package evomap (Smaers & Rohlf, 2016; Smaers & Mongle, 2018). We defined a group with consistently positive residuals (i.e., chelonioids, kinosternoids, trionychians) and another with the remaining taxa for which models could be compared. Results of the pANCOVA show that a model with varying intercepts is significantly supported over a single intercept-single slope model (F value = 14.53, p < 0.0003), and that this model is favored over a model with varying slopes and intercept (F = 3.53, p = 0.06). These results demonstrate that kinosternoids, chelonioids, and trionychians follow a regression line with an elevated intercept in comparison to other turtles (Fig. 21B; Table 5). These taxa have larger internal carotid cross-sectional areas than other turtles (as the intercept is elevated), but show the same proportional size increase (allometric scaling relationship) with increasing stapedial artery size than other turtles (as regression slopes are near identical). These observations quantitatively confirm previous hypotheses, as particularly trionychids and kinosternoids are known for their small stapedial artery canals. However, and somewhat unexpectedly, the same is also true for some taxa with relatively large stapedial arteries, such as carettochelyids or chelonioids, indicating that these taxa have a greater ”medial” blood flow than expected based on their stapedial artery sizes, which are relatively large to begin with. Much of the variation we observe can be attributed to the placement of the mandibular artery, which draws blood from the carotid system, but does not supply the anterior head region. In trionychids and kinosternoids, the mandibular artery is supplied by the lateral branch of the internal carotid artery (Albrecht, 1967, 1976). It is therefore not surprising that the diameter of the internal carotid artery is larger than expected. The high residuals found in Carettochelys insculpta, however, cannot be explained by this model, as our observations indicate that the mandible of this taxon is supplied by the stapedial artery prior to its passage through the stapedial canal. In chelonioids, the mandible is partially supplied by the stapedial artery, but prior to its passing through the stapedial canal, and by the lateral branch of the internal carotid artery (Albrecht, 1976). The internal carotid artery is therefore enlarged again, but not to the extent as seen in trionychids and kinosternids. The opposite situation is seen in chelids, where the mandible is completely supplied by the stapedial artery after its passage through the stapedial canal (Albrecht, 1976). The stapedial canal is therefore disproportionaly enlarged. A similar enlargement, however, is not apparent for the remaining pleurodires, even though mandibles are fully (Pelomedusidae) or partially (Podocnemididae) supplied by the stapedial artery following its passage through the stapedial canal (Albrecht, 1976). In all remaining turtles, the mandible is supplied by the stapedial artery, but prior to its passage through the stapedial canal (Albrecht, 1967, 1976). In summary, our canal size data quantitatively confirm that variation in the stapedial artery size inversely correlates with the internal carotid artery size and that departure from the mean can partially be explained by the configuration of the mandibular artery.

Evolution of the internal carotid arterial system of turtles

Plesiomorphically, the split of the internal carotid artery into its cerebral and palatal branches occurs extracranially (i.e., not embedded in bone), with the palatine branch extending anteriorly through the interpterygoid vacuity, and the cerebral artery extending through the basisphenoid. This condition is known in basal taxa such as Proganochelys quenstedtii and Kayentachelys aprix (Gaffney, 1990; Sterli & Joyce, 2007; Gaffney & Jenkins, 2010). The plesiomorphic presence of the palatine artery, which leaves no direct osteological correlate in these turtles, is inferred based on outgroup comparisons (Müller, Sterli & Anquetin, 2011), and the presence of palatine artery canals in slightly derived stem turtles with closed interpterygoid openings. Examples of those are Kallokibotion bajazidi (Gaffney & Meylan, 1992) and Mongolochelys efremovi (Sterli et al., 2010), in which the split of the internal carotid artery into its cerebral and palatal branches still occurs extracranially, but with the closure of the interpterygoid vacuity, a distinct canalis caroticus lateralis is present. As all extant turtles have the internal carotid artery embedded in bone posterior to its split, this condition could be symplesiomorphic for crown-group turtles and may possibly have evolved somewhere on the stem-lineage of turtles. Examination of fossil representatives of the stem-lineages for the major turtle subclades will provide important clues as to whether the complete embedding of the carotid system in bone is indeed ancestral for the crown-group. The evolutionary reconstruction of the carotid embedding is also hindered by varying phylogenetic hypotheses regarding the position of several turtle clades as stem or crown-group turtles, including paracryptodires, thalassochelydians, xinjiangchelyids, and sinemydids/macrobaenids (Gaffney et al., 2007; Joyce, 2007; Sterli, 2010; Anquetin, 2012; Zhou & Rabi, 2015; Cadena & Parham, 2015; Joyce et al., 2016; Evers & Benson, 2019). Partial embedding of the internal carotid artery is observed in macrobaenids/sinemydids such as Dracochelys bicuspis (Gaffney & Ye, 1992), xinjiangchelyids like Xinjiangchelys wusu (Rabi et al., 2013), or thalassochelydians like Plesiochelys etalloni (NMS 40870). These turtles all share that the posterior part of the internal carotid is embedded in bone, but the anterior part around the area of the split into palatine and cerebral arteries is exposed in a fenestra caroticus (Rabi et al., 2013). If these turtles are evolutionary intermediates between the earliest known turtles and crown-group turtles with completely embedded carotids, as indicated by some phylogenetic analyses (Evers & Benson, 2018), their morphology indicates that the encasing of the carotid artery followed a “posterior-section-first”-pattern. The converse pattern is observed among some paracryptodires: In the baenid Eubaena cephalica, a short anterior section of the internal carotid artery prior to its entry into the basisphenoid (upon which it becomes the cerebral artery), is embedded in bone (Rollot, Lyson & Joyce, 2018), whereas the palatine artery is absent. This could provide evidence for an “anterior-section-first” embedding of the internal carotid artery. However, it is questionable if the baenid condition is truly informative about the evolution of crown-group turtles. Baenids are most frequently inferred to be deeply nested within Paracryptodira (Lyson & Joyce, 2011), but pleurosternids and some of the earliest paracryptodires, such as Uluops uluops, have carotid patterns quite distinct from those of baenids (Anquetin & André, 2020; Evers, Rollot & Joyce, 2020). In Uluops uluops, the carotid split seems to be exposed (Y. Rollot, 2019, personal observations, UCM 53971), indicating that no foramen posterius canalis carotici interni is present. The same condition has been reported for Dorsetochelys typocardium (Anquetin & André, 2020). In pleurosternids, the far-anteriorly positioned foramen posterius canalis carotici interni has likely been misidentified, and the visible foramen in the suture of the basisphenoid and pterygoid is for the cerebral artery instead, whereas the palatine artery is likely lost (Y. Rollot, 2019, personal observations, UMZC T1041: Pleurosternon bullockii; Evers, Rollot & Joyce, 2020). Thus, Uluops and pleurosternids possibly have entirely uncovered internal carotid arteries and therefore the plesiomorphic condition of Testudinata. Depending on paracryptodiran in-group relationships, the loss of the palatine branch of the internal carotid artery has occurred independently at least one time within paracryptodires (Rollot, Lyson & Joyce, 2018), and several times within both extant lineages.

The use of CT data will probably yield new results about the cranial morphology and circulation system of turtles. Particularly important will be the inclusion of fossil taxa belonging to the stem of lineages of the primary clades within Testudines (i.e., former families or superfamilies such as Trionychia, Geoemydidae, Chelonioidea, etc.), as those will provide new insights into evolutionary changes during the early diversification of turtles.

Conclusions

We here describe the carotid circulation and facial nerve systems of all major extant clades of turtles based on micro-CT scans of 69 specimens representing 65 species. Our main results include reinterpretations of the carotid arterial canal system and the facial nerve system and show that some canals pertaining to the facial nerve system have been misinterpreted in the past as carotid canals. Our observations warrant nomenclatural updates to previous canal definitions, which facilitate the precise description of the observed disparity. We demonstrate that the complete loss of the palatine artery and the respective canalis caroticus lateralis is more widespread among turtles than previously recognized, and happened independently in pleurodires, carettochelyids, platysternids, testudinids, and some geoemydids. Quantitative canal size data show that variation of stapedial vs non-stapedial carotid canal size can largely be attributed to differences in mandibular artery course. We review these differences across turtle groups and add novel observation regarding osteological correlates for the mandibular artery, which help to distinguish mandibular blood supply patterns in cryptodires even in the absence of direct arterial observation. Our data furthermore show unexpected variation with regard to the canal system for the facial nerve, particularly with respect to the position of the geniculate ganglion, as well as the course of the subordinate facial nerve branches, the vidian and hyomandibular nerves through their specific canals. These data show that previously hypothesized distinctions in the facial nerve system of pleurodires and cryptodires cannot be upheld. The carotid artery and facial nerve systems appear to vary independently from each other, although the facial nerve and carotid systems in part share the same canals. The carotid circulation and facial nerve systems have been used as a source of phylogenetic characters in many studies, but our observations provide the basis for the revision of previously phrased characters and the conception of new phylogenetic characters. Important future tasks include the integration of fossil data both from stem-lineages of extant groups to facilitate understanding of the evolution of modern diversity regarding these systems, as well as from stem-turtles to understand the evolution of the derived arterial embedding found in turtles.

We would like to acknowledge the curators and staff of museums that provided us access to specimens used in this study, particularly Loïc Costeur (NMB), Alan Resetar (FMNH), Patrick Campbell (NHMUK), Linda Mogk (SMF), Alexander Kupfer (SMNS), and Jason Head (UMZC). We thank research and technical staff and PIs at the CT scanning facilities we used for this research, namely Tom Davies and Ben Moon (University of Bristol), Farah Ahmed (NHMUK), Zhe-Xi Luo and April Isch Neander (University of Chicago), and Ingmar Werneburg (University of Tübingen). We also extend our gratitude to the contributors and curators of MorphoSource. We finally thank Guilherme Hermanson (University of São Paulo) for making a specimen of Podocnemis expansa available for additional examination. Gabriel Ferreira, Heather Jamniczky, Olivier Rieppel, and an anonymous reviewer are thanked for numerous thoughtful comments that greatly helped improve the quality of this manuscript.

Additional Information and Declarations

Competing Interests

Author Contributions

Data Availability

The authors declare that they have no competing interests.

Yann Rollot performed the experiments, analyzed the data, prepared figures and/or tables, authored or reviewed drafts of the paper, and approved the final draft.

Serjoscha W. Evers analyzed the data, prepared figures and/or tables, authored or reviewed drafts of the paper, and approved the final draft.

Walter G. Joyce conceived and designed the experiments, analyzed the data, prepared figures and/or tables, authored or reviewed drafts of the paper, and approved the final draft.

The following information was supplied regarding data availability:

Data is available on Morphobank, project number: 3732.

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
