# Peer review of "A review of the carotid artery and facial nerve canal systems in extant turtles"

_PeerJ, doi:10.7717/peerj.10475_

## Round 0.1 · original submission · Minor Revisions

The reviewers are very positive about your work (and so am I), but still offer some suggestions for improvements, both in form and substance.

As per PeerJ policies (https://peerj.com/about/policies-and-procedures/#data-materials-sharing), all the raw data (including microCT scan data) have to be made available in a permanent public repository prior to formal acceptance.

Please, together with your unmarked revised manuscript, provide a marked-up copy as well as a document explaining how you have addressed each of the points raised by the reviewers.

·

Basic reporting

no comment

Experimental design

no comment

Validity of the findings

no comment

Additional comments

For the uninitiated reader, this lengthy study is a marathon through turtle diversity (both fossil and extant) and the variation of cranial innervation and vascularization patterns – with possible systematic and phylogenetic implications. Starting with Gaffney’s ground-breaking papers published in the 1970s, a tradition became established in turtle systematics to reconstruct with increasing sophistication – both technologically [CT scanning] as well as with respect to taxon sampling – the variation of cranial (‘bony’) ‘canals’ that transmit arterial systems (most prominently derivatives of the internal carotid) and cranial nerves (most prominently derivatives of the facial nerve), and the possible systematic/phylogenetic implications. One of the central issues of contention here is the pleurodire – cryptodire distinction, as well as the placement of certain ‘stem-turtles’. The whole conceptual frame of the study is evolution from bottom up – along the stem towards the crown. There is no question that the authors are very much on top of the task they set themselves. The manuscript is detailed, well informed, and well written (the occasional unavoidable typo notwithstanding). The references are up-to-date and exhaustive (for the squamate terminology that relates to cranial nerves and blood vessels [line 104] I would personally have consulted Oelrich, T.M., 1956. The Anatomy of the Head of Ctenosaura pectinata (Iguanidae). Misc. Publ. Museum of Zoology, University of Michigan, No. 94 (available online). The illustrations are excellent and informative. In that sense, the paper is publishable as is (once it has been sent through the spell-check one more time). It also conforms in style and substance to other recent analyses of turtle cranial morphology and phylogeny, as exemplified by the frequently cited paper by Evers and Benson (2019). But ever since the groundbreaking work by Gaffney (1972, 1975), there remains a big pink elephant in the room: the correlation between osteological characteristics and inferred soft anatomy relies on preciously thin ice, with only a few studies investigating a limited number of taxa providing any guidance to inferred correlations (most importantly the studies of Albrecht [1967, 1976], investigating the arterial vascularization of the turtle head based on the dissection of injected specimens). As the authors note (line 275): “As our primary source of information are skulls, we are not able to observe the cranial circulation and innervation systems directly” – but they still base far-reaching conclusions about character evolution on these latter systems. There is no ‘quick fix’ of this situation – short of a time-consuming investigation of serially sectioned and properly stained turtle heads.

·

Basic reporting

- Generally the manuscript is very well written and easy to read. Some typographical/grammatical errors are indicated in the annotated PDF
- The relevant literature is well-reviewed and relevant. How the authors’ work agrees or disagrees with that of others is well-documented
- Structure conforms to PeerJ standard. While this is a very long descriptive piece, I don’t believe it can be substantially shortened without compromising quality.
- Figures are relevant. See annotated PDF for comments, and please pay close attention to the following:
o Throughout, the authors use a red-green-blue colour scheme to refer to canals and their contents. Please be aware that this colour scheme may be difficult for readers who are colour-blind; consider a friendlier colour scheme
o Fig 19 is hard to read, and the dashed lines are almost invisible on the review copy at 100% magnification. Further, the legend does not indicate the meaning of the colours on Panel B.
- Raw data are not supplied, but these constitute CT scans which are mostly held on public repositories (MorphoSource and Digimorph), with a few exceptions. It is not clear if those not publicly deposited are available from the authors.

Experimental design

- This is a descriptive piece, but as such constitutes original primary research. While the work builds on that of others, it goes well beyond the most recent literature on the topic and is thus both novel and valuable.
- The research fills a knowledge gap by substantially broadening the taxonomic coverage for our understanding a key feature of turtle morphology that is relevant for systematic and evolutionary studies, using higher quality imaging and therefore better describing key characters within this system.
- The work is rigorous and of a high standard
- I have suggested moving a portion of the Discussion to the Methods to improve clarity.

Validity of the findings

- Findings are valid and statistical techniques are appropriate.
- I would like to see clarification about how the possible contribution of allometric variation might be affecting these results, however. Allometry is not mentioned at all, but is well understood to affect turtle skull shape (e.g. Claude J. et al. 2004. Ecological Correlates and Evolutionary Divergence in the Skull of Turtles: A Geometric Morphometric Assessment, Syst Biol 53(6): 933–948, https://doi.org/10.1080/10635150490889498). I would be surprised if it isn’t in play here, and this should be confirmed or ruled out. Indeed, it may add further complexity (and evolutionary interest) to the results.
- The Results section implies a discussion of canal development on the basis of a highly asymmetrical specimen. I looked forward to this discussion, and did not really find it. More would be useful here, especially since trouble was taken to figure this interesting specimen. Epigenetic interactions are known to structure development of cranial canals and vasculature, and it may be worth some discussion of how that might relate to this specimen and to the variability present in the patterns you have observed. See for example Jamniczky HA, Hallgrímsson B. 2011. Modularity in the skull and cranial vasculature of laboratory mice: implications for the evolution of complex phenotypes. Evol Dev. 2011;13(1):28-37. doi:10.1111/j.1525-142X.2010.00453.x. While this work is in mice and therefore tangential, the idea that much variation in these systems may not necessarily be phylogenetic is likely worth exploring here.
- Conclusions drawn are appropriate to the data presented (but see above for clarification regarding effects of allometry on canal size).

Additional comments

Congratulations to the authors on a very large amount of work and a very nice dataset to address the continuing importance of the cranial canal networks in turtles. This is a very extensive re-examination of ideas proposed more than 40 years ago and makes some important revisions to our understanding of this system. The authors make good use of state-of-the-art 3D imaging and reconstruction tools in order to provide new insights that will be useful for the study of turtle evolution. The paper is well written and easy to read (thank you!), and contains useful figures with only minor corrections required (see below). Minor criticisms relate to the fullness of the discussion regarding canal development, where I was left disappointed after being shown the tantalizingly asymmetrical Manouria specimen; and the lack of discussion of allometric effects on canal size that may confound studies of cross-sectional area. This second point in particular should be addressed before publication.

·

Basic reporting

This manuscript presents a comprehensive set of reconstructions, descriptions, and comparisons of the carotid circulation and (some) cranial nerves for 66 species of turtles, supplemented by quantitative and statistical evaluations of relative arterial sizes that supply blood to the turtle head. Even though the manuscript is long, it is not unnecessarily long, being full of detailed and useful data, easy to understand, and a pleasant read for turtle specialists. I have only some very minor suggestions in the manuscript (provided in a word file with marked changes). A more noteworthy comment is regarding the description of the spaces, canals, and foramina, in which, in a number of times, the authors refer the osseous structure solely to a soft tissue organ, which hampers its identification in fossils or macerated specimens. I'm sure for most of those cases you have osteological correlates that could be used to identify those structures, so I just suggested using those in this part.
In conclusion, this is an outstanding contribution to turtle anatomists and systematists, which can also be useful for experts in other taxa, which is well-written and provides very important data. As such, I congratulate the authors and recommend it for publication after a few minor revisions.

Experimental design

no comment

Validity of the findings

no comment

Additional comments

no comment

Reviewer 4 ·

Basic reporting

x

Experimental design

x

Validity of the findings

x

Additional comments

“A review of the carotid artery and facial nerve canal systems in extant turtles” is an impressive review of the cranial circulation in turtles, which has been a subject of extensive anatomical studies. This manuscript is a rich source of anatomical information and a welcome contribution. However, the description is impenetrably dense and frustratingly crunchy, and the discussion misses some key citations.

The authors are busy positioning their paper as a definitive account instead of letting their work speak for itself. They repeatedly make a claim, such as: “global comprehension was hindered by poor sampling and a lack of synthetic studies that addressed both systems (Line 16–17).” This is a false narrative. For example, Kuratani (1987) described the chondrocranial development of Caretta with extensive observation of both the carotid and facial nerve systems (J. Anat. 154, pp. 187-200). Rieppel (1990) described similar observations on Chelydra (Zool. J. Linn. Soc. 98, pp. 27-62). Their observations were picked up later by Miyashita (2012), who proposed that the spatial association between the internal carotid artery and vidian branch acts as a constraint, thus interactions among the carotid artery, facial nerve, and chondrocranium at early developmental stages control configuration of the foramina and canals (Gene Gaffney Festschrift, Springer). None of these papers was cited, even though their relevance to the authors’ discussion seems clear. This lack of attributing due credit to a long string of previous works is quite frustrating.

The authors provide quite puzzling and imprecise anatomical description by ignoring standard practice. Numerous examples caught my attention across the entire text.

- Artery irrigates, does not “support” (Line 94–95 and throughout) or “feed” (Line 626 and elsewhere). “Supply” may be acceptable but is unclear unless specific structure is named as “supplied” (supply brain, mandible, rostrum? An artery does not exactly supply a bony element, unless discussing capillaries invading the bone.) “Cranial arterial circulation (Line 34)” is simply cranial irrigation.
- Vein does not “pass” (Line 1033). Vein drains, as opposed to artery that irrigates.
- Sensory component of a nerve fires from their innervation to the ganglion. So it does not “leave” or “exit” a canal closer to innervation. If purely a sensory branch, the nerve may enter the canal from a point closer to innervation toward the ganglion. However, a motor neuron has a reverse polarity (it fires toward innervation). So avoid all polarity terms in general. The authors can say it is no longer enclosed by a particular bone, but fundamentally speaking it cannot “leave” or “start”.
- Artery or nerve does not pick itself up and “run” or “travel” (Line 112 and throughout). It is the observer’s focus that “runs” or “travels” on the structure. Artery or nerve extends.
- Artery (Line 83 and throughout) or ganglion (Line 697 and throughout) does not “give off” branches. Artery or ganglion has branches, splits into branches, or branch into (or leads/connects to) subsidiary structures.
- A canal itself has no inherent polarity so it does not “start” (Line 587 and elsewhere). It extends.
- Canals do not fuse (Lines 782-783). Canal is a space, and spaces do not fuse to each other. Canals may merge, join, or become confluent.


The rest of the description is not helpful. The practice of sticking to Latin terms in the turtle literature downstream of Gaffney always puzzled me. Up to 45% of characters in some sections of the manuscript pertain to those Latin terms like “canalis caroticus interni” and frankly these sections are non-comprehensible to most who needs the information. Curiously, the authors are inconsistent. They return to Anglicized common terms for soft tissues (e.g., “mandibular artery”) and some of these osteological features in Discussion. I do not see a rationale for this. Either use accessible English terms throughout (and provide the proper Latin terms in glossary in Nomenclature and Homology), or find some formatting scheme to distinguish Latinized terms, such as italicize (muscle names are commonly italicized), bold, or underline.

In addition to the imprecision and the terminology, the description is meandering, redundant, and inaccessible. Under each family, create subheadings by each main structure discussed (just like in Nomenclature and Homology) so the readers can find the information they are looking for. In more extreme form, the authors should even consider creating character shorthand and adopt telegraphic or bullet point style.

The meandering description exposes another fundamental issue: the authors present a rhetorical/semantic argument for many of the osteological correlates. Despite the rationales of homology presented at the outset, often I was unable to tell if the absence of a particular structure is purely of semantics (e.g., canal versus foramen, or how the homology of “canals” is interpreted when a nerve and a vessel extend endosseously as a bundle), pertains to configuration of the soft tissue interacting with the bone, or arises from the loss of the tissue altogether.

The first example is in Line 424-441, and it is a recurring issue through the rest of text. Or Line 542-545 is symptomatic: what would otherwise be identified as a cavernous canal is a groove within a canal and therefore considered “absent”, even though the difference is whether this space is fully enclosed within the bone or partly open. This only muddies anatomical understanding. Another extreme example is the podocnemidid fossa: correlates are absent but structural configurations remain generally same for the soft tissues.

In these parts of the ms, the description is so bone centric (apparently canals “fuse” according to the authors, Line 782-783) that the authors seem to disregard the fact that most structures they describe are merely correlates. That is, their identity (canals, foramina, cavities) is conferred by a tissue that the bone interacts with, so applying a blanket definition of homology to all is inappropriate. The authors should ask themselves if they are describing an anatomical state or a terminological state. When a canal is absent or present, it is contingent on spatial configuration of the soft tissue(s) filling the canal. To avoid confusion, the authors should present possible arrangement(s) of the carotid and facial systems first, and then interpret their osteological correlates (=identify canals and foramina with proper names), for each family.

At the very least, distinguish semantic/terminological issue versus homology issue. Line 72-74 does not make sense, for example. The identification of a canal where a vessel and a nerve co-occur is a terminological issue, not a homology issue (=pertaining to presence/absence of a structure). It is not that “inconsistent nomenclature […] makes a deeper understanding of homology difficult (Line 90–91).” It is the authors’ expectation for homology to be perfectly translated into nomenclature, that makes a deeper understanding difficult.

Discussion is valuable, but disorganized. Construct taxonomically segregated comparative table(s) to contrast which lineages have what state.

Figures are generally excellent, but Figures 16-18 are confusing. These describe “patterns” of entirely different nature, and distinguish them by Roman numerals, alphabet, and Arabic numerals. They are confusing both in figures and text. Please come up with a better system so we don’t have to get ourselves lost which pattern is discussed. Why can’t they be differentiated as “Facial Patterns 1a-2c” ” “Hyomandibular Subpatterns 1-3”, “Vidian Subpatterns 1-5”?

For description and figures, it is often difficult to verify the characters discussed in text against illustrations in the figures. Each character described in should have reference to relevant labels and panels of the figure.

The authors should provide basic schemes of cranial circulation and facial innervation in turtles at the outset, and refer to the schematics in Lines 77 onward. This is to introduce readers unfamiliar to this problem (likely >90% of pdf downloads) to the basic scheme of the anatomical system.

Line 214-272: This method description is unhelpful and non-repeatable. There is no mention of statistical procedure or dataset (it is available with the paper?). I cannot figure out how measurements were collected (e.g., where in the canals were diameters taken?), whether the measurements were corrected for body size or ontogenetic stages, etc. Please ensure repeatability in methods description.

The order in which families are treated in the description puzzles me. The authors should provide a tree showing their general relationships.

In multiple places in the manuscript (Line 1057, 1267, elsewhere) the authors “contradict” Jamniczky and Russell (2007). But the authors provide no explanation for the discrepancies. Jamniczky and Russell (2007) also measured the same canals in different specimens using CT scan. Why are these observations at odds with each other? Are their observations/mesurements still valid? Did they misidentify structures? Are these due to differences in methods? Different sections of the canals? Different ontogenetic stages?


Line 525 and throughout: when discussing canal sizes, is it in diameter? Is it corrected for differences in body size, or allometry, or ontogeny? Is it maximum, minimum, or average diameter?

Line 547, Line 550 “infer” — what ‘infers’ each of these?

Line 603-604: I do not understand what is “trifurcation formed by the prootic, pterygoid, and basisphenoid.”

Line 629: “The canalis cariticus basisphenoidalis is formed by the basisphenoid.” This sentence is awkward. It’s like reading “the brain is housed within the braincase.”

Line 631: “separated” from each other.

Line 778-779: “while the others either get lost in the porosity of the bone…”
This should be “while the others either become unidentifiable/untraceable in the porosity of the bone…”

Line 829: “mandibular artery traverses…”
Again, an artery does not travel, run, or traverse. In this case, it extends within the cavernous canal (laterally?).

Line 1034: Lateral head vein does not pass “anteriorly.” The lateral head vein drains, and the blood flow is toward the cardinal vein (posteriorly).

Line 1039: “The canalis nervus facialis exits the prootic…”
A canal does not exit bone. A nerve may (to the direction of firing polarity; opposite between sensory and motor).

Line 1041-1042: Another issue with rhetorical/nomenclatural homology. Is canal absent or merged with the carotid canal? Go back to my original comment for this and many other examples.

Line 1075-1076: “the anterior wall of the canalis cavernosus is extremely thin and that the “foramina” are elongate slits.”
This is a good place for the authors to consider how they are mixing up nomenclatural and anatomical states.

Line 1234: “The canal does not extend along the pterygoid/basisphenoid suture, as in all turtles with an unambiguous canalis caroticus lateralis.”
As in all turtles do, or do not?

Line 1239-1240: an artery does not “split” from a canal. It splits from another artery.

Line 1257: an artery does not “join” a chondrocranial element. It penetrates a chondrocranial element.

Line 1262-1267: The majority of characters in these two sentences pertain to Latin terms, and as written these sentences make zero sense to me.

Line 1411 “all extant turtles…”
This should be “almost all extant turtles…” (the authors immediately follow this sentence with ‘exceptions’).

Line 1415-1417: It is good that the authors clarify here about “a matter of semantic preference” but the problem is that they are absolutely vague about this where this clarification matters in Description.

Line 1413, 1419: plural/singular inconsistent in multiple words.

Line 1429-1432: Is only a bony canal considered a canal? Part of the canal wall unossified and it becomes something else? The authors should remind themselves that canal is not the variation. It is a substate of the presence/absence of ossification in that particular locale, that varies.

Line 1446-1448: “A canalis caroticus lateralis is furthermore absent in chelonioids, but not a result of the absence of the palatine artery, but due to its anterior displacement.”
Again, it is nice of the authors to clarify, but this has to be explicit in description. It is vague in the original section, and therefore highly confusing.

Line 1453: what is a “trough let”?

Line 1456-1459: “Our re-interpretation is more parsimonious…”
In the absence of tree, I am not able to evaluate this claim, or what is the difference between Hermanson et al.’s interpretation and their re-interpretation.

Line 1593-1604: Informative and valuable paragraph. A perfect example where a comparative table would greatly enhance the communication.

Line 1620 onward: This is informative but meandering, long, and redundant. Enhance with a comparative table.

Line 1728: “internal carotid and palatine artery canals…”
A sudden introduction of Anglicized terms. Inconsistent.

Line 1828: “higher ‘medial’ blood flow”
“greater ‘medial’ blood flow”

Line 1846-1848: “In summary, our canal size data quantitatively confirm that variation in the stapedial artery size is countered by oppositional variation in the internal carotid size and that variation from the mean can partially be explained by the placement of the mandibular artery.”
This should be: “In summary, our canal size data quantitatively confirm that variation in the stapedial artery inversely correlates with the internal carotid size and that departure from the mean can partially be explained by the configuration of the mandibular artery.

Line 1877 “basal stem-turtles”
I don’t know what this means.

Line 1902-1904: “fossil taxa belonging to stem lineages of extant clades of Testudines”
Awkward.

Line 1911: “slight nomenclatural updates”
Slight nearly equals none. But the authors propose many changes.

---

## Round 0.2 · Minor Revisions

Please address or reply to the points raised by Reviewer 4. I will make my final decision rapidly after your resubmission.

·

Basic reporting

The authors have made substantial changes in response to mine and the other reviewers' comments. These changes are adequate to respond to my concerns. While I am not in position to re-read the entire manuscript closely, I am satisfied that the changes have been made that allow publication of this version.

Experimental design

No comment.

Validity of the findings

No comment.

Additional comments

No additional comment; see above.

·

Basic reporting

The authors satisfactorily replied to all of my comments on the original manuscript and made the pertinent modifications. I have nothing further to add and am content with the resulting manuscript and suggest its publication as it is.

Experimental design

no further comment

Validity of the findings

no further comment

Reviewer 4 ·

Basic reporting

The authors addressed many of the comments and concerns raised. For a few items, however, they relied on rhetorical arguments instead of providing evidence.

In their response they argue their purpose is to describe "the evolutionary development of the carotid and facial nerve system" while they "in no way claim to contribute to the question as how these systems originate or interplay during ontogeny."
"Evolutionary development" means phylogenetic investigation of development (ontogeny). In that sense, once again, it is puzzling why they choose not to include the studies of turtle chondrocranial development in which these systems are described together (such as those on Caretta and Chelydra) and considered in the context of development.

They proceed to attribute this to the misunderstanding arising from the word "global" and revised this as:

"Although a significant number of studies related to these structures exists, A BROADER comprehension OF VARIATION ACROSS THE TREE HAS BEEN hindered by poor sampling and a lack of synthetic studies that addressed both systems together"
(UPPERCASE refers to their change)

Simple substitution of words accomplishes little here, and the same problem remains. Why is the structure that has generated "a significant number of studies" so "poorly sampled" at the same time? And to the point of "the lack of synthetic studies that addressed both systems together", there are studies that addressed these systems together, but these works are either not cited or relegated to the sideline along with other previous works.

Is this the "synthetic" study that the authors consider has been "lacking" in the literature, or is this a compendium of anatomical information? Or is this a follow-up to the series of works that has a deep historical root?

I repeat that arteries do not typically "supply" a bone. So the arterial blood supply for the face is correct, but "the arterial blood supply for the facial region of the skull" (Line 225) is not. Examples of this usage persist in Line 1922, 1931, 1938, 1949, 1953, 1995, .

Line 1949-1950: "these arterial reductions affect the blood supply by shifting the origin of arteries supplying the anterior skull region or the mandible to whichever artery is not reduced."
This sentence does not make sense. Please revise.

I do not intend this to be the place to fight this issue of Latinized terms in turtle literature (for example, chondrocrania are also full of them, even though they are used widely across different taxa and therefore more accessible), and the authors have clearly spoken in defence of the common practice in that field. However, if the authors wish to deliver the turtle data to a wider audience so they are also useful in a broader context, they should also be aware that these rather insular practices are not exactly helping that cause.

Line 2042: "basal-stem turtles"
"Basal" and "stem" are redundant. "Basal" is also imprecise. Why not simply stem turtles, or, to be more precise, non-XXX stem turtles?

Line 2067-2068: "fossil taxa belonging to stem lineages of the primary crown clades of Testudines"
This is as clear as mud. What are "primary crown clades of Testudines"? Why can't this be "stem cryptodires" "paracryptodires" etc.? Or simply "fossil taxa within the testudine crown"?

Experimental design

No comment.

Validity of the findings

No comment.

---

## Round 0.3 · accepted · Accept

I am pleased to confirm that your MS has been accepted for publication.